# THE EXPRESSIVE POWER OF LOW-RANK ADAPTATION

**Yuchen Zeng**
Department of Computer Science
University of Wisconsin-Madison
yzeng58@wisc.edu

**Kangwook Lee**
Department of Electrical and Computer Engineering
University of Wisconsin-Madison
kangwook.lee@wisc.edu

## ABSTRACT

*Low-Rank Adaptation* (LoRA), a parameter-efficient fine-tuning method that leverages low-rank adaptation of weight matrices, has emerged as a prevalent technique for fine-tuning pre-trained models such as large language models and diffusion models. Despite its huge success in practice, the theoretical underpinnings of LoRA have largely remained unexplored. This paper takes the first step to bridge this gap by theoretically analyzing the expressive power of LoRA. We prove that, for fully connected neural networks, LoRA can adapt any model $f$ to accurately represent any smaller target model $\bar{f}$ if LoRA-rank $\geq$ (width of $f$) $\times \frac{\text{depth of } \bar{f}}{\text{depth of } f}$, under a mild assumption. We quantify the approximation error when the LoRA-rank is lower than the threshold. For Transformer networks, we show any model can be adapted to a target model of the same size with rank-($\frac{\text{embedding size}}{2}$) LoRA adapters. Our study reveals numerous theoretical insights on hyperparameter tuning and algorithm development for LoRA, all of which are empirically validated.

## 1 INTRODUCTION

Recent foundation models, such as large language models (OpenAI, 2023; Liu et al., 2019; Touvron et al., 2023), have achieved remarkable success in a wide range of applications. Due to their substantial size, the standard full fine-tuning approach—where all the model's parameters are updated for specialized tasks—is becoming increasingly difficult and inefficient. This leads to the growing popularity of parameter-efficient fine-tuning approaches (Hu et al., 2022a; Liu et al., 2022b; Ben Zaken et al., 2022; Hu et al., 2022b). Instead of updating all parameters, these approaches selectively update smaller subsets of weights or introduce lightweight adapters, thereby greatly decreasing the computational and storage costs.

The most dominant approach along this line is *Low-Rank Adaptation* (LoRA) (Hu et al., 2022a), which employs lightweight low-rank adapters to pre-trained weight matrices. Far from merely enhancing computational efficiency, empirical evidence has shown that LoRA can match or even exceed the performance of full fine-tuning (Hu et al., 2022a). To date, LoRA has been widely used and achieved considerable success in adapting large language models (Hu et al., 2022a; Dinh et al., 2022b) and image generation models (Ryu, 2023; Fan et al., 2023) for various downstream tasks.

Despite the empirical success of LoRA, little is known in theory about how it works. A notable exception (Malladi et al., 2023) showed that LoRA finetuning is approximately equivalent to full fine-tuning in the lazy regime. However, many theoretical questions remain open, such as: What is the minimum rank of the LoRA adapters required to adapt a (pre-trained) model $f$ to match the functionality of the target model $\bar{f}$? How does the model architecture (i.e., depth, width) affect the minimal rank? If the adapter rank is lower than this threshold, what is the resulting approximation error? Answering such questions will provide important theoretical insights into when and why LoRA achieves effective adaptation.

**Our Contributions.** In this paper, we present the first set of theoretical results that characterize the expressive power of Low-Rank Adaptation (LoRA) for Fully Connected Neural Networks (FNN) and Transformer Networks (TFN). In particular, we identify the necessary LoRA-rank for adapting a frozen model to exactly match a target model. For FNN cases, we also establish the required LoRA-rank for closely approximating the target model when a small approximation error is allowed. Our work focuses solely on the expressive power of the model with low-rank adapters, i.e., we show that under which conditions, effective low-rank adapters exist for the given adaptation task. This excludes other aspects such as optimization and generalization.

We now present the essence of our main theoretical findings in the following informal statement.

**Theorem 1** (Informal). *Let $\bar{f}$ be a target FNN and $f_0$ be an arbitrary frozen FNN. Under mild conditions on ranks and network architectures, there exist low-rank adapters such that a low-rank adapted version of $f_0$ is exactly equal to $\bar{f}$.*

We present the detailed formulations of Theorem 1 under two scenarios: (i) applying a uniform rank across all LoRA adapters, as detailed in Theorem 3 and the specialized instance Corollary 4 for randomly drawn frozen and target models; and (ii) allowing different ranks applied to each LoRA adapter, as described in Theorem 6. To the best of our knowledge, this is the first known theoretical results on the expressive power of LoRA. While this informal theorem is for exact approximation, we also derive the approximation bounds as well, i.e., we characterize the approximation error between the finetuned model and the target model as a function of the LoRA-rank, as provided in Theorem 5 for the uniform LoRA-rank scenario and Theorem 6 for general cases. Furthermore, within the same framework, we investigate the expressive power of tuning the final layers for randomly generated frozen models, as described in Lemma 4. This result allows us to contrast LoRA and final layer tuning, thereby providing insights for future algorithm development.

We summarize our main findings on TFN in the following informal theorem.

**Theorem 2** (Informal). *Let $\bar{f}$ be the target TFN and $f_0$ be the frozen TFN. Under mild conditions on ranks and network architectures, there exist low-rank adapters for attention weight matrices such that a low-rank adapted version of $f_0$ is exactly equal to $\bar{f}$.*

The formal statement of Theorem 2 is provided in Theorem 7, with a specialized version in Corollary 10 tailored for randomly generated frozen and target models.

In Sec. 5 and G, we perform experiments on both synthetic and real datasets to substantiate our theoretical results, demonstrating the practical applicability of our theoretical findings in algorithm development and hyperparameter tuning.

## 1.1 RELATED WORKS

**Expressive Power of Neural Networks**   Theoretical study of the expressive power of unfrozen neural networks has progressed since the first universal approximation theorem (Hornik et al., 1989), showing that sufficient network width and depth can guarantee function approximation (Bengio & Delalleau, 2011; Eldan & Shamir, 2016; Liang & Srikant, 2017). Many recent studies obtained similar results for deep neural networks with modern twists such as ReLU activations and Transformer networks (Yun et al., 2020a; Raghu et al., 2017; Telgarsky, 2016; 2015; Bietti & Bach, 2021; Oymak et al., 2023; Lee et al., 2017; Shen & Zhang, 2020; Likhosherstov et al., 2021; Hsu et al., 2021; Park et al., 2021; Yun et al., 2020b; Giannou et al., 2023b). Metrics like Vapnik-Chervonenkis and Rademacher complexities (Vapnik & Chervonenkis, 2015; Bartlett & Mendelson, 2001) assess classification capacity. However, these theories *cannot* fully explain the performance of frozen neural networks as they generally cannot factor in pre-trained model parameters and adaptation methods.

**Expressive Power of Adaptation Methods**   In stark contrast to the flourishing research on the expressive power of neural networks, there exists a limited number of works investigating the expressive power of adaptation methods. A notable exception is Giannou et al. (2023a), investigating the expressive power of normalization parameter fine-tuning. They demonstrate that fine-tuning the normalization layers alone can adapt a randomly initialized ReLU network to match any target network that is $O(\text{width})$ times smaller. We borrow some proof techniques from this work, including techniques for extending results from linear neural networks to ReLU neural networks. In another recent work (Englert & Lazic, 2022), the authors show that *neural reprogramming* (Elsayed et al., 2019; Engel et al., 2018; Lee et al., 2020; Dinh et al., 2022a; Chen, 2022), a technique that modifies only the inputs while keeping the pretrained network frozen, can adapt any random two-layer ReLU network to achieve arbitrarily high accuracy on a Bernoulli data model over hypercube vertices. Despite these early attempts, no existing study has yet explored the expressive power of LoRA, the current leading adaptation method.

A more detailed discussion of related works is provided in Sec. B.

## 1.2 NOTATIONS

Define $[N] := \{1, 2, \ldots, N\}$. Let the operators $\wedge$ and $\vee$ denote the minimum function and the maximum function, respectively. We use $\boldsymbol{I}$ to represent the identity matrix.

For a sequence of $L$ matrices $(\boldsymbol{W}_l)_{l=1}^L$, we simplify the product of these matrices $\boldsymbol{W}_L\boldsymbol{W}_{L-1}\cdots\boldsymbol{W}_1$ as $\prod_{l=1}^L \boldsymbol{W}_l$, with matrices multiplied in descending order from $\boldsymbol{W}_L$ to $\boldsymbol{W}_1$. When $m > n$, we define $\sum_{i=m}^n a_i = 0$ and $\prod_{i=m}^n a_i = 1$ for scalars $(a_i)_{i=m}^n$, and $\sum_{i=m}^n \boldsymbol{W}_i = \boldsymbol{O}$ and $\prod_{i=m}^n \boldsymbol{W}_i = \boldsymbol{I}$ for square matrices $(\boldsymbol{W}_i)_{i=m}^n$.

*Singular Value Decomposition* (SVD) of the matrix $\boldsymbol{W}$ can be expressed as $\boldsymbol{W} = \boldsymbol{U}\boldsymbol{D}\boldsymbol{V}^\top$, where $\boldsymbol{U}, \boldsymbol{V} \in \mathbb{R}^{D\times D}$ are orthonormal matrices and $\boldsymbol{D} \in \mathbb{R}^{D\times D}$ is a diagonal matrix. The singular values, sorted in descending sequence, are represented on the diagonal of $\boldsymbol{D}$, denoted as $\sigma_1(\boldsymbol{W}) \geq \sigma_2(\boldsymbol{W}) \geq \cdots \geq \sigma_D(\boldsymbol{W}) \geq 0$, where $\sigma_d(\boldsymbol{W})$ denotes the $d$-th largest singular value for all $d \in [D]$. When $d > D$, $\sigma_d(\boldsymbol{W})$ is defined as zero. The best rank-$r$ approximation (in the Frobenius norm or the 2-norm) of $\boldsymbol{W}$ is $\sum_{i=1}^r \sigma_i \boldsymbol{u}_i \boldsymbol{v}_i^T$, where $\boldsymbol{u}_i$ and $\boldsymbol{v}_i$ are the $i$-th column of $\boldsymbol{U}$ and $\boldsymbol{V}$, respectively (Eckart & Young, 1936; Mirsky, 1960). We denote this best rank-$r$ approximation by $\mathrm{LR}_r(\boldsymbol{W})$, where LR is a shorthand for "Low-Rank". When $r \geq \mathrm{rank}(\boldsymbol{W})$, it is clear that $\mathrm{LR}_r(\boldsymbol{W}) = \boldsymbol{W}$. Occasionally, the subscript $r$ may be omitted to indicate a general low-rank approximation without specifying the rank.

## 2 WARM UP: EXPRESSIVE POWER OF LINEAR MODELS WITH LoRA

Before delving into the expressive power of LoRA for FNN and TFN, we begin by investigating the simplest scenario: both the target model $\bar{f}$ and the frozen model $f_0$ are linear, i.e.,

$$\text{Target Model}\quad \bar{f}(\boldsymbol{x}) = \overline{\boldsymbol{W}}\boldsymbol{x}, \qquad \text{Frozen Model}\quad f_0(\boldsymbol{x}) = \boldsymbol{W}_L\cdots\boldsymbol{W}_1\boldsymbol{x} = \left(\prod_{l=1}^L \boldsymbol{W}_l\right)\boldsymbol{x}.$$

This problem serves as a simplified version of approximating a target FNN, where the target model $\bar{f}$ has a single layer, the frozen model $f_0$ has $L$ layers, all bias vectors in both two models are zero, and the activation functions are linear. Throughout this paper, for the sake of simplicity, we will assume that both models have the same number of neurons in each layer, i.e., $\overline{\boldsymbol{W}}, \boldsymbol{W}_1, \ldots, \boldsymbol{W}_L \in \mathbb{R}^{D\times D}$. Nevertheless, our results are readily extendable to situations where the frozen model is wider than the target model, which is a more natural setting as the frozen models are often overparameterized to ensure high capacity and good performance across diverse tasks in practice. See the discussion in Sec. H for more details.

The objective here is to incorporate low-rank adapters into the frozen model so that the adapted model can effectively approximate the target model. Unless otherwise specified, we always consider a uniform LoRA-rank for all low-rank adapters throughout this paper. For a given LoRA-rank $R \in [D]$, we apply LoRA adapters $\Delta\boldsymbol{W}_1, \ldots, \Delta\boldsymbol{W}_L$ to the frozen model, and the adapted model can be represented as

$$\text{Adapted Model}\quad f(\boldsymbol{x}) = (\boldsymbol{W}_L + \Delta\boldsymbol{W}_L)\cdots(\boldsymbol{W}_1 + \Delta\boldsymbol{W}_1)\boldsymbol{x},$$

where $\mathrm{rank}(\Delta\boldsymbol{W}_l) \leq R$ for all $l \in [L]$.

Since the frozen model and adpated model are all linear, we can focus on quantifying the discrepancy between the linear coefficients, i.e., $\prod_{l=1}^L (\boldsymbol{W}_l + \Delta\boldsymbol{W}_l) - \overline{\boldsymbol{W}}$. In the subsequent lemma, we establish the minimal achievable norm, and identify the smallest LoRA-rank required for the adapted model to exactly represent the target model, i.e., $f = \bar{f}$, under a non-singularity assumption. We will demonstrate in Sec. 3.3 that this non-singularity assumption is mild, as it can be satisfied even by randomly generated weight matrices.

**Lemma 1.** *Define error matrix $\boldsymbol{E} := \overline{\boldsymbol{W}} - \prod_{l=1}^L \boldsymbol{W}_l$, and denote its rank by $R_{\boldsymbol{E}} = \mathrm{rank}(\boldsymbol{E})$. For a given LoRA-rank $R \in [D]$, assume that all the weight matrices of the frozen model $(\boldsymbol{W}_l)_{l=1}^L$, and $\prod_{l=1}^L \boldsymbol{W}_l + \mathrm{LR}_r(\boldsymbol{E})$ are non-singular for all $r \leq R(L-1)$. Then, we have the following:*

$$\min_{\Delta\boldsymbol{W}_l:\mathrm{rank}(\Delta\boldsymbol{W}_l)\leq R}\left\|\prod_{l=1}^L (\boldsymbol{W}_l + \Delta\boldsymbol{W}_l) - \overline{\boldsymbol{W}}\right\|_2 = \sigma_{RL+1}(\boldsymbol{E}).$$

*Thus, when $R \geq \lceil \frac{R_{\boldsymbol{E}}}{L} \rceil$, the optimal solution satisfies $\prod_{l=1}^L (\boldsymbol{W}_l + \Delta\boldsymbol{W}_l) = \overline{\boldsymbol{W}}$, implying $f = \bar{f}$.*

*Proof Sketch.* We start the proof by noting that the distance between the adapted and target models

$$\left\|\prod_{l=1}^L (\boldsymbol{W}_l + \Delta\boldsymbol{W}_l) - \overline{\boldsymbol{W}}\right\|_2 = \left\|\left(\prod_{l=1}^L (\boldsymbol{W}_l + \Delta\boldsymbol{W}_l) - \prod_{l=1}^L \boldsymbol{W}_l\right) - \left(\overline{\boldsymbol{W}} - \prod_{l=1}^L \boldsymbol{W}_l\right)\right\|_2.$$

The remaining proof aims to minimize the right-hand side under the constraint $\mathrm{rank}(\Delta \boldsymbol{W}_l) \leq R$ for all $l \in [L]$. The basic idea here is to match $\prod_{l=1}^{L}(\boldsymbol{W}_l + \Delta \boldsymbol{W}_l) - \prod_{l=1}^{L} \boldsymbol{W}_l$ with the best rank-$r$ approximation of $\overline{\boldsymbol{W}} - \prod_{l=1}^{L} \boldsymbol{W}_l$. The key steps to solve this problem are as follows.

1. Demonstrate that $\prod_{l=1}^{L}(\boldsymbol{W}_l + \Delta \boldsymbol{W}_l) - \prod_{l=1}^{L} \boldsymbol{W}_l$ can be decomposed into $L$ terms:

   $$\textstyle\prod_{l=1}^{L}(\boldsymbol{W}_l + \Delta \boldsymbol{W}_l) - \prod_{l=1}^{L} \boldsymbol{W}_l = \sum_{l=1}^{L} \left( \prod_{i=l+1}^{L} \boldsymbol{W}_i \right) \Delta \boldsymbol{W}_l \left( \prod_{i=1}^{l-1}(\boldsymbol{W}_i + \Delta \boldsymbol{W}_i) \right).$$

   Since $\mathrm{rank}(\Delta \boldsymbol{W}_l) \leq R$, it follows that $\mathrm{rank}\left( \prod_{l=1}^{L}(\boldsymbol{W}_l + \Delta \boldsymbol{W}_l) - \prod_{l=1}^{L} \boldsymbol{W}_l \right) \leq RL.$

2. Consider the rank-$RL$ approximation of $\overline{\boldsymbol{W}} - \prod_{l=1}^{L} \boldsymbol{W}_l$. Decompose this low-rank approximation into $L$ terms $\sum_{l=1}^{L} \boldsymbol{E}_l$ such that $\mathrm{rank}(\boldsymbol{E}_l) \leq R$, where $\boldsymbol{E}_l$'s will be determined later.

3. To match $\prod_{l=1}^{L}(\boldsymbol{W}_l + \Delta \boldsymbol{W}_l) - \prod_{l=1}^{L} \boldsymbol{W}_l$ with the rank-$RL$ approximation of $\overline{\boldsymbol{W}} - \prod_{l=1}^{L} \boldsymbol{W}_l$, we let $\left( \prod_{i=l+1}^{L} \boldsymbol{W}_i \right) \Delta \boldsymbol{W}_l \left( \prod_{i=1}^{l-1}(\boldsymbol{W}_i + \Delta \boldsymbol{W}_i) \right) = \boldsymbol{E}_l$ by choosing $\Delta \boldsymbol{W}_l = \left( \prod_{i=l+1}^{L} \boldsymbol{W}_i \right)^{-1} \boldsymbol{E}_l \left( \prod_{i=1}^{l-1}(\boldsymbol{W}_i + \Delta \boldsymbol{W}_i) \right)^{-1}.$

4. Select appropriate $(\boldsymbol{E}_l)_{l=1}^{L}$ such that $\boldsymbol{W}_i + \Delta \boldsymbol{W}_i$ are invertible for $i \in [L]$. $\qquad\square$

The complete proof and the explicit construction of optimal LoRA adapters, are detailed in Sec. D.

In fact, this lemma delivers a crucial insight. When we consider $L = 1$ and $R = D$, the lemma becomes strikingly similar to the Eckart–Young–Mirsky theorem (Eckart & Young, 1936; Mirsky, 1960). However, there is a significant difference from the classical theorem on the optimal low-rank approximation, which involves a single target matrix and a single matrix as an optimization variable. Our lemma demonstrates that a comparable result can be achieved for a "product of matrices," where each matrix is optimized subject to a low-rank constraint. That being said, even though each matrix is constrained by a low rank, the "effective rank" is the sum of these low ranks, i.e., in this scenario, is $LR$. Consequently, once the low-rank adapters are optimally configured, one can make the product equal to the best rank $LR$-approximation of the target matrix. This can be viewed as an extension of the matrix approximation theorem to a product of matrices, each subject to low-rank constraints. Our main theoretical results on the expressive power of LoRA, which we will present in the subsequent sections, will build upon this core matrix approximation result.

## 3 EXPRESSIVE POWER OF FNNs WITH LoRA

### 3.1 PROBLEM SETTING

We use $\mathrm{FNN}_{L,D}(\cdot; (\boldsymbol{W}_l)_{l=1}^{L}, (\boldsymbol{b}_l)_{l=1}^{L})$ to denote a $L$-layer width-$D$ fully connected ReLU neural network with weight matrices $\boldsymbol{W}_l \in \mathbb{R}^{D \times D}$ and biases $\boldsymbol{b}_l \in \mathbb{R}^D$, where $l \in [L]$. The target FNN $\overline{f}$ and frozen FNN $f_0$ can be represented as follows:

Target FNN $\overline{f} := \mathrm{FNN}_{\overline{L},D}(\cdot; (\overline{\boldsymbol{W}}_l)_{l=1}^{L}, (\overline{\boldsymbol{b}}_l)_{l=1}^{L})$, Frozen FNN $f_0 := \mathrm{FNN}_{L,D}(\cdot; (\boldsymbol{W}_l)_{l=1}^{L}, (\boldsymbol{b}_l)_{l=1}^{L})$,

where $\overline{\boldsymbol{W}}_l \in \mathbb{R}^{D \times D}$ and $\overline{\boldsymbol{b}}_l \in \mathbb{R}^D$ represent the weight matrix and bias vector for the $l$-th layer of the target model $\overline{f}$, respectively. Likewise, $\boldsymbol{W}_l \in \mathbb{R}^{D \times D}$, $\boldsymbol{b}_l \in \mathbb{R}^D$ are those for $f_0$, for layer $l \in [L]$. Given a specified LoRA-rank $R \in [D]$, we adapt the frozen FNN $f_0$ into a new model $f$ via LoRA. The adapted model $f$ is defined as

$$\text{Adapted FNN} \quad f := \mathrm{FNN}_{L,D}(\cdot; (\boldsymbol{W}_l + \Delta \boldsymbol{W}_l)_{l=1}^{L}, (\widehat{\boldsymbol{b}}_l)_{l=1}^{L}),$$

where the weight matrix for the low-rank adapter $\Delta \boldsymbol{W}_l \in \mathbb{R}^{D \times D}$ satisfies specified rank constraints, updated bias vector $\widehat{\boldsymbol{b}}_l \in \mathbb{R}^D$ for $l \in [L]$[1].

As noted in Sec. 2, it is common for the pretrained model to be larger than necessary. Therefore, we focus on a setting where the frozen model is deeper than the target model, i.e., $L \geq \overline{L}$. Furthermore, in this section, we let the input space $\mathcal{X} \in \mathbb{R}^{D \times D}$ be bounded.

---

[1]We consider the case where the bias parameters can also be updated, as suggested by Hu et al. (2022a). Experiments investigating the impact of updating bias parameters are presented in Sec. G.5.

### 3.2 ONE-LAYER ReLU FNN APPROXIMATION

We start with investigating the expressive power of LoRA on one-layer FNN. In this setting, our aim is to identify LoRA adapters $(\Delta \boldsymbol{W}_l)_{l=1}^L$ and bias vectors $(\widehat{\boldsymbol{b}}_l)_{l=1}^L$ such that the adapted model

$$\texttt{ReLU}((\boldsymbol{W}_L + \Delta \boldsymbol{W}_L) \cdot \texttt{ReLU}((\boldsymbol{W}_{L-1} + \Delta \boldsymbol{W}_{L-1}) \cdot \texttt{ReLU}(\cdots) + \widehat{\boldsymbol{b}}_{L-1}) + \widehat{\boldsymbol{b}}_L)$$

closely approximates the target one-layer ReLU FNN model $\texttt{ReLU}(\overline{\boldsymbol{W}}_1 \cdot + \bar{\boldsymbol{b}}_1)$.

This differs from the setting described in Sec. 2, where a multi-layer FNN with linear activation functions and zero biases was used to approximate a one-layer FNN with the same properties. In the current setting, we introduce non-linearity through the use of ReLU activation functions in the frozen model and also take biases into account. Consequently, to generalize the findings to this new setting, addressing the introduced non-linearity due to the ReLU activation functions in the frozen model is the main challenge.

We employ the following two steps to extend the results in Sec. 2 to the current setting.

1. (Linearization) We eliminate the nonlinearity in the first $L-1$ layers of the adapted model, making it equivalent to a one-layer ReLU FNN. This can be readily achieved by choosing sufficiently large bias vectors for the first $L-1$ layers to ensure that all ReLUs in these layers are activated. This technique of eliminating non-linearity is inspired by Giannou et al. (2023a).

2. (Weight Matrix Alignment) We update the bias vectors of the last layer $\widehat{\boldsymbol{b}}_L$ to align with that of the target model $\bar{f}$, and apply the linear model approximation results (i.e., Lemma 1) to identify the low-rank adapters that match the weight matrix $\bar{f}$.

Following the steps above, we arrive at the subsequent lemma, which demonstrates that any one-layer FNN can be closely approximated by a multi-layer FNN finetuned via LoRA. The complete proof is provided in Sec. E.1.

**Lemma 2.** *Define error matrix $\boldsymbol{E} := \overline{\boldsymbol{W}}_1 - \prod_{l=1}^L \boldsymbol{W}_l$, with its rank represented by $R_{\boldsymbol{E}} = \text{rank}(\boldsymbol{E})$. Consider a LoRA-rank $R \in [D]$. Assume that the weight matrices $\boldsymbol{W}_1, \ldots, \boldsymbol{W}_L \in \mathbb{R}^{D \times D}$ and $\prod_{l=1}^L \boldsymbol{W}_l + \text{LR}_r(\boldsymbol{E})$ for all $r \le R(L-1)$ are non-singular. Let $\mathbf{x}$ be a random input sampled from a distribution with bounded support $\mathcal{X}$ and let $\Sigma = \mathbb{E}\mathbf{x}\mathbf{x}^\top$. Then, there exists rank-$R$ or lower matrices $\Delta \boldsymbol{W}_1, \ldots, \Delta \boldsymbol{W}_L \in \mathbb{R}^{D \times D}$ and bias vectors $\widehat{\boldsymbol{b}}_1, \ldots, \widehat{\boldsymbol{b}}_L \in \mathbb{R}^D$ such that the expected squared error can be bounded as*

$$\mathbb{E} \left\| f(\mathbf{x}) - \bar{f}(\mathbf{x}) \right\|_2^2 \le \|\Sigma\|_F \, \sigma_{RL+1}^2(\boldsymbol{E}).$$

*Moreover, when $R \ge \lceil \frac{R_{\boldsymbol{E}}}{L} \rceil$, we have $f(\boldsymbol{x}) = \bar{f}(\boldsymbol{x})$ for all $\boldsymbol{x} \in \mathcal{X}$.*

### 3.3 MULTI-LAYER ReLU FNN APPROXIMATION

We now generalize our discussion to the approximation of multi-layer ReLU FNNs. The key strategy for extending the results to approximating multi-layer ReLU FNNs under LoRA is model partition, inspired from Giannou et al. (2023a). To elucidate this, we start with a specific example.

**Example 1.** *Consider the case where $\bar{L} = 2$ and $L = 4$. We view a two-layer target model $\bar{f}$ as a composition of two one-layer ReLU FNNs. Accordingly, we partition the four-layer adapted model $f$ into two submodels, each consisting of two layers. For each layer in the target model, we utilize two corresponding layers in the frozen/adapted model for approximation. This problem then simplifies into a one-layer FNN approximation problem, which has already been addressed in Lemma 2.*

Based on this example, we introduce a ordered partition $\mathcal{P} = \{P_1, \ldots, P_{\bar{L}}\}$ to partition the layers in the adapted model $f$, where $\bigcup_{i=1}^{\bar{L}} P_i = [L]$. Each element $P_i \in \mathcal{P}$ consists of consecutive integers. Given a partition $\mathcal{P}$, each element $P_i$ specifies that the layers with index $l \in P_i$ in the adapted model will be used to approximate the $i$-th layer in the target model. Example 1, which uses every two layers in the adapted model to approximate each layer in the target model, can be considered as a partition represented as $\{\{1, 2\}, \{3, 4\}\}$. Similarly, we extend this simple uniform partition into general cases for $\bar{L}$-layer target FNN and $L$-layer frozen FNN:

$$\mathcal{P}^{\text{u}} = \left\{ P_1^{\text{u}}, \ldots, P_{\bar{L}}^{\text{u}} \right\} := \left\{ \{1, \ldots, M\}, \{M+1, \ldots, 2M\}, \ldots, \{(\bar{L}-1)M+1, \ldots, L\} \right\},$$

where $M := \lfloor L/\overline{L} \rfloor$. The uniform partition indicates that every $M$ layers in the adapted model are employed to approximate each layer in the target model. We use $\prod_{l \in P_i} \boldsymbol{W}_l$ to denote the product of the weight matrices from the layers $l \in P_i$, with the later layer positioned to the left and the earlier layer to the right in the matrix product. For example, $\prod_{l \in P_1^u} \boldsymbol{W}_l = \prod_{l=1}^{M} \boldsymbol{W}_l = \boldsymbol{W}_M \cdots \boldsymbol{W}_1$.

We first extend Lemma 2 to multi-layer FNN approximation setting using this uniform partition.

**Uniform Model Partition.** Given a specified LoRA-rank $R \in [D]$, to derive our results, we introduce a mild non-singularity assumption on the weight matrices of the target model and frozen model for the feasibility of our analysis. This assumption is mild, supported by Lemma 3 that even weight matrices initialized at random can meet this requirement.

**Assumption 1** (Non-Singularity). *For a fixed LoRA-rank $R \in [D]$, the weight matrices of the frozen model $(\boldsymbol{W}_l)_{l=1}^{L}$ and matrices $\left( \prod_{l \in P_i^u} \boldsymbol{W}_l \right) + \mathrm{LR}_r(\overline{\boldsymbol{W}}_i - \prod_{l \in P_i^u} \boldsymbol{W}_l)$ are non-singular for all $r \leq R(M-1)$ and $i \in [\overline{L}]$.*

**Lemma 3.** *Let $(\overline{\boldsymbol{W}}_l)_{l=1}^{\overline{L}}, (\boldsymbol{W}_l)_{l=1}^{L} \in \mathbb{R}^{D \times D}$ be matrices whose elements are drawn independently from arbitrary continuous distributions. Then, with probability 1, Assumption 1 holds $\forall R \in [D]$.*

Given this assumption, here we present our first main result, which shows that any frozen FNN can be adapted to exactly approximate the target FNN via LoRA.

**Theorem 3.** *Under Assumption 1, if LoRA-rank $R \geq \lceil \max_{i \in [\overline{L}]} \mathrm{rank}(\overline{\boldsymbol{W}}_i - \prod_{l \in P_i^u} \boldsymbol{W}_l)/M \rceil$, then there exists rank-$R$ or lower matrices $\Delta\boldsymbol{W}_1, \dots, \Delta\boldsymbol{W}_L \in \mathbb{R}^{D \times D}$ and bias vectors $\widehat{\boldsymbol{b}}_1, \dots, \widehat{\boldsymbol{b}}_L \in \mathbb{R}^D$ such that the low-rank adapted model $f$ can exactly approximate the target model $\overline{f}$, i.e., $f(\boldsymbol{x}) = \overline{f}(\boldsymbol{x})$, $\forall \boldsymbol{x} \in \mathcal{X}$.*

Moreover, combining Lemma 3 and Theorem 3 gives the following corollary.

**Corollary 4.** *Assume that the elements of $(\overline{\boldsymbol{W}}_l)_{l=1}^{\overline{L}}, (\boldsymbol{W}_l)_{l=1}^{L}$ are independently drawn from arbitrary continuous distributions. When $R \geq D/M$, with probability 1, there exists rank-$R$ or lower matrices $\Delta\boldsymbol{W}_1, \dots, \Delta\boldsymbol{W}_L \in \mathbb{R}^{D \times D}$ and bias vectors $\widehat{\boldsymbol{b}}_1, \dots, \widehat{\boldsymbol{b}}_L \in \mathbb{R}^D$ such that low-rank adapted model $f$ can exactly approximate the target model $\overline{f}$ on $\mathcal{X}$, i.e., $f(\boldsymbol{x}) = \overline{f}(\boldsymbol{x})$, $\forall \boldsymbol{x} \in \mathcal{X}$.*

To understand the implications of this corollary, let us consider $L \gg \overline{L}$. In this scenario, the required LoRA-rank is sufficiently small such that the dimension of the rank-$R$ matrix is approximately $2RD$. This corollary suggests that with $2RDL \geq 2D^2L/M \approx 2D^2\overline{L}$ learnable parameters, even a random FNN can be adapted into the target model $\overline{f}$. It is noteworthy that the total number of parameters of the target model is $D^2\overline{L}$. This indicates that even though the learnable parameters under LoRA finetuning appear to be highly constrained (low-rank constrained learnable parameters distributed across many layers), the effective expressive power of LoRA is nearly optimal up to a constant factor of 2. Our discovery provides the first theoretical insights into the practical success of LoRA. Furthermore, Theorem 3 indicates that if the model $f$ is 'close' to $\overline{f}$ such that $\max_{i \in [\overline{L}]} \mathrm{rank}(\overline{\boldsymbol{W}}_i - \prod_{l \in P_i^u} \boldsymbol{W}_l)$ is small, the number of learnable parameters used by LoRA can be lower than $D^2\overline{L}$.

Meanwhile, when the employed LoRA-rank is lower than the critical threshold, the following theorem provides an upper bound for the approximation error.

**Theorem 5.** *Define the approximation error of $i$-th layer as $E_i = \sigma_{RM+1}(\overline{\boldsymbol{W}}_i - \prod_{l \in P_i^u} \boldsymbol{W}_l)$, and the magnitude of the parameters and the input as $\beta := \max_{i \in [\overline{L}]} \left( \sqrt{\|\Sigma\|_F} \prod_{j=1}^{i} \|\overline{\boldsymbol{W}}_j\|_F + \sum_{j=1}^{i} \prod_{k=j+1}^{i-1} \|\overline{\boldsymbol{W}}_k\|_F \|\overline{\boldsymbol{b}}_j\|_2 \right) \vee \sqrt{\|\Sigma\|_F}$.*

*Under Assumption 1, there exists rank-$R$ or lower matrices $(\Delta\boldsymbol{W}_l)_{l=1}^{L}$ with $\Delta\boldsymbol{W}_l \in \mathbb{R}^{D \times D}$ and bias vectors $(\widehat{\boldsymbol{b}}_l)_{l=1}^{L}$ with $\widehat{\boldsymbol{b}}_l \in \mathbb{R}^D$ such that for input $\mathbf{x} \in \mathcal{X}$ with $\mathbb{E}\mathbf{x}\mathbf{x}^\top = \Sigma$,*

$$\mathbb{E}\left\| f(\mathbf{x}) - \overline{f}(\mathbf{x}) \right\|_2 \leq \beta \sum_{i=1}^{\overline{L}} \max_{k \in [\overline{L}]} \left( \|\overline{\boldsymbol{W}}_k\|_F + E_k \right)^{\overline{L}-i} E_i.$$

Theorem 5 provides an upper bound on the approximation error for the adapted model. This bound is influenced by several factors: (i) magnitude of the target model's parameters and the input, which

is captured by $\beta$ and $\left\|\overline{\boldsymbol{W}}_k\right\|_{\mathrm{F}}$, (ii) the rank of the adapter $R$ and the discrepancy between the frozen model and the target model $(\overline{\boldsymbol{W}}_i - \prod_{l \in P_i^{\mathrm{u}}} \boldsymbol{W}_l)_{i=1}^{\overline{L}}$, both of which contribute to the term $E_i$, (iii) the depth of the frozen model $L$, reflected in $M$ and consequenly $E_i$.

All the proofs of the results derived for uniform partition are provided in Sec. E.2.

**General Model Partition.** We note that employing this uniform partition strategy for approximating the target model may not always yield optimal results. To illustrate this, we revisit the case considered by Example 1, where $\overline{L} = 2$ and $L = 4$. Consider a scenario where the first layer of the frozen model has been pretrained to match the first layer of the target model. In this case, we can use just the first layer in $f$ to approximate the first layer in $\overline{f}$, and a zero LoRA-rank is sufficient for the exact representation of the first layer. The remaining three layers in $f$ can then be used to approximate the second layer in $\overline{f}$. Compared to uniform partition, this partition leverages more layers to approximate the second layer in $\overline{f}$, allowing us to achieve the desired performance with a lower LoRA-rank, as per Lemma 2. This suggests that our approximation error bounds could be further optimized by considering partitioning schemes tailored to specific scenarios.

We now extend our results to a more general setting, where we do not assume a uniform partition. Concurrently, recent research by Zhang et al. (2023) has shown that the application of varying LoRA-ranks leads to improved results. Consequently, we permit each layer in the frozen model to utilize adapters with different LoRA-ranks. The rank of the LoRA adapter associated with the $l$-th layer in the frozen model is denoted by $R_l$, where $l \in [L]$. This result relies on Assumption 2, an analog of Assumption 1, but revised to include a general model partition. More details, including the proofs, are provided in Sec. E.3.

**Theorem 6.** *Consider a partition $\mathcal{P}$ for the frozen model. Let Assumption 2 hold. If $\sum_{l \in P_i} R_l \geq \mathrm{rank}(\overline{\boldsymbol{W}}_i - \prod_{l \in P_i} \boldsymbol{W}_l)$ for all $i \in [\overline{L}]$, there exists LoRA adapters $(\Delta \boldsymbol{W}_l)_{l=1}^L$ with $\mathrm{rank}(\Delta \boldsymbol{W}_l) \leq R_l$ and biases $(\widehat{\boldsymbol{b}}_l)_{l=1}^L$ such that the adapted model $f$ can exactly approximate the target model.*

*Moreover, define the approximation error of the $i$-th layer as $E_i = \sigma_{\sum_{l \in P_i} R_l + 1}(\overline{\boldsymbol{W}}_i - \prod_{l \in P_i} \boldsymbol{W}_l)$, and the magnitude of the parameters and the input as $\beta := \max_{i \in [\overline{L}]} \left( \sqrt{\|\Sigma\|_F} \prod_{j=1}^i \left\|\overline{\boldsymbol{W}}_j\right\|_F + \sum_{j=1}^i \prod_{k=j+1}^{i-1} \left\|\overline{\boldsymbol{W}}_k\right\|_F \left\|\overline{\boldsymbol{b}}_j\right\|_2 \right) \vee \sqrt{\|\Sigma\|_F}$.*

*Then, there exists LoRA adapters $(\Delta \boldsymbol{W}_l)_{l=1}^L$ with $\mathrm{rank}(\Delta \boldsymbol{W}_l) \leq R_l$ and biases $(\widehat{\boldsymbol{b}}_l)_{l=1}^L$ such that for any input $\mathbf{x} \in \mathcal{X}$ with $\mathbb{E}\mathbf{x}\mathbf{x}^\top = \Sigma$, the approximation error can be bounded as*

$$\mathbb{E}\left\|f(\mathbf{x}) - \overline{f}(\mathbf{x})\right\|_2 \leq \beta \sum_{i=1}^{\overline{L}} \max_{k \in [\overline{L}]} \left(\left\|\overline{\boldsymbol{W}}_k\right\|_F + E_k\right)^{\overline{L}-i} E_i.$$

**Comparison to Tuning Final Layers.** Updating the final layers and keeping the initial layers frozen (Chatfield et al., 2014; Donahue et al., 2014; Sharif Razavian et al., 2014; Rahimi & Recht, 2007) is another popular model adaptation method. However, unlike LoRA, which can adapt even randomly generated networks to match a target model, empirical studies (Kornblith et al., 2019) suggest that the effectiveness of final layers tuning heavily depends on the quality of the initial layers. This indicates that merely tuning the final layers of randomly generated networks may not yield desirable performance. The following lemma rigorously supports this assertion, demonstrating that regardless of how the final layers are tuned, it is impossible to adapt a randomly generated model into even a one-layer FNN, a model of very low complexity.

**Lemma 4.** *Let $D \geq 2$ and $\overline{f}$ be a one-layer target FNN. Assume that the elements of weight matrices $(\boldsymbol{W}_l)_{l=1}^L$ are independently drawn from arbitrary continuous distributions. With probability 1, for any tuning of the last $L - 1$ layers, $f \neq \overline{f}$.*

In Corollary 4, we demonstrate that LoRA can adapt any randomly generated models to match the target model, using at most twice the number of learnable parameters as the target model. However, this lemma reveals that final layers tuning, even with $L - 1$ times the learnable parameters of the target model, cannot achieve performance comparable to LoRA. In other words, LoRA requires at most $2RDL \leq 2D^2$ learnable parameters to achieve an exact approximation, while final layers

tuning fails to approximate the target model even with $(L-1)D^2$ learnable parameters. Therefore, when $L \geq 3$, LoRA can deliver strictly superior performance than final layers tuning with the same or fewer parameters. This provides insights into the empirical observation that LoRA outperforms final layers tuning (Kaplun et al., 2023; Ding et al., 2023).

## 4 EXPRESSIVE POWER OF TRANSFORMER NETWORKS WITH LORA

### 4.1 PROBLEM SETTING

Transformer network, denoted as $\texttt{TFN}_{L,D}$, is a composition of $L$ Transformer blocks and an output layer, parameterized by weight $\boldsymbol{W}_o \in \mathbb{R}^{D \times D}$. Each transformer block comprises a $H$-head self-attention layer, parameterized by weight $((\boldsymbol{W}_{Ol}^h, \boldsymbol{W}_{Vl}^h, \boldsymbol{W}_{Kl}^h, \boldsymbol{W}_{Ql}^h)_{h=1}^H)_{l=1}^L$, followed by a token-wise feedforward layer, parameterized by weight $(\boldsymbol{W}_{1l}, \boldsymbol{W}_{2l})_{l=1}^L$ and bias $(\boldsymbol{b}_{1l}, \boldsymbol{b}_{2l})_{l=1}^L$. We assume that all weight matrices have a dimension of $D \times D$, while the bias vectors are of dimension $D$.

We employ the same formulations of transformer blocks as Yun et al. (2020a), with one exception: we exclude skip connections for analytical feasibility. As before, we use $\bar{\cdot}$ (e.g., $\overline{\boldsymbol{W}}_{1l}$) to represent the corresponding parameters for the target model, and $\Delta \cdot$ (e.g., $\Delta \boldsymbol{W}_{Ol}^h$) to represent the corresponding low-rank update. For TFN cases, we consider scenarios where both the frozen model and the target model have $L$ Transformer blocks. For an explicit formulation, please refer to Sec. F.2.

### 4.2 MAIN RESULTS ON TRANSFORMER NETWORKS

We now present our main findings on TFNs. The first result relies on a non-singularity assumption (Assumption 4) tailored for TFN. This assumption is mild, and models with randomly generated weights can satisfy its criteria (Lemma 14). Further details are deferred to Sec. F.2.

The following theorem shows that adding LoRA adapters primarily to the self-attention layers enables the adapted model $f$ to exactly approximate the target model $\bar{f}$. This finding is consistent with a recent observation made by Hu et al. (2022a), which indicates that a good performance can be achieved by adapting only the attention layers when applying LoRA to TFNs.

**Theorem 7.** *Consider a given LoRA-rank $R \in [D]$. Let Assumption 4 hold. Let $G_i$ be the rank-based functionality gap to $i$-th transformer block ($i \in [L]$) or output layer ($i = L+1$) defined in (23). If $R \geq \max_{i \in [L+1]} \lceil \frac{G_i}{2} \rceil$, then there exists low-rank adapters with rank lower than $R \in [D]$ $((\Delta \boldsymbol{W}_{Kl}^h, \Delta \boldsymbol{W}_{Ql}^h, \Delta \boldsymbol{W}_{Vl}^h, \Delta \boldsymbol{W}_{Ol}^h)_{h=1}^H)_{l=1}^L, \Delta \boldsymbol{W}_{2L}, \Delta \boldsymbol{W}_o$ with other low-rank adapters set to $\boldsymbol{O}$, and updated bias vectors $(\widehat{\boldsymbol{b}}_{1l}, \widehat{\boldsymbol{b}}_{2l})_{l=1}^L$, such that for any $\boldsymbol{X} \in \mathbb{R}^{D \times N}$, the adapted model $f$ exactly approximates target model $\bar{f}$, i.e., $f(\boldsymbol{X}) = \bar{f}(\boldsymbol{X})$.*

*Proof Sketch.* The primary challenge for extending our analysis to TFNs, similar to FNN cases, is the nonlinearity introduced by softmax and ReLU. To manage this, we segment a sequence of transformer blocks based on the softmax and ReLU functions. Specifically, we align the output of attention scores before the softmax is applied, and then match the output of the first feedforward layer before ReLU is applied. □

The complete proof of Theorem 7 and results for randomly generated models can be found in Sec. F.2. Meanwhile, our results here are specifically for TFNs with multi-head attention layers. For TFNs with single-head attention layers, the construction of LoRA adapters differs due to the absence of $\boldsymbol{W}_{Oi}^h$. Since the results are similar, we defer the problem setting and results for TFNs with single-head attention layers to Sec. F.1.

## 5 EXPERIMENTS

Recall that all our theoretical statements are based on our construction of the LoRA adapters presented in their corresponding proofs. To validate these results, here we empirically examine the relationship between approximation error and rank by integrating the LoRA adapters, which are constructed with the uniform partition in our proof, into the frozen model.

**Validation of Our LoRA Adapter Construction.** We employ the *Mean Squared Error* (MSE) to assess the approximation error, comparing the MSE of the LoRA adapter as derived from the gradient update method with that from our construction. We consider linear models and FNNs with model dimension $D = 16$. For linear model cases, we set $\bar{L} = 1, L = 2$, while for FNN cases, we set $\bar{L} = 2, L = 4$. We include two variants of the frozen model for fine-tuning: one with randomly initialized parameters (Random) and another pretrained on the target distribution (Pretrained).

Our results for linear model approximation and FNN approximation via LoRA are depicted in Fig. 1a and 1b, respectively. Firstly, we observe that the MSE of both two cases is close to zero when $R \geq \frac{D}{L/\bar{L}} = 8$, which corroborates our claims. Meanwhile, a comparison between the left and right columns of Fig. 1a suggests that pretraining can further reduce the required rank to achieve near-zero approximation error. Furthermore, the curves of our construction align well with those of the gradient update method in linear model approximation cases, confirming the optimality claimed in Lemma 1. However, for FNN approximation cases, the gradient update method outperforms our construction in the small rank region. We conjecture that the suboptimality of our construction for this multi-layer FNN case could arise from unnecessarily matching the intermediate outputs of the frozen model with those of the target model during adapter construction. Additionally, the uniform partition could also be one contributing factor.

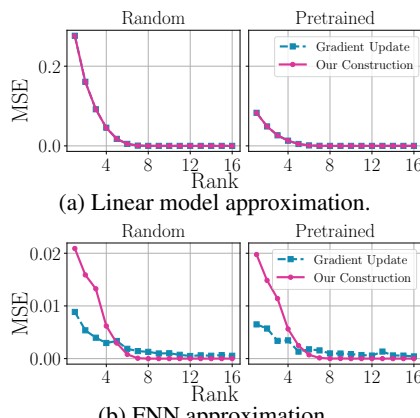

(a) Linear model approximation.

(b) FNN approximation.

Figure 1: Approximation error (measured by MSE) versus LoRA-rank.

| Findings | Empirical Observation | Theoretical Insights |
|---|---|---|
| For a fixed downstream task, larger models require a lower LoRA-rank to achieve the desired performance. | Sec. G.9 | Lemma 1, 2, and Theorem 5, 6 |
| When the frozen model is closer to the target model, a lower LoRA-rank is sufficient to attain the desired performance. | Sec. G.9 and 6-th footnote in Hu et al. (2022a) | Lemma 1, 2, and Theorem 5, 6, 7 |
| LoRA outperforms final layers tuning if the quality of shared representation is not good. | Sec. G.4 and observations by Kaplun et al. (2023) and Ding et al. (2023) | Lemma 4 |
| In addition to applying low-rank updates to weight matrices, it is crucial to also update the bias. | Sec. G.5 and 2-nd footnote in Hu et al. (2022a) | Proofs in Sec. 3.2 and E.1 |
| Tuning attention weights is sufficient for achieving good performance on TFNs. | Sec. 4.2 in Hu et al. (2022a) | Theorem 7 |
| Current optimization algorithms for LoRA training might be suboptimal. | Fig. 4, 5, and 9 | — |

Table 1: Summary of our findings, supported by empirical evidence and theoretical results.

**Detailed Experimental Setup and Additional Experiments in the Appendix.** Further experiment details and a series of additional experiments, including simulations on FNNs and TFNs at different depths, evaluation of classification tasks, empirical comparison between LoRA and the final layers tuning, investigation of the importance of updatable bias, LoRA's generalization and optimization properties, and experiments on GLUE benchmark (Wang et al., 2018), are provided in Sec. G. Table 1 summarizes all the empirical findings and aligns them with theoretical insights.

## 6 CONCLUSIONS

This work pioneers the theoretical analysis of LoRA fine-tuning's expressive capabilities in FNNs and TFNs, offering novel insights into how rank, model depth, and proximity to the target model influence LoRA's effectiveness. Our theoretical findings are validated by empirical evidence. Future work includes quantifying approximation errors for TFNs when the LoRA-ranks are lower than required and refining LoRA adapter update algorithms based on our construction of LoRA adapters.

## ACKNOWLEDGEMENT

This work was supported by NSF Award DMS-2023239, NSF/Intel Partnership on Machine Learning for Wireless Networking Program under Grant No. CNS-2003129, and a grant by FuriosaAI. We extend our heartfelt gratitude to Angeliki Giannou, Kartik Sreenivasan, Tuan Dinh, Jy-yong Sohn, Jingpeng Liu, and anonymous reviewers for their insightful comments that significantly enhanced the quality of our paper.

## REPRODUCIBILITY STATEMENT

The code for all experiments reported in this paper is publicly accessible. For the purpose of reproducibility, the code can be found at the following anonymized GitHub repository: https://github.com/UW-Madison-Lee-Lab/Expressive_Power_of_LoRA.

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

# Appendix

This appendix encompasses more discussions, experiments, and proofs of the results presented in the main body. Given the extensive use of notations in our paper, we begin by presenting a list of common notations in Sec. A for the reader's convenience. We then delve into a more detailed discussion of related works in Sec. B. Following this, we present the proofs of results from the main body and auxiliary results in Sec. C, D, E, and F. Specifically, we provide additional results for TFN with single-head attention layers, and TFN with multi-head attention layers under random model cases in Sec. F. Further experimental details and interesting experiment findings are provided in Sec. G. Finally, we discuss how to extend our results to cases with varying model dimensions in Sec. H, while this work primarily focuses on instances where both the target model and the frozen model possess the same model width $D$. More potential future works are outlined in Sec. I.

# A    LIST OF COMMON NOTATIONS

We first give a list of common notations that are used in the main body and appendix for reference.

- $f$: LoRA-adapted model.

- $\bar{f}$: target model.

- $f_0$: frozen/pretrained model.

- $R$: rank of LoRA adapters.

- $D$: dimensionality of the model, representing the number of neurons in each layer for FNNs and the embedding size for TFNs.

- $L$: depth of the (frozen) model, representing the number of layers for FNNs and the number of transformer blocks for TFNs.

- $N$: sequence length of the input for TFNs.

- $\boldsymbol{x}$: input.

- $\mathbf{x}$: random input.

- $\boldsymbol{X}$: matrix input.

- $\mathcal{X}$: input space.

- $\Sigma$: $\mathbb{E}\mathbf{x}\mathbf{x}^\top$.

- $\boldsymbol{W}$: a weight matrix associated with (frozen) model. Subscripts and superscripts may be added for specificity.

- $\boldsymbol{b}$: a bias vector associated with the (frozen) model. Subscripts may be added for specificity.

- $\boldsymbol{z}_l$: the output of the first $l$ layers in the (frozen) FNN.

- $\boldsymbol{Z}_l$: the output of the first $l$ transformer blocks in a (frozen) TFN.

- $\overline{\boldsymbol{W}}$: a weight matrix associated with the target model. Subscripts and superscripts may be added for specificity.

- $\bar{\boldsymbol{b}}$: a bias vector associated with the target model. Subscripts may be added for specificity.

- $\bar{\mathbf{z}}_l$: the intermediate output of the first $l$ layers in target FNN given the random input $\mathbf{x}$.

- $\overline{\boldsymbol{Z}}_l$: the output of the first $l$ transformer blocks in a target TFN.

- $\bar{L}$: depth of the target model, representing the number of layers for FNNs and the number of transformer blocks for TFNs.

- $\Delta\boldsymbol{W}$: the weight matrix of a LoRA adapter.

- $\widehat{\boldsymbol{b}}$: a bias vector associated with the LoRA-adapted model.

- $\widehat{\mathbf{z}}_l$: the output of the first $l$ layers in the LoRA-adapted model given the random input $\mathbf{x}$.

- $\widehat{\boldsymbol{Z}}_l$: the output of the first $l$ transformer blocks in the LoRA-adapted model.

- $M$: the ratio of the depth of the frozen model to that of the target model, i.e., $L/\bar{L}$.

- $\mathcal{P}$: partition $\mathcal{P} = \{P_1, \ldots, P_{\bar{L}}\}$, each element $P_i$ specifies that the layers with index $l \in P_i$ in the adapted model will be used to approximate the $i$-th layer in the target model.

- $P_i$: the $i$-th element in partition $\mathcal{P}$.

- $\mathcal{P}^{\mathrm{u}}$: uniform partition $\mathcal{P}^{\mathrm{u}} := \{\{1, \ldots, M\}, \{M+1, \ldots, 2M\}, \ldots, \{(\bar{L}-1)M+1, \ldots, L\}\}$. The uniform partition indicates that every $M$ layers in the adapted model are employed to approximate each layer in the target model.

- $P_i^{\mathrm{u}}$: the $i$-th element in uniform partition $\mathcal{P}^{\mathrm{u}}$.

- $\boldsymbol{I}_D$: the $D \times D$ identity matrix. When the context permits, the subscript $D$ of $\boldsymbol{I}_D$ may be omitted, simplifying the notation to $\boldsymbol{I}$.

- $\boldsymbol{I}_{a:b,D}$: a diagonal matrix where the diagonal entries from the $a$th to $b$th position are set to 1, while all remaining entries are 0s.

- $\sigma_d(\cdot)$: the $d$-th largest singular value for the given square matrix. When $d$ is greater than the width of the matrix, $\sigma_d(\cdot) = 0$.

- $\mathrm{LR}_r(\cdot)$: best rank-$r$ approximation of a square matrix in Frobenuis norm and spectral norm. The subscript $r$ may be omitted to indicate a general low-rank approximation without specifying the rank.

- $\prod_{l \in P_i} \boldsymbol{W}_l$: product of the weight matrices from the layers $l \in P_i$, with the later layer positioned to the left and the earlier layer to the right in the matrix product. For example, $\prod_{l \in P_1^u} \boldsymbol{W}_l = \prod_{l=1}^{M} \boldsymbol{W}_l = \boldsymbol{W}_M \cdots \boldsymbol{W}_1$.

## B  EXPANDED RELATED WORKS

**Expressive Power of Fully Connected Neural Networks**   The theoretical exploration of the expressive power of unfrozen fully connected neural networks has advanced since the introduction of the first universal approximation theorem (Hornik et al., 1989; Cybenko, 1989). Subsequent studies have demonstrated the benefits of depth, asserting that sufficient depth can ensure function approximation (Bengio & Delalleau, 2011; Eldan & Shamir, 2016; Liang & Srikant, 2017; Telgarsky, 2016; 2015). There are also works that have examined the expressive power of FNN from a view of width (Lu et al., 2017; Park et al., 2021; Bietti & Bach, 2021) and the number of neurons (Shen & Zhang, 2020). While these results assume that weight matrices can be arbitrarily adjusted for optimal performance, Hsu et al. (2021) examined the expressive power of randomly generated two-layer FNNs. Our work shares similarities with this direction, as we also delve into scenarios with randomly generated models. Beyond characterizing expressive power by approximation error, alternative metrics have been proposed. Metrics such as Vapnik-Chervonenkis (Vapnik & Chervonenkis, 2015; Seo et al., 2021) and Rademacher complexities (Bartlett & Mendelson, 2001) are utilized to assess classification capacity. Furthermore, Raghu et al. (2017) introduced a novel metric that captures the structural properties of an FNN, and Lee et al. (2017) investigated the ability of FNNs to express distributions.

**Expressive Power of Transformers**   As TFNs have grown increasingly popular, a few studies have been conducted to investigate their expressive power. Yun et al. (2020a) established the universal approximation theorem for TFNs in approximating sequence-to-sequence functions. Likhosherstov et al. (2021) characterized the self-attention layer as a matrix and demonstrated that this matrix can approximate any sparse matrices. Beyond approximation, further research has delved into other facets of TFNs' expressive power. For instance, Giannou et al. (2023b) found that looped transformers can emulate an instruction-set computer, while Pérez et al. (2019) demonstrated that TFNs attain Turing completeness when operating with infinite precision.

However, all these theories above *cannot* fully explain the performance of frozen neural networks as they generally cannot factor in pre-trained model parameters and adaptation methods.

**Expressive Power of Adaptation Methods**   Our work focuses on investigating the expressive power of adaptation methods. In stark contrast to the flourishing research on the expressive power of neural networks, there exists a limited number of works investigating the expressive power of adaptation methods. A notable exception is Giannou et al. (2023a), investigating the expressive power of normalization parameter fine-tuning. They demonstrate that fine-tuning the normalization layers alone can adapt a randomly initialized ReLU network to match any target network that is $O(\text{width})$ times smaller. We borrow some proof techniques from this work, including techniques for extending results from linear neural networks to ReLU neural networks. In another recent work (Englert & Lazic, 2022), the authors show that *neural reprogramming* (Elsayed et al., 2019; Engel et al., 2018; Lee et al., 2020; Dinh et al., 2022a; Chen, 2022), a technique that modifies only the inputs while keeping the pretrained network frozen, can adapt any random two-layer ReLU network to achieve arbitrarily high accuracy on a Bernoulli data model over hypercube vertices. Oymak et al. (2023) explores prompt-tuning within a one-layer attention architecture, revealing that the model resulting from prompt tuning (Lester et al., 2021) is more expressive than the naive self-attention model. Petrov et al. (2023) shows that prompt-tuning and prefix tuning (Li & Liang, 2021) are strictly less expressive than full fine-tuning. Despite these early attempts, no existing study has yet explored the expressive power of LoRA, the current leading adaptation method.

**Other Theoretical Analysis of Adaptation Methods**  Lots of efforts have been taken to theoretically analyze other properties of adaptation methods such as generalization. Maurer et al. (2016) provides the generalization bounds for transfer learning, particularly for final layers tuning, demonstrating that the estimation error reduces as the pretrained task diversity and the number of samples for the target task increase. Tripuraneni et al. (2020) further refines this bound by studying the effect of the number of samples in the pre-trained tasks. Interestingly, the estimation error of the final layers tuning provided in Tripuraneni et al. (2020) heavily depends on the quality of the shared representation. This insight aligns with our finding on final layers tuning (Lemma 4), which implies that tuning the final layers fails to adapt an $L$-layer randomly generated FNN to approximate any one-layer target FNN if the first layer remains frozen. This failure is attributed to the poor quality of the shared random representation. Du et al. (2021) further investigates final layers tuning in few-shot cases, i.e., when there are only a few samples for the target task. A recent study by Malladi et al. (2023), which examined LoRA and full fine-tuning through the lens of the Neural Tangent Kernel (Jacot et al., 2018), suggested that if the kernel view describes full fine-tuning, then LoRA approximates full fine-tuning. However, their theoretical analysis of LoRA is based on linear models, thus limiting its applicability. In contrast, our study considers a more general setting.

With the rapid advancement of large language models, new adaptation methods such as in-context learning (Brown et al., 2020), prefix tuning, and prompt-tuning (Lester et al., 2021) are gaining increasing attention. A particular focus of research is the exploration of the theoretical underpinnings of in-context learning (Akyürek et al., 2023; Bai et al., 2023; Wies et al., 2023; Xie et al., 2022; von Oswald et al., 2022; Ahn et al., 2023). Akyürek et al. (2023) demonstrates that transformer-based in-context learners implicitly implement standard learning algorithms, while Bai et al. (2023) presents a similar finding and posits that in-context learning performs algorithm selection like a statistician. Wies et al. (2023) delves into the analysis of the sample complexity of in-context learning. Other works find that in-context learning is equivalent to gradient descent (von Oswald et al., 2022; Ahn et al., 2023), and Bayesian inference (Xie et al., 2022). Beyond in-context learning, a recent research by Tutunov et al. (2023) developed a theoretical framework elucidating how LLMs can accurately generate chain-of-thought reasoning (Wei et al., 2022).

## C  PROOFS RELATED TO LINEAR ALGEBRA

In this section, we present a collection of commonly used matrix inequalities and the basic properties of randomly generated matrices.

### C.1  COMMON MATRIX INEQUALITIES

Here, we present some commonly used basic properties for matrix multiplication including rank computation, norm inequalities, as well as key results involving the trace and Frobenius norm of matrices for reference:

$$
\begin{aligned}
&\operatorname{rank}(\boldsymbol{AB}) \leq \operatorname{rank}(\boldsymbol{A}) \wedge \operatorname{rank}(\boldsymbol{B}); \\
&\|\boldsymbol{Ax}\|_2 \leq \|\boldsymbol{A}\|_2 \|\boldsymbol{x}\|_2; \\
&\mathbb{E}\boldsymbol{x}^\top \boldsymbol{Ax} = \operatorname{tr}(\boldsymbol{A}\operatorname{Cov}(\mathbf{x})) + (\mathbb{E}\mathbf{x})^\top \boldsymbol{A}(\mathbb{E}\mathbf{x}) = \operatorname{tr}(\boldsymbol{A}\mathbb{E}\mathbf{xx}^\top); \\
&\operatorname{tr}(\boldsymbol{AB}) = \operatorname{tr}(\boldsymbol{BA}); \\
&\operatorname{tr}(\boldsymbol{AB}) \leq \operatorname{tr}(\boldsymbol{A})\operatorname{tr}(\boldsymbol{B}); \\
&\|\boldsymbol{A}\|_\mathrm{F} = \sqrt{\operatorname{tr}(\boldsymbol{AA}^\top)}; \\
&\|\boldsymbol{A}\|_\mathrm{F} = \operatorname{tr}(\boldsymbol{A}) \text{ for symmetric } \boldsymbol{A}; \\
&\|\boldsymbol{A}\|_\mathrm{F} = \sqrt{\sum_i \sigma_i^2(\boldsymbol{A})}.
\end{aligned}
\tag{1}
$$

### C.2  NON-SINGULARITY OF RANDOMLY GENERATED MATRICES

Although the non-singularity of randomly generated matrices is already established, we include a proof for completeness.

To facilitate the proof, we introduce a lemma which states that if a polynomial is non-zero, then the set of roots corresponding to a zero value of the polynomial has a Lebesgue measure of zero.

**Lemma 5** (Caron & Traynor (2005)). *Let $p(\boldsymbol{x})$ be a polynomial of degree $d$, $\boldsymbol{x} \in \mathbb{R}^n$. If $p$ is not the zero polynomial, then the set $\mathcal{S} := \{\boldsymbol{x} \in \mathbb{R}^n \mid p(\boldsymbol{x}) = 0\}$ is of Lebesgue measure zero.*

We note that the determinant of a matrix can be viewed as a polynomial function of its vectorized version. Based on this insight, we proceed with our proof.

**Lemma 6.** *Let $\mathbf{X} \in \mathbb{R}^{D \times D}$ be a random matrix that follows arbitrary continuous distribution with support having non-zero Lebesgue measure on $\mathbb{R}^{D \times D}$. Then, $\mathbf{X}$ is non-singular with probability 1.*

*Proof of Lemma 6.* The result is a direct consequence of Lemma 5. Let $\mathbf{x} = \text{vec}(\mathbf{X})$. Then, $\mathbf{x}$ is a random vector following arbitrary continuous distribution with a support having non-zero Lebesgue measure on $\mathbb{R}^{D \times D}$.

First, we establish the relationship:

$$\mathbb{P}\left(\det(\mathbf{X}) = 0\right) = \mathbb{P}\left(p(\mathbf{x}) = \mathbf{0}\right)$$

for some polynomial function $p$. We denote the support of random vector $\mathbf{x}$ by $\mathcal{X} \subset \mathbb{R}^{D^2}$, and the probability density function (PDF) of $\mathbf{x}$ by $q$. Then,

$$\mathbb{P}\left(p(\mathbf{x}) = \mathbf{0}\right) = \int_{\mathcal{X}} \mathbf{1}\left\{p(\boldsymbol{x}) = \mathbf{0}\right\} q(\boldsymbol{x}) \mathrm{d}\boldsymbol{x} = \int_{\mathcal{X} \cap \{\boldsymbol{x}: p(\boldsymbol{x}) = \mathbf{0}\}} q(\boldsymbol{x}) \mathrm{d}\boldsymbol{x}.$$

By Lemma 5, the Lebesgue measure of $\{\boldsymbol{x} : p(\boldsymbol{x}) = \mathbf{0}\}$ is zero. Hence,

$$\int_{\mathcal{X} \cap \{\boldsymbol{x}: p(\boldsymbol{x}) = \mathbf{0}\}} q(\boldsymbol{x}) \mathrm{d}\boldsymbol{x} = 0.$$

By combining all the equations above, we conclude that $\mathbb{P}(\det(\mathbf{X}) = 0) = 0$, which implies $\mathbf{X}$ is non-singular with probability 1. $\qquad\square$

## D  PROOFS FOR LINEAR MODEL APPROXIMATION

In this section, we present the results and corresponding proofs for the linear model approximation problem introduced in Sec. 2. The *deep linear model* is a common technique in theoretical deep learning research, which offers valuable insights into deep nonlinear models, and has been employed in many notable studies, including those by Saxe et al. (2014); Kawaguchi (2016); Lu & Kawaguchi (2017); Hardt & Ma (2017) and Laurent & von Brecht (2018). We employ this toy model as a preliminary model, which serves as a foundation for extending our results to nonlinear models (i.e., FNN and TFN).

We first provide a slightly more detailed version of Lemma 1 along with its proof. Then, we present a variant of it that allows for different LoRA-ranks for each low-rank adapter. The proof for this variant involves only a minor modification of the proof for Lemma 7.

**Lemma 7.** *[Detailed version of Lemma 1] Define error matrix $\boldsymbol{E} := \overline{\boldsymbol{W}} - \prod_{l=1}^{L} \boldsymbol{W}_l$, and denote its rank by $R_{\boldsymbol{E}} = \text{rank}(\boldsymbol{E})$. For a given LoRA-rank $R \in [D]$, assume that all the weight matrices of the frozen model $(\boldsymbol{W}_l)_{l=1}^{L}$, and $\prod_{l=1}^{L} \boldsymbol{W}_l + \text{LR}_r(\boldsymbol{E})$ are non-singular for all $r \leq R(L-1)$. Then, the approximation error*

$$\min_{\Delta \boldsymbol{W}_l : \text{rank}(\Delta \boldsymbol{W}_l) \leq R} \left\| \prod_{l=1}^{L} (\boldsymbol{W}_l + \Delta \boldsymbol{W}_l) - \overline{\boldsymbol{W}} \right\|_2 = \sigma_{RL+1} \underbrace{\left( \overline{\boldsymbol{W}} - \prod_{l=1}^{L} \boldsymbol{W}_l \right)}_{\textit{Error matrix } \boldsymbol{E}},$$

*and the optimal solution to the matrix approximation problem satisfies $\prod_{l=1}^{L} (\boldsymbol{W}_l + \Delta \boldsymbol{W}_l) = \prod_{l=1}^{L} \boldsymbol{W}_l + \text{LR}_{RL \wedge R_{\boldsymbol{E}}}(\boldsymbol{E})$. Therefore, when $R \geq \lceil \frac{R_{\boldsymbol{E}}}{L} \rceil$, we have $\prod_{l=1}^{L} (\boldsymbol{W}_l + \Delta \boldsymbol{W}_l) = \overline{\boldsymbol{W}}$, implying $f \equiv \bar{f}$.*

*Proof of Lemma* 7. Our goal is to find matrices $\Delta \boldsymbol{W}_1, \ldots, \Delta \boldsymbol{W}_L$ of rank $R$ or lower such that the product of the adapted matrices approximates the target matrix well, i.e., we aim to solve the following constrained optimization problem:

$$\min_{\Delta \boldsymbol{W}_l : \operatorname{rank}(\Delta \boldsymbol{W}_l) \leq R} \left\| \prod_{l=1}^{L} (\boldsymbol{W}_l + \Delta \boldsymbol{W}_l) - \overline{\boldsymbol{W}} \right\|_2 .$$

By subtracting $\prod_{l=1}^{L} \boldsymbol{W}_l$ from both terms, the constrain optimization problem becomes

$$\min_{\Delta \boldsymbol{W}_l : \operatorname{rank}(\Delta \boldsymbol{W}_l) \leq R} \left\| \underbrace{\left( \prod_{l=1}^{L} (\boldsymbol{W}_l + \Delta \boldsymbol{W}_l) - \prod_{l=1}^{L} \boldsymbol{W}_l \right)}_{:=\boldsymbol{A}} - \underbrace{\left( \overline{\boldsymbol{W}} - \prod_{l=1}^{L} \boldsymbol{W}_l \right)}_{:=\boldsymbol{E}} \right\|_2 . \tag{2}$$

To perform analysis on (2), we start with the analysis of $\boldsymbol{A}$ as follows:

$$\begin{aligned} \boldsymbol{A} &= \prod_{l=1}^{L} (\Delta \boldsymbol{W}_l + \boldsymbol{W}_l) - \prod_{l=1}^{L} \boldsymbol{W}_l \\ &= \Delta \boldsymbol{W}_L \prod_{l=1}^{L-1} (\Delta \boldsymbol{W}_l + \boldsymbol{W}_l) + \boldsymbol{W}_L \prod_{l=1}^{L-1} (\Delta \boldsymbol{W}_l + \boldsymbol{W}_l) - \prod_{l=1}^{L} \boldsymbol{W}_l. \end{aligned}$$

Here, we have separated the first term in the product $\prod_{l=1}^{L} (\Delta \boldsymbol{W}_l + \boldsymbol{W}_l)$, breaking it into two parts: one involving $\Delta \boldsymbol{W}_L$ and the other $\boldsymbol{W}_L$. We can further expand the part involving $\boldsymbol{W}_L$:

$$\begin{aligned} \boldsymbol{A} =& \Delta \boldsymbol{W}_L \prod_{l=1}^{L-1} (\Delta \boldsymbol{W}_l + \boldsymbol{W}_l) \\ &+ \boldsymbol{W}_L \left( \Delta \boldsymbol{W}_{L-1} \prod_{l=1}^{L-2} (\Delta \boldsymbol{W}_l + \boldsymbol{W}_l) + \boldsymbol{W}_{L-1} \prod_{l=1}^{L-2} (\Delta \boldsymbol{W}_l + \boldsymbol{W}_l) \right) - \prod_{l=1}^{L} \boldsymbol{W}_l. \end{aligned}$$

At this point, it becomes clear that this expression can be iteratively decomposed. Following this pattern, we can express $\boldsymbol{A}$ as:

$$\begin{aligned} \boldsymbol{A} =& \Delta \boldsymbol{W}_L \prod_{l=1}^{L-1} (\Delta \boldsymbol{W}_l + \boldsymbol{W}_l) + \boldsymbol{W}_L \Delta \boldsymbol{W}_{L-1} \prod_{l=1}^{L-2} (\Delta \boldsymbol{W}_l + \boldsymbol{W}_l) \qquad (3) \\ &+ \ldots + (\prod_{l=2}^{L} \boldsymbol{W}_l)(\Delta \boldsymbol{W}_1 + \boldsymbol{W}_1) - \prod_{l=1}^{L} \boldsymbol{W}_l \\ =& \sum_{l=1}^{L} \underbrace{\left[ (\prod_{i=l+1}^{L} \boldsymbol{W}_i) \Delta \boldsymbol{W}_l (\prod_{i=1}^{l-1} (\boldsymbol{W}_i + \Delta \boldsymbol{W}_i)) \right]}_{:=\boldsymbol{A}_l} . \end{aligned}$$

In this final form, $\boldsymbol{A}$ is decomposed as $\boldsymbol{A} = \sum_{l=1}^{L} \boldsymbol{A}_l$. It is important to note that $\operatorname{rank}(\boldsymbol{A}_l) \leq \operatorname{rank}(\Delta \boldsymbol{W}_l) \leq R$. Consequently, $\operatorname{rank}(\boldsymbol{A}) \leq \sum_{l=1}^{L} \operatorname{rank}(\boldsymbol{A}_l) \leq RL$.

Then, the optimization problem (2) can be relaxed into a low-rank approximation problem

$$(2) \geq \min_{\boldsymbol{A} : \operatorname{rank}(\boldsymbol{A}) \leq RL} \| \boldsymbol{A} - \boldsymbol{E} \|_2 , \tag{4}$$

where the optimal solution is $\boldsymbol{A} = \operatorname{LR}_{RL \wedge R_{\boldsymbol{E}}}(\boldsymbol{E}) := \boldsymbol{E}'$. Therefore, if we can identify rank-$R$ or lower matrices $(\Delta \boldsymbol{W}_l)_{l=1}^{L}$ such that

$$\underbrace{\prod_{l=1}^{L} (\boldsymbol{W}_l + \Delta \boldsymbol{W}_l) - \prod_{l=1}^{L} \boldsymbol{W}_l}_{:=\boldsymbol{A}} = \underbrace{\operatorname{LR}_{RL \wedge R_{\boldsymbol{E}}}(\overline{\boldsymbol{W}} - \prod_{l=1}^{L} \boldsymbol{W}_l)}_{:=\boldsymbol{E}'}, \tag{5}$$

then we effectively solve the matrix approximation problem as defined in (2). Moreover, it is straightforward to verify that (5) directly implies all statements in this lemma. Therefore, our remaining proof focuses on proving (5).

Denote $R_{E'} = RL \wedge R_E$. To derive the explicit form of $E'$, we first refer to the SVD of $E$ as

$$E = UDV^\top,$$

where $U$ and $V$ are orthonormal matrices and the first $R_E$ diagonal entries of $D$ are non-zero, with all remaining entries being zero. Based on this, $E'$ is expressed as

$$E' = UDI_{1:RL,D}V^\top.$$

Having already derived the decomposition $A = \sum_{l=1}^L A_l$, we next aim to decompose $E'$ as $E' = \sum_{l=1}^L E'Q_l$, where $Q_1, \ldots, Q_L \in \mathbb{R}^{D \times D}$. The goal now shifts to identifying $\Delta W_l, Q_l$ such that $A_l = E'Q_l$ for each $l \in [L]$. Achieving this would complete the proof of (5).

Therefore, our goal becomes finding $\Delta W_1, \ldots, \Delta W_L$ with $\mathrm{rank}(\Delta W_l) \leq R$ for all $l \in [L]$ such that

$$A_l = (\prod_{i=l+1}^L W_i)\Delta W_l(\prod_{i=1}^{l-1}(W_i + \Delta W_i)) = E'Q_l, \quad \text{for all } l \in [L]. \tag{6}$$

One sufficient condition for achieving (6) is that the decomposed matrices $Q_1, Q_L$ and low-rank adapters $\Delta W_1, \ldots, \Delta W_L$ meet the following conditions:

$$\sum_{l=1}^L E'Q_l = E', \tag{7}$$

$$\Delta W_l = (\prod_{i=l+1}^L W_i)^{-1}E'Q_l(\prod_{i=1}^{l-1}(W_i + \Delta W_i))^{-1}, \text{ for all } l \in [L] \tag{8}$$

$$\mathrm{rank}(\Delta W_l) \leq R, \text{ for all } l \in [L], \tag{9}$$

$$\mathrm{rank}(W_l + \Delta W_l) = D, \text{ for all } l \in [L-1]. \tag{10}$$

Here (7) describes the decomposition of $E'$, (8) provides one simple solution to (6) when (10) holds, and (9) is the rank constraint on the low-rank adapter. In particular, the (10) is used to ensure the invertibility of $\prod_{i=1}^l (W_i + \Delta W_i)$ for $l \in [L-1]$. This condition is not necessary for $l = L$ as the inverse of $W_L + \Delta W_L$ is not required for computing any low-rank adapters.

We will show that the matrices $(Q_l)_{l=1}^L$ defined by

$$Q_l = VI_{(R(l-1)+1) \wedge R_{E'}:Rl \wedge R_{E'},D}V^\top, \quad \text{for all } l \in [L], \tag{11}$$

and $\Delta W_l$ defined by (8) for all $l \in [L]$ satisfies the all four conditions (7), (8), (9), and (10). We note that the definition of $(Q_l)_{l=1}^L$ clearly satisfies condition (7). For the remaining conditions, namely (8), (9), (10), we proceed the proof by induction.

**When $l = 1$.** We begin by examining the three conditions (8), (9) and (10) under the base case $l = 1$. We first determine $Q_1$ and $\Delta W_1$ based on (11) and (8):

$$\Delta W_1 = (\prod_{i=2}^L W_i)^{-1}E'Q_1, \; Q_1 = I_{1:R,D}. \tag{12}$$

By the choice of $\Delta W_1$, we satisfy the condition (8). Moreover, it directly follows that $\mathrm{rank}(\Delta W_1) \leq \mathrm{rank}(Q_1) = R$, thereby fulfilling the rank constraint in (9).

Therefore, we just need to prove that $\boldsymbol{W}_1 + \Delta\boldsymbol{W}_1$ is full-rank, as required by condition (10). To compute $\mathrm{rank}(\boldsymbol{W}_1 + \Delta\boldsymbol{W}_1)$, we proceed as follows:

$$\mathrm{rank}(\boldsymbol{W}_1 + \Delta\boldsymbol{W}_1)$$

$$\overset{(12)}{=} \mathrm{rank}(\boldsymbol{W}_1 + (\prod_{i=2}^{L}\boldsymbol{W}_i)^{-1}\boldsymbol{E}'\boldsymbol{Q}_1) \qquad\qquad \text{(Substituting for } \Delta\boldsymbol{W}_1)$$

$$= \mathrm{rank}((\prod_{i=1}^{L}\boldsymbol{W}_i) + \boldsymbol{E}'\boldsymbol{Q}_1) \qquad \text{(Left multiplying with invertible } (\prod_{i=2}^{L}\boldsymbol{W}_i)^{-1})$$

$$= \mathrm{rank}((\prod_{i=1}^{L}\boldsymbol{W}_i) + \mathrm{LR}_{R\wedge R_{\boldsymbol{E}'}}(\boldsymbol{E})). \qquad\qquad \text{(Simplifying)}$$

Given the assumption that $\prod_{l=1}^{L}\boldsymbol{W}_l + \mathrm{LR}_r(\boldsymbol{E})$ is full rank for all $r \leq R(L-1)$, $\mathrm{rank}(\boldsymbol{W}_1 + \Delta\boldsymbol{W}_1) = \mathrm{rank}((\prod_{i=1}^{L}\boldsymbol{W}_i) + \mathrm{LR}_{R\wedge R_{\boldsymbol{E}'}}(\boldsymbol{E})) = D$, satisfying the last condition (10).

**When $l > 1$.** Consider $l = 2, \ldots, L$. We assume that for $i \in [l-1]$, we have determined matrices $\boldsymbol{Q}_i$ and $\Delta\boldsymbol{W}_i$ based on (11) and (8), respectively, and we assume that they satisfy the conditions (8), (9), and (10).

First, under the induction assumption that $\boldsymbol{W}_i + \Delta\boldsymbol{W}_i$ is invertible for all $i \in [l-1]$, to achieve $\boldsymbol{A}_l = \boldsymbol{E}'\boldsymbol{Q}_l$, we set $\Delta\boldsymbol{W}_l$ based on (8). This definition ensures $\mathrm{rank}(\Delta\boldsymbol{W}_l) \leq \mathrm{rank}(\boldsymbol{Q}_l) = R$, thereby satisfying the condition (9). To prove that $\boldsymbol{W}_l + \Delta\boldsymbol{W}_l$ is full-rank (condition (10)), we focus on computing $\mathrm{rank}(\boldsymbol{W}_l + \Delta\boldsymbol{W}_l)$. We proceed as follows:

$$\mathrm{rank}(\boldsymbol{W}_l + \Delta\boldsymbol{W}_l)$$

$$\overset{(8)}{=} \mathrm{rank}(\boldsymbol{W}_l + (\prod_{i=l+1}^{L}\boldsymbol{W}_i)^{-1}\boldsymbol{E}'\boldsymbol{Q}_l(\prod_{i=1}^{l-1}(\boldsymbol{W}_i + \Delta\boldsymbol{W}_i)^{-1})) \qquad \text{(Substituting for } \Delta\boldsymbol{W}_l)$$

$$= \mathrm{rank}(\boldsymbol{I}_D + (\prod_{i=l}^{L}\boldsymbol{W}_i)^{-1}\boldsymbol{E}'\boldsymbol{Q}_l(\prod_{i=1}^{l-1}(\boldsymbol{W}_i + \Delta\boldsymbol{W}_i))^{-1}) \qquad \text{(Left multiplying invertible } \boldsymbol{W}_l^{-1})$$

$$= \mathrm{rank}\Big(\prod_{i=1}^{l-1}(\boldsymbol{W}_i + \Delta\boldsymbol{W}_i) + (\prod_{i=l}^{L}\boldsymbol{W}_i)^{-1}\boldsymbol{E}'\boldsymbol{Q}_l\Big) \qquad \text{(Right multiplying invertible } \prod_{i=1}^{l-1}(\boldsymbol{W}_i + \Delta\boldsymbol{W}_i))$$

$$= \mathrm{rank}\Big((\boldsymbol{W}_{l-1} + \Delta\boldsymbol{W}_{l-1})\prod_{i=1}^{l-2}(\boldsymbol{W}_i + \Delta\boldsymbol{W}_i) + (\prod_{i=l}^{L}\boldsymbol{W}_i)^{-1}\boldsymbol{E}'\boldsymbol{Q}_l\Big) \qquad \text{(Rearranging terms)}$$

$$\overset{(8)}{=} \mathrm{rank}\Big((\boldsymbol{W}_{l-1} + (\prod_{i=l}^{L}\boldsymbol{W}_i)^{-1}\boldsymbol{E}'\boldsymbol{Q}_{l-1}(\prod_{i=1}^{l-2}(\boldsymbol{W}_i + \Delta\boldsymbol{W}_i))^{-1})\prod_{i=1}^{l-2}(\boldsymbol{W}_i + \Delta\boldsymbol{W}_i)$$

$$+ (\prod_{i=l}^{L}\boldsymbol{W}_i)^{-1}\boldsymbol{E}'\boldsymbol{Q}_l\Big) \qquad\qquad \text{(Substituting for } \Delta\boldsymbol{W}_{l-1})$$

$$= \mathrm{rank}\Big((\prod_{i=l-1}^{L}\boldsymbol{W}_i + \boldsymbol{E}'\boldsymbol{Q}_{l-1}(\prod_{i=1}^{l-2}(\boldsymbol{W}_i + \Delta\boldsymbol{W}_i))^{-1})\prod_{i=1}^{l-2}(\boldsymbol{W}_i + \Delta\boldsymbol{W}_i)$$

$$+ \boldsymbol{E}'\boldsymbol{Q}_l\Big) \qquad\qquad \text{(Left multiplying } \prod_{i=l}^{L}\boldsymbol{W}_i)$$

$$= \mathrm{rank}\Big((\prod_{i=l-1}^{L}\boldsymbol{W}_i\prod_{i=1}^{l-2}(\boldsymbol{W}_i + \Delta\boldsymbol{W}_i) + \boldsymbol{E}'\boldsymbol{Q}_{l-1} + \boldsymbol{E}'\boldsymbol{Q}_l\Big) \qquad \text{(Rearranging terms)}$$

$$= \cdots$$

$$= \text{rank}(\prod_{i=1}^{L} \boldsymbol{W}_i + \boldsymbol{E}'(\sum_{i=1}^{l} \boldsymbol{Q}_i)) \qquad \text{(Taking similar steps)}$$

$$= \text{rank}(\prod_{i=1}^{L} \boldsymbol{W}_i + \text{LR}_{Rl \wedge R_{\boldsymbol{E}'}}(\boldsymbol{E})). \qquad \text{(Simplifying)}$$

By the assumption that $\prod_{l=1}^{L} \boldsymbol{W}_l + \text{LR}_r(\boldsymbol{E})$ is full-rank for $r \leq R(L-1)$ and consequently, $\text{rank}(\boldsymbol{W}_l + \Delta\boldsymbol{W}_l) = \text{rank}(\prod_{i=1}^{L} \boldsymbol{W}_i + \text{LR}_{Rl \wedge R_{\boldsymbol{E}'}}(\boldsymbol{E})) = D$, satisfying the last condition (10).

**Conclusion of Inductive Proof.** Thus, by induction, we show that the definitions of $(\Delta\boldsymbol{W}_l)_{l=1}^{L}$ in (8) and $(\boldsymbol{Q}_l)_{l=1}^{L}$ in (11) ensure that $\boldsymbol{A}_l = \boldsymbol{E}'\boldsymbol{Q}_l$ for all $l \in [L]$. Summing over $l$ from 1 to $L$ satisfies condition (5), thereby completing the proof. □

The following lemma extends the results to a more general setting where different LoRA-ranks can be employed across layers.

**Lemma 8.** *Define error matrix* $\boldsymbol{E} := \overline{\boldsymbol{W}} - \prod_{l=1}^{L} \boldsymbol{W}_l$, *and denote its rank by* $R_{\boldsymbol{E}} = \text{rank}(\boldsymbol{E})$. *For a sequence of LoRA-ranks for all layers* $(R_l)_{l=1}^{L}$, *assume that all the weight matrices of the frozen model* $(\boldsymbol{W}_l)_{l=1}^{L}$, *and* $\prod_{l=1}^{L} \boldsymbol{W}_l + \text{LR}_r(\boldsymbol{E})$ *are non-singular for all* $r \leq \sum_{l=1}^{L-1} R_l$. *Then, the approximation error*

$$\min_{\Delta\boldsymbol{W}_l:\text{rank}(\Delta\boldsymbol{W}_l)\leq R_l} \left\| \prod_{l=1}^{L}(\boldsymbol{W}_l + \Delta\boldsymbol{W}_l) - \overline{\boldsymbol{W}} \right\|_2 = \sigma_{\sum_{l=1}^{L} R_l + 1} \underbrace{\left( \overline{\boldsymbol{W}} - \prod_{l=1}^{L} \boldsymbol{W}_l \right)}_{\text{Error matrix } \boldsymbol{E}},$$

*and the optimal solution to the matrix approximation problem satisfies* $\prod_{l=1}^{L}(\boldsymbol{W}_l + \Delta\boldsymbol{W}_l) = \prod_{l=1}^{L} \boldsymbol{W}_l + \text{LR}_{(\sum_{l=1}^{L} R_l) \wedge R_{\boldsymbol{E}}}(\boldsymbol{E})$. *Therefore, when* $\sum_{l=1}^{L} R_l \geq R_{\boldsymbol{E}}$, *we have* $\prod_{l=1}^{L}(\boldsymbol{W}_l + \Delta\boldsymbol{W}_l) = \overline{\boldsymbol{W}}$, *implying* $f \equiv \overline{f}$.

*Proof of Lemma 8.* The proof follows the same steps of Lemma 7 with only minor modifications.

In the current setting, we target the following constrained optimization problem:

$$\min_{\Delta\boldsymbol{W}_l:\text{rank}(\Delta\boldsymbol{W}_l)\leq R_l} \left\| \prod_{l=1}^{L}(\boldsymbol{W}_l + \Delta\boldsymbol{W}_l) - \overline{\boldsymbol{W}} \right\|_2,$$

where we allow each LoRA adapter $\Delta\boldsymbol{W}_l$ can possess different LoRA-ranks $R_l$, i.e., $\text{rank}(\Delta\boldsymbol{W}_l) \leq R_l, l \in [L]$. Subtracting $\prod_{l=1}^{L} \boldsymbol{W}_l$ from both terms leads us to a similar constrained optimization problem as (2). The only distinction lies in the rank constraint:

$$\min_{\Delta\boldsymbol{W}_l:\text{rank}(\Delta\boldsymbol{W}_l)\leq R_l} \left\| \underbrace{\left( \prod_{l=1}^{L}(\boldsymbol{W}_l + \Delta\boldsymbol{W}_l) - \prod_{l=1}^{L} \boldsymbol{W}_l \right)}_{:=\boldsymbol{A}} - \underbrace{\left( \overline{\boldsymbol{W}} - \prod_{l=1}^{L} \boldsymbol{W}_l \right)}_{:=\boldsymbol{E}} \right\|_2. \qquad (13)$$

Following the same steps, we decompose $\boldsymbol{A}$ into (3). Given that $\text{rank}(\boldsymbol{A}_l) \leq \text{rank}(\Delta\boldsymbol{W}_l) \leq R_l$, we deduce that $\text{rank}(\boldsymbol{A}) \leq \sum_{l=1}^{L} \text{rank}(\boldsymbol{A}_l) \leq \sum_{l=1}^{L} R_l$. Consequently, the optimization problem above can be eased into a low-rank approximation problem analogous to (4):

$$(13) \geq \min_{\boldsymbol{A}:\text{rank}(\boldsymbol{A})\leq\sum_{l=1}^{L} R_l} \|\boldsymbol{A} - \boldsymbol{E}\|_2,$$

where the optimal solution is $\boldsymbol{A} = \text{LR}_{(\sum_{l=1}^{L} R_l) \wedge R_{\boldsymbol{E}}}(\boldsymbol{E}) := \boldsymbol{E}'$. Therefore, if we can identify the LoRA adapters $(\Delta\boldsymbol{W}_l)_{l=1}^{L}$ with $\text{rank}(\Delta\boldsymbol{W}_l) \leq R_l$ such that

$$\underbrace{\prod_{l=1}^{L}(\boldsymbol{W}_l + \Delta\boldsymbol{W}_l) - \prod_{l=1}^{L} \boldsymbol{W}_l}_{:=\boldsymbol{A}} = \underbrace{\text{LR}_{(\sum_{l=1}^{L} R_l) \wedge R_{\boldsymbol{E}}}(\overline{\boldsymbol{W}} - \prod_{l=1}^{L} \boldsymbol{W}_l)}_{:=\boldsymbol{E}'},$$

the proof is completed.

The remaining part of the proof adheres to the steps outlined in the proof of Lemma 7 deriving (5). The only difference is that we consider a different selection of $(\boldsymbol{Q}_l)l = 1^L$ that satisfies (9) here:

$$\boldsymbol{Q}_l = \boldsymbol{V} \boldsymbol{I}_{(\sum_{i=1}^{l-1} R_i) \wedge R_{\boldsymbol{E}'} : (\sum_{i=1}^{l} R_i) \wedge R_{\boldsymbol{E}'}, D} \boldsymbol{V}^\top.$$

Applying the same steps with this change yields the desired outcomes. $\square$

This lemma illustrates that in linear cases, the total number of parameters needed to achieve an exact approximation is constant, regardless of LoRA-rank assignment. It suggests that applying a LoRA-rank of $R$ per layer is equivalent to applying a LoRA-rank of $RL$ at the final layer. As a result, fine-tuning only the last layer, which involves assigning a LoRA-rank of $D$ to the last layer, is equivalent to implementing LoRA where each adapter is constrained to have a rank of $D/L$. Both methods can achieve an exact approximation and maintain the same parameter efficiency.

## E    PROOFS FOR FNN APPROXIMATION

In this section, we provide the full proof for deriving the main results outlined in Sec. 3. For the sake of completeness, we restate our results from the main body before presenting the proof.

### E.1    APPROXIMATING ONE-LAYER ReLU FNN VIA LoRA

We first provide a slightly more detailed result on the one-layer ReLU FNN approximation (Lemma 9) along with its corresponding proof. Then, we present a variant of this lemma by allowing for different LoRA-ranks for each low-rank adapter. The proof for this variant involves only a minor modification of the proof for Lemma 9.

**Lemma 9** (Detailed version of Lemma 2). *Define error matrix $\boldsymbol{E} := \overline{\boldsymbol{W}}_1 - \prod_{l=1}^{L} \boldsymbol{W}_l$, with its rank represented by $R_{\boldsymbol{E}} = \mathrm{rank}(\boldsymbol{E})$. Consider a LoRA-rank $R \in [D]$. Assume that the weight matrices $\boldsymbol{W}_1, \ldots, \boldsymbol{W}_L \in \mathbb{R}^{D \times D}$ and $\prod_{l=1}^{L} \boldsymbol{W}_l + \mathrm{LR}_r(\boldsymbol{E})$ for all $r \le R(L-1)$ are non-singular. Let $\mathbf{x}$ be a random input sampled from a distribution with bounded support $\mathcal{X}$ and let $\Sigma = \mathbb{E}\mathbf{x}\mathbf{x}^\top$. Then, there exists rank-$R$ or lower matrices $\Delta\boldsymbol{W}_1, \ldots, \Delta\boldsymbol{W}_L \in \mathbb{R}^{D \times D}$ and bias vectors $\widehat{\boldsymbol{b}}_1, \ldots, \widehat{\boldsymbol{b}}_L \in \mathbb{R}^D$ such that for any input $\boldsymbol{x} \in \mathcal{X}$,*

$$f(\boldsymbol{x}) - \bar{f}(\boldsymbol{x}) = ReLU\left( \left( \mathrm{LR}_{RL \wedge R_{\boldsymbol{E}}}(\overline{\boldsymbol{W}}_1 - \prod_{l=1}^{L} \boldsymbol{W}_l) - (\overline{\boldsymbol{W}}_1 - \prod_{l=1}^{L} \boldsymbol{W}_l) \right) \boldsymbol{x} \right).$$

*Therefore, when $R \ge \lceil R_{\boldsymbol{E}}/L \rceil$, the adapted model exactly approximates the target model, i.e., $f(\boldsymbol{x}) = \bar{f}(\boldsymbol{x})$ for all $\boldsymbol{x} \in \mathcal{X}$.*

*Furthermore, let $\mathbf{x}$ be a random input sampled from a distribution with bounded support $\mathcal{X}$ and let $\Sigma = \mathbb{E}\mathbf{x}\mathbf{x}^\top$. Then, the expected squared error is bounded as*

$$\mathbb{E}\left\| f(\mathbf{x}) - \bar{f}(\mathbf{x}) \right\|_2^2 \le \|\Sigma\|_F\, \sigma_{RL \wedge R_{\boldsymbol{E}}+1}^2 (\overline{\boldsymbol{W}}_1 - \prod_{l=1}^{L} \boldsymbol{W}_l).$$

*Proof of Lemma 9.* This proof consists of three main steps: (i) linearize the first $L-1$ layers of the adapted model $f$ to reduce it to a single-layer FNN, (ii) align the weight matrices and bias vectors of this simplified $f$ with those of the target model $\bar{f}$, (iii) derive an upper bound of the error $\mathbb{E}\left\| f(\mathbf{x}) - \bar{f}(\mathbf{x}) \right\|_2^2$.

**Linearization.**    The main challenge here stems from the non-linearities introduced by the ReLU activation function. To remove the non-linearities in the first $L-1$ layers of updated model $f$, since the input space $\mathcal{X}$ is bounded, we can set all the entries of $\widehat{\boldsymbol{b}}_1, \ldots, \widehat{\boldsymbol{b}}_{L-1}$ sufficiently large, thereby

activating all ReLUs in the first $L - 1$ layers of $f$. Consequently, we have

$$
\begin{aligned}
f(\boldsymbol{x}) &= \text{ReLU}((\boldsymbol{W}_L + \Delta \boldsymbol{W}_L)\boldsymbol{z}_{L-1} + \widehat{\boldsymbol{b}}_L) \\
&= \text{ReLU}\left((\boldsymbol{W}_L + \Delta \boldsymbol{W}_L)\text{ReLU}((\boldsymbol{W}_{L-1} + \Delta \boldsymbol{W}_{L-1})\boldsymbol{z}_{L-2} + \widehat{\boldsymbol{b}}_{L-1}) + \widehat{\boldsymbol{b}}_L\right) \\
&= \text{ReLU}\left((\boldsymbol{W}_L + \Delta \boldsymbol{W}_L)((\boldsymbol{W}_{L-1} + \Delta \boldsymbol{W}_{L-1})\boldsymbol{z}_{L-2} + \widehat{\boldsymbol{b}}_{L-1}) + \widehat{\boldsymbol{b}}_L\right) \\
&= \text{ReLU}\left((\boldsymbol{W}_L + \Delta \boldsymbol{W}_L)(\boldsymbol{W}_{L-1} + \Delta \boldsymbol{W}_{L-1})\boldsymbol{z}_{L-2} + (\boldsymbol{W}_L + \Delta \boldsymbol{W}_L)\widehat{\boldsymbol{b}}_{L-1} + \widehat{\boldsymbol{b}}_L\right) \\
&= \cdots \\
&= \text{ReLU}\left(\prod_{l=1}^{L}(\boldsymbol{W}_l + \Delta \boldsymbol{W}_l)\boldsymbol{x} + (\sum_{l=1}^{L-1}\prod_{i=l+1}^{L}(\boldsymbol{W}_i + \Delta \boldsymbol{W}_i)\widehat{\boldsymbol{b}}_l) + \widehat{\boldsymbol{b}}_L\right),
\end{aligned}
$$

which is equivalent to a single-layer ReLU neural network with weight matrix $\prod_{l=1}^{L}(\boldsymbol{W}_l + \Delta \boldsymbol{W}_l)$ and bias vector $(\sum_{l=1}^{L-1}\prod_{i=l+1}^{L}(\boldsymbol{W}_i + \Delta \boldsymbol{W}_i)\widehat{\boldsymbol{b}}_l) + \widehat{\boldsymbol{b}}_L$.

**Parameter Alignment.** To match the updated model $f(\boldsymbol{x})$ and target model $\bar{f}(\boldsymbol{x})$, we proceed as follows. For weight matrix, Lemma 7 guarantees the existence of rank-$R$ or lower matrices $\Delta \boldsymbol{W}_1, \ldots, \Delta \boldsymbol{W}_L \in \mathbb{R}^{D \times D}$ such that

$$
\prod_{l=1}^{L}(\boldsymbol{W}_l + \Delta \boldsymbol{W}_l) = \prod_{l=1}^{L} \boldsymbol{W}_l + \text{LR}_{RL \wedge R_{\boldsymbol{E}}}(\overline{\boldsymbol{W}} - \prod_{l=1}^{L} \boldsymbol{W}_l). \tag{14}
$$

For the bias vector, we set $\widehat{\boldsymbol{b}}_L = \bar{\boldsymbol{b}}_1 - \sum_{l=1}^{L-1}\prod_{i=l+1}^{L}(\boldsymbol{W}_i + \Delta \boldsymbol{W}_i)\widehat{\boldsymbol{b}}_l$ such that $\sum_{l=1}^{L-1}\prod_{i=l+1}^{L}(\boldsymbol{W}_i + \Delta \boldsymbol{W}_i)\widehat{\boldsymbol{b}}_l + \widehat{\boldsymbol{b}}_L = \bar{\boldsymbol{b}}_1$. Therefore, we obtain

$$
f(\boldsymbol{x}) - \bar{f}(\boldsymbol{x}) = \text{ReLU}\left(\left(\text{LR}_{RL \wedge R_{\boldsymbol{E}}}(\overline{\boldsymbol{W}}_1 - \prod_{l=1}^{L} \boldsymbol{W}_l) - (\overline{\boldsymbol{W}}_1 - \prod_{l=1}^{L} \boldsymbol{W}_l)\right)\boldsymbol{x}\right).
$$

**Error Derivation.** We compute the expected squared error as follows:

$$
\begin{aligned}
&\mathbb{E}\left\|f(\mathbf{x}) - \bar{f}(\mathbf{x})\right\|_2^2 \\
&\leq \mathbb{E}\left\|\left(\text{LR}_{RL \wedge R_{\boldsymbol{E}}}(\overline{\boldsymbol{W}}_1 - \prod_{l=1}^{L} \boldsymbol{W}_l) - (\overline{\boldsymbol{W}}_1 - \prod_{l=1}^{L} \boldsymbol{W}_l)\right)\mathbf{x}\right\|_2^2 && \text{(ReLU is 1-Lipschitz)} \\
&\overset{(1)}{\leq} \left\|\text{LR}_{RL \wedge R_{\boldsymbol{E}}}(\overline{\boldsymbol{W}}_1 - \prod_{l=1}^{L} \boldsymbol{W}_l) - (\overline{\boldsymbol{W}}_1 - \prod_{l=1}^{L} \boldsymbol{W}_l)\right\|_2^2 \mathbb{E}\left\|\mathbf{x}\right\|_2^2 \\
&= \|\Sigma\|_{\text{F}}\, \sigma_{RL \wedge R_{\boldsymbol{E}}+1}^2(\overline{\boldsymbol{W}}_1 - \prod_{l=1}^{L} \boldsymbol{W}_l). && \text{(By the definition of } \text{LR}_{RL \wedge R_{\boldsymbol{E}}}(\cdot))
\end{aligned}
$$

This completes the proof. $\qquad\square$

Lemma 9 is extended to cases where different LoRA-ranks can be used for different low-rank adapters, as detailed in the following lemma.

**Lemma 10.** *Define error matrix $\boldsymbol{E} := \overline{\boldsymbol{W}}_1 - \prod_{l=1}^{L} \boldsymbol{W}_l$, and denote its rank by $R_{\boldsymbol{E}} = \text{rank}(\boldsymbol{E})$. Consider a sequence of LoRA-ranks $(R_l)_{l=1}^{L}$. Assume that the weight matrices $\boldsymbol{W}_1, \ldots, \boldsymbol{W}_L \in \mathbb{R}^{D \times D}$ and $\prod_{l=1}^{L} \boldsymbol{W}_l + \text{LR}_r(\boldsymbol{E})$ for all $r \leq \sum_{l=1}^{L-1} R_l$ are non-singular. Then, there LoRA adapters $(\Delta \boldsymbol{W}_l)_{l=1}^{L}$ satisfying the rank constraints $\text{rank}(\Delta \boldsymbol{W}_l) \leq R_l$ for all $l \in [L]$ and bias vectors $\widehat{\boldsymbol{b}}_1, \ldots, \widehat{\boldsymbol{b}}_L \in \mathbb{R}^D$ such that for any input $\boldsymbol{x} \in \mathcal{X}$,*

$$
f(\boldsymbol{x}) - \bar{f}(\boldsymbol{x}) = \text{ReLU}\left(\left(\text{LR}_{(\sum_{l=1}^{L} R_l) \wedge R_{\boldsymbol{E}}}(\overline{\boldsymbol{W}}_1 - \prod_{l=1}^{L} \boldsymbol{W}_l) - (\overline{\boldsymbol{W}}_1 - \prod_{l=1}^{L} \boldsymbol{W}_l)\right)\boldsymbol{x}\right).
$$

*Therefore, when $\sum_{l=1}^{L} R_l \geq R_E$, the adapted model exactly approximates the target model, i.e., $f(\boldsymbol{x}) = \bar{f}(\boldsymbol{x})$ for all $\boldsymbol{x} \in \mathcal{X}$.*

*Furthermore, for a random input $\mathbf{x}$ drawn from a distribution supported on $\mathcal{X}$, and with $\Sigma = \mathbb{E}\mathbf{x}\mathbf{x}^\top$, the expected squared error is bounded by:*

$$\mathbb{E}\left\|f(\mathbf{x}) - \bar{f}(\mathbf{x})\right\|_2^2 \leq \|\Sigma\|_F \, \sigma_{(\sum_{l=1}^{L} R_l) \wedge R_E + 1}^2 (\overline{\boldsymbol{W}}_1 - \prod_{l=1}^{L} \boldsymbol{W}_l).$$

*Proof of Lemma 10.* This proof closely adheres to the steps detailed in the proof of Lemma 9.

The primary change implemented here is that, when we draw the analogy to (14), we apply Lemma 8 instead of Lemma 7. This results in

$$\prod_{l=1}^{L}(\boldsymbol{W}_l + \Delta\boldsymbol{W}_l) = \prod_{l=1}^{L} \boldsymbol{W}_l + \mathrm{LR}_{(\sum_{l=1}^{L} R_l) \wedge R_E}(\overline{\boldsymbol{W}} - \prod_{l=1}^{L} \boldsymbol{W}_l).$$

Utilizing the steps from the proof of Lemma 9 and integrating the modification specified above, we can establish the desired result. $\square$

## E.2 Approximating Multi-Layer ReLU FNN via LoRA with Uniform Model Parition

In this part, we restate all the results considering uniform model partition from Sec. 3.3, along with their corresponding proofs, presented in the same order.

**Assumption 1** (Non-Singularity). *For a fixed LoRA-rank $R \in [D]$, the weight matrices of the frozen model $(\boldsymbol{W}_l)_{l=1}^{L}$ and matrices $\left(\prod_{l \in P_i^{\mathrm{u}}} \boldsymbol{W}_l\right) + \mathrm{LR}_r(\overline{\boldsymbol{W}}_i - \prod_{l \in P_i^{\mathrm{u}}} \boldsymbol{W}_l)$ are non-singular for all $r \leq R(M-1)$ and $i \in [\overline{L}]$.*

**Lemma 3.** *Let $(\overline{\boldsymbol{W}}_l)_{l=1}^{\overline{L}}, (\boldsymbol{W}_l)_{l=1}^{L} \in \mathbb{R}^{D \times D}$ be matrices whose elements are drawn independently from arbitrary continuous distributions. Then, with probability 1, Assumption 1 holds $\forall R \in [D]$.*

*Proof of Lemma 3.* We first use Lemma 6 to establish that $\overline{\boldsymbol{W}}_1, \ldots, \overline{\boldsymbol{W}}_{\overline{L}}, \boldsymbol{W}_1, \ldots, \boldsymbol{W}_L$ are non-singular with probability 1. The goal of the remaining proof is to demonstrate that $\left(\prod_{l \in P_i^{\mathrm{u}}} \boldsymbol{W}_l\right) + \mathrm{LR}_r(\overline{\boldsymbol{W}}_i - \prod_{l \in P_i^{\mathrm{u}}} \boldsymbol{W}_l)$ is full-rank with probability 1. In this proof, we use $p.$ to denote the probability density function, where the subscript indicates the associated random variable.

Fix an arbitrary $i \in [\overline{L}]$ and $r \in [R]$. Then probability of the $\left(\prod_{l \in P_i^{\mathrm{u}}} \boldsymbol{W}_l\right) + \mathrm{LR}_r\left(\overline{\boldsymbol{W}}_i - \prod_{l \in P_i^{\mathrm{u}}} \boldsymbol{W}_l\right)$ being full-rank can be computed as

$$\mathbb{P}\left\{\det\left(\left(\prod_{l \in P_i^{\mathrm{u}}} \boldsymbol{W}_l\right) + \mathrm{LR}_r\left(\overline{\boldsymbol{W}}_i - \prod_{l \in P_i^{\mathrm{u}}} \boldsymbol{W}_l\right)\right) \neq 0\right\}$$

$$= \int_{\mathcal{E}} \mathbb{P}\left\{\det\left(\left(\prod_{l \in P_i^{\mathrm{u}}} \boldsymbol{W}_l\right) + \mathrm{LR}_r(\boldsymbol{E})\right) \neq 0 \,\middle|\, \overline{\boldsymbol{W}}_i - \prod_{l \in P_i^{\mathrm{u}}} \boldsymbol{W}_l = \boldsymbol{E}\right\} p_{\overline{\boldsymbol{W}}_i - \prod_{l \in P_i^{\mathrm{u}}} \boldsymbol{W}_l}(\boldsymbol{E})\mathrm{d}\boldsymbol{E}.$$

If the conditional random matrix $\left(\prod_{l \in P_i^{\mathrm{u}}} \boldsymbol{W}_l\right) + \mathrm{LR}_r(\boldsymbol{E}) \mid \overline{\boldsymbol{W}}_i - \prod_{l \in P_i^{\mathrm{u}}} \boldsymbol{W}_l = \boldsymbol{E}$ has a continuous distribution with support of non-zero Lebesgue measure on $\mathbb{R}^{D \times D}$, then

$$\mathbb{P}\left\{\det\left(\left(\prod_{l \in P_i^{\mathrm{u}}} \boldsymbol{W}_l\right) + \mathrm{LR}_r(\boldsymbol{E})\right) \neq 0 \,\middle|\, \overline{\boldsymbol{W}}_i - \prod_{l \in P_i^{\mathrm{u}}} \boldsymbol{W}_l = \boldsymbol{E}\right\} = 1$$

ensuring $\left(\prod_{l \in P_i^{\mathrm{u}}} \boldsymbol{W}_l\right) + \mathrm{LR}_r\left(\overline{\boldsymbol{W}}_i - \prod_{l \in P_i^{\mathrm{u}}} \boldsymbol{W}_l\right)$ is full-rank with probability 1.

Consequently, the remaining part of the proof aims to show that the conditional random matrix $\left(\prod_{l \in P_i^u} \mathbf{W}_l\right) + \mathrm{LR}_r(\boldsymbol{E}) \mid \overline{\mathbf{W}}_i - \prod_{l \in P_i^u} \mathbf{W}_l = \boldsymbol{E}$ follows arbitrary continuous distribution with support having non-zero Lebesgue measure on $\mathbb{R}^{D \times D}$. Denote $\mathbf{W} = \prod_{l \in P_i^u} \mathbf{W}_l$. Now, consider the conditional distribution of $\prod_{l \in P_i^u} \mathbf{W}_l \mid \overline{\mathbf{W}}_i - \prod_{l \in P_i^u} \mathbf{W}_l = \boldsymbol{E}$, which can be written as

$$p_{\mathbf{W} \mid \overline{\mathbf{W}}_i - \mathbf{W} = \boldsymbol{E}}(\boldsymbol{W}) = p_{\overline{\mathbf{W}}_i}(\boldsymbol{E} + \boldsymbol{W}).$$

Since $p_{\overline{\mathbf{W}}_i}$ is continuous with support of non-zero Lebesgue measure on $\mathbb{R}^{D \times D}$, the same holds for $\prod_{l \in P_i^u} \mathbf{W}_l \mid \overline{\mathbf{W}}_i - \prod_{l \in P_i^u} \mathbf{W}_l = \boldsymbol{E}$. Furthermore, adding a constant matrix $\mathrm{LR}_r(\boldsymbol{E})$ to this conditional distribution preserves the desired properties, thus completing the proof. $\qquad \square$

**Theorem 3.** *Under Assumption 1, there exists rank-$R$ or lower matrices $(\Delta \boldsymbol{W}_l)_{l=1}^L$ with $\Delta \boldsymbol{W}_l \in \mathbb{R}^{D \times D}$ and bias vectors $(\widehat{\boldsymbol{b}}_l)_{l=1}^L$ with $\widehat{\boldsymbol{b}}_l \in \mathbb{R}^D$ when the rank of the low-rank adapter $R \geq \lceil \max_{i \in [\overline{L}]} \mathrm{rank}(\overline{\boldsymbol{W}}_i - \prod_{l \in P_i^u} \boldsymbol{W}_l)/M \rceil$, the low-rank adapted model $f$ can exactly approximate the target model $\overline{f}$, i.e., $f(\boldsymbol{x}) = \overline{f}(\boldsymbol{x})$ for all input $\boldsymbol{x} \in \mathcal{X}$.*

*Proof of Theorem 3.* The key to this proof lies in a simple idea: for each layer $i \in [\overline{L}]$ in the target model, we can update $M$ layers (i.e., $(i-1)M + 1$-th layer to $iM$-th layer) in the frozen model to approximate it as guaranteed by Lemma 9. Hence, all layers of the target model can be approximated by the adapted model.

**Model Decomposition.** We partition the adapted model $f$ into $\overline{L}$ sub-models, each defined as

$$f_i(\cdot) = \mathrm{FNN}_{\overline{L}, D}(\cdot; (\boldsymbol{W}_l + \Delta \boldsymbol{W}_l)_{l \in P_i^u}, (\widehat{\boldsymbol{b}}_l)_{l \in P_i^u}), \quad i \in [\overline{L}].$$

In a similar manner, we break down $\overline{f}$ into $\overline{L}$ sub-models, each is a one-layer FNN:

$$\overline{f}_i(\cdot) = \mathrm{FNN}_{1, D}(\cdot; \overline{\boldsymbol{W}}_i, \overline{\boldsymbol{b}}_i), \ i \in [\overline{L}].$$

We can then express $f(\boldsymbol{x})$ and $\overline{f}(\boldsymbol{x})$ as compositions of their respective sub-models:

$$f(\cdot) = f_{\overline{L}} \circ \cdots f_1(\cdot), \quad \overline{f}(\cdot) = \overline{f}_{\overline{L}} \circ \cdots \overline{f}_1(\cdot).$$

To analyze the error $\mathbb{E} \left\| f(\mathbf{x}) - \overline{f}(\mathbf{x}) \right\|_2 = \mathbb{E} \left\| f(\mathbf{x}) - \overline{f}(\mathbf{x}) \right\|_2$, we consider the error caused by each submodel. Let $\widetilde{R}_i = \mathrm{rank}(\overline{\boldsymbol{W}}_i - \prod_{l \in P_i^u} \boldsymbol{W}_l)$ denote the rank of the discrepancy between the target weight matrix and the frozen weight matrices, where $i \in [\overline{L}]$. By Lemma 9, we can select $\Delta \boldsymbol{W}_1, \ldots, \Delta \boldsymbol{W}_L, \widehat{\boldsymbol{b}}_1, \ldots, \widehat{\boldsymbol{b}}_L$ such that

$$f_i(\boldsymbol{z}) - \overline{f}_i(\boldsymbol{z}) = \mathrm{ReLU}\left(\left(\mathrm{LR}_{RL \wedge \widetilde{R}_i}(\overline{\boldsymbol{W}}_i - \prod_{l \in P_i^u} \boldsymbol{W}_l) - (\overline{\boldsymbol{W}}_i - \prod_{l \in P_i^u} \boldsymbol{W}_l)\right) \boldsymbol{z}\right), \quad (15)$$

$$\mathbb{E} \left\| f_i(\mathbf{z}) - \overline{f}_i(\mathbf{z}) \right\|_2^2 \leq \left\| \mathbb{E} \mathbf{z} \mathbf{z}^\top \right\|_{\mathrm{F}} \sigma_{RL \wedge \widetilde{R}_i + 1}^2 (\overline{\boldsymbol{W}}_i - \prod_{l=1}^L \boldsymbol{W}_l). \quad (16)$$

Given these selected parameters, $f_i$ is functionally equivalent to a one-layer FNN:

$$f_i(\boldsymbol{z}) = \mathrm{ReLU}\left(\left(\mathrm{LR}_{RL \wedge \widetilde{R}_i}(\overline{\boldsymbol{W}}_i - \prod_{l \in P_i^u} \boldsymbol{W}_l) + \prod_{l \in P_i^u} \boldsymbol{W}_l\right) \boldsymbol{z}\right).$$

Clearly, when $R \geq \max_i \lceil \frac{\widetilde{R}_i}{M} \rceil$, it follows that $f_i = g_i$ for all $i \in [\overline{L}]$, which implies $f = g$. $\qquad \square$

**Corollary 4.** *Assume that the elements of matrices $(\overline{\boldsymbol{W}}_l)_{l=1}^{\overline{L}}, (\boldsymbol{W}_l)_{l=1}^L$ are independently drawn from arbitrary continuous distributions. When $R \geq D/M$, there exists rank-$R$ or lower matrices $\Delta \boldsymbol{W}_1, \ldots, \Delta \boldsymbol{W}_L \in \mathbb{R}^{D \times D}$ and bias vectors $\widehat{\boldsymbol{b}}_1, \ldots, \widehat{\boldsymbol{b}}_L \in \mathbb{R}^D$ such that low-rank adapted model $f$ can functionally cover the target model $\overline{f}$ on $\mathcal{X}$, i.e., $f(\boldsymbol{x}) = \overline{f}(\boldsymbol{x})$ for all input $\boldsymbol{x} \in \mathcal{X}$, with probability 1.*

*Proof of Corollary 4.* To prove the statement, we start by noting that combining Lemma 3 and Theorem 3 directly gives us $f(\boldsymbol{x}) = \bar{f}(\boldsymbol{x})$ on $\mathcal{X}$ when $R \geq \max_{i \in [\bar{L}]} \lceil \text{rank}(\overline{\boldsymbol{W}}_i - \prod_{l \in P_i^{\mathrm{u}}} \boldsymbol{W}_l)/M \rceil$. Therefore, the only thing left is to show that $\text{rank}(\overline{\mathbf{W}}_i - \prod_{l \in P_i^{\mathrm{u}}} \mathbf{W}_l) = D$ for $i \in [\bar{L}]$ with probability 1. In this proof, we use $p_{\cdot}$ to denote the probability density function, where the subscript indicates the associated random variable.

To establish this, consider the following probability expression:

$$\mathbb{P}\left\{ \det\left( \overline{\mathbf{W}}_i - \prod_{l \in P_i^{\mathrm{u}}} \mathbf{W}_l \right) \neq 0 \right\}$$

$$= \int \mathbb{P}\left\{ \det\left( \overline{\mathbf{W}}_i - \boldsymbol{W} \right) \neq 0 \,\middle|\, \prod_{l \in P_i^{\mathrm{u}}} \mathbf{W}_l = \boldsymbol{W} \right\} p_{\prod_{l \in P_i^{\mathrm{u}}} \mathbf{W}_l}(\boldsymbol{W}) \mathrm{d}\boldsymbol{W}.$$

Since $\overline{\mathbf{W}}$ is independent of $\prod_{l \in P_i^{\mathrm{u}}} \mathbf{W}_l$, we have

$$\mathbb{P}\left\{ \det\left( \overline{\mathbf{W}}_i - \boldsymbol{W} \right) \neq 0 \,\middle|\, \prod_{l \in P_i^{\mathrm{u}}} \mathbf{W}_l = \boldsymbol{W} \right\} = \mathbb{P}\left\{ \det\left( \overline{\mathbf{W}}_i - \boldsymbol{W} \right) \neq 0 \right\} \xlongequal{\text{Lemma 6}} 1.$$

Therefore, we conclude that $\mathbb{P}\left\{ \det\left( \overline{\mathbf{W}}_i - \prod_{l \in P_i^{\mathrm{u}}} \mathbf{W}_l \right) \neq 0 \right\} = 1$, which completes the proof. $\square$

**Theorem 5.** *Define the approximation error of $i$-th layer as $E_i = \sigma_{RM+1}(\overline{\boldsymbol{W}}_i - \prod_{l \in P_i^{\mathrm{u}}} \boldsymbol{W}_l)$, and the magnitude of the parameters and the input as $\beta := \max_{i \in [\bar{L}]} \left( \sqrt{\|\Sigma\|_F} \prod_{j=1}^{i} \|\overline{\boldsymbol{W}}_j\|_F + \sum_{j=1}^{i} \prod_{k=j+1}^{i-1} \|\overline{\boldsymbol{W}}_k\|_F \|\overline{\boldsymbol{b}}_j\|_2 \right) \vee \sqrt{\|\Sigma\|_F}$.*

*Under Assumption 1, there exists rank-$R$ or lower matrices $(\Delta \boldsymbol{W}_l)_{l=1}^{L}$ with $\Delta \boldsymbol{W}_l \in \mathbb{R}^{D \times D}$ and bias vectors $(\widehat{\boldsymbol{b}}_l)_{l=1}^{L}$ with $\widehat{\boldsymbol{b}}_l \in \mathbb{R}^D$ such that for input $\mathbf{x} \in \mathcal{X}$ with $\mathbb{E}\mathbf{x}\mathbf{x}^\top = \Sigma$,*

$$\mathbb{E}\left\| f(\mathbf{x}) - \bar{f}(\mathbf{x}) \right\|_2 \leq \beta \sum_{i=1}^{\bar{L}} \max_{k \in [\bar{L}]} \left( \|\overline{\boldsymbol{W}}_k\|_F + E_k \right)^{\bar{L}-i} E_i.$$

*Proof of Theorem 5.* This proof is a continuation of the proof of Theorem 3. In this proof, we will consider a more general case, without enforcing any constraints on the rank of the adapters $R$. We use $\widehat{\boldsymbol{W}}_i$ to denote the corresponding weight matrix, i.e., $\widehat{\boldsymbol{W}}_i = \text{LR}_{RL \wedge \widetilde{R}_i}(\overline{\boldsymbol{W}}_1 - \prod_{l \in P_i^{\mathrm{u}}} \boldsymbol{W}_l) + \prod_{l \in P_i^{\mathrm{u}}} \boldsymbol{W}_l$.

**Error Decomposition.** For submodel $i = 2, \ldots, \bar{L}$, we calculate the expected error of the composition of the first $i$ sub-models,

$$\mathbb{E}\left\| \widehat{\mathbf{z}}_i - \overline{\mathbf{z}}_i \right\|_2 = \mathbb{E}\left\| f_i(\widehat{\mathbf{z}}_{i-1}) - \bar{f}_i(\overline{\mathbf{z}}_{i-1}) \right\|_2 \tag{17}$$

$$= \mathbb{E}\left\| \left( f_i(\widehat{\mathbf{z}}_{i-1}) - f_i(\overline{\mathbf{z}}_{i-1}) \right) + \left( f_i(\overline{\mathbf{z}}_{i-1}) - \bar{f}_i(\overline{\mathbf{z}}_{i-1}) \right) \right\|_2 \qquad \text{(Rearranging terms)}$$

$$\leq \underbrace{\mathbb{E}\left\| f_i(\widehat{\mathbf{z}}_{i-1}) - f_i(\overline{\mathbf{z}}_{i-1}) \right\|_2}_{A_i} + \underbrace{\mathbb{E}\left\| f_i(\overline{\mathbf{z}}_{i-1}) - \bar{f}_i(\overline{\mathbf{z}}_{i-1}) \right\|_2}_{B_i}. \qquad \text{(Applying triangle inequality)}$$

Here $A_i$ represents the error resulting from the discrepancy between the first $i-1$ submodels, while $B_i$ represents the error arising from the mismatch between the $i$-th submodel.

**Computing $A_i$.** We start by computing the error introduced by the first $i-1$ submodels, denoted by $A_i$:

$$A_i = \mathbb{E} \left\| f_i(\widehat{\mathbf{z}}_{i-1}) - f_i(\overline{\mathbf{z}}_{i-1}) \right\|_2 = \mathbb{E} \left\| \text{ReLU}(\widehat{\boldsymbol{W}}_i(\widehat{\mathbf{z}}_{i-1} - \overline{\mathbf{z}}_{i-1})) \right\|_2$$

$$\leq \mathbb{E} \left\| \widehat{\boldsymbol{W}}_i(\widehat{\mathbf{z}}_{i-1} - \overline{\mathbf{z}}_{i-1}) \right\|_2 \qquad \text{(ReLU is 1-Lipschitz)}$$

$$\overset{(1)}{\leq} \left\| \widehat{\boldsymbol{W}}_i \right\|_{\text{F}} \mathbb{E} \left\| \widehat{\mathbf{z}}_{i-1} - \overline{\mathbf{z}}_{i-1} \right\|_2. \tag{18}$$

Here,

$$\left\| \widehat{\boldsymbol{W}}_i \right\|_{\text{F}} = \left\| \prod_{l \in P_i^{\text{u}}} \boldsymbol{W}_l + \text{LR}_{RM \wedge \widetilde{R}_i}(\overline{\boldsymbol{W}}_i - \prod_{l \in P_i^{\text{u}}} \boldsymbol{W}_l) \right\|_{\text{F}}$$

$$= \left\| \overline{\boldsymbol{W}}_i + \left( \prod_{l \in P_i^{\text{u}}} \boldsymbol{W}_l - \overline{\boldsymbol{W}}_i \right) + \text{LR}_{RM \wedge \widetilde{R}_i}(\overline{\boldsymbol{W}}_i - \prod_{l \in P_i^{\text{u}}} \boldsymbol{W}_l) \right\|_{\text{F}} \qquad \text{(Rearranging terms)}$$

$$\leq \left\| \overline{\boldsymbol{W}}_i \right\|_{\text{F}} + \left\| \left( \prod_{l \in P_i^{\text{u}}} \boldsymbol{W}_l - \overline{\boldsymbol{W}}_i \right) + \text{LR}_{RM \wedge \widetilde{R}_i}(\overline{\boldsymbol{W}}_i - \prod_{l \in P_i^{\text{u}}} \boldsymbol{W}_l) \right\|_{\text{F}}$$

$$\text{(Applying triangle inequality)}$$

$$= \left\| \overline{\boldsymbol{W}}_i \right\|_{\text{F}} + \sqrt{\sum_{j=RM \wedge \widetilde{R}_i+1}^{D} \sigma_j^2(\overline{\boldsymbol{W}}_i - \prod_{l \in P_i^{\text{u}}} \boldsymbol{W}_l)} \tag{19}$$

$$\text{(By the definition of } \overline{\boldsymbol{W}}_i \text{ and } \text{LR}_{RM \wedge \widetilde{R}_i+1}(\cdot))$$

$$\leq \max_{k \in [\overline{L}]}(\left\| \overline{\boldsymbol{W}}_k \right\|_{\text{F}} + E_i) \coloneqq \alpha.$$

By combining (18) and (19), we get

$$A_i \leq \max_{k \in [\overline{L}]} \left( \left\| \overline{\boldsymbol{W}}_k \right\|_{\text{F}} + E_i \right) \mathbb{E} \left\| \widehat{\mathbf{z}}_{i-1} - \overline{\mathbf{z}}_{i-1} \right\|_2 \leq \alpha \mathbb{E} \left\| \widehat{\mathbf{z}}_{i-1} - \overline{\mathbf{z}}_{i-1} \right\|_2. \tag{20}$$

**Computing $B_i$.** We proceed to compute the error associated with the $i$-th submodel, which we denote as $B_i$. It can be evaluated as follows:

$$B_i = \mathbb{E} \left\| f_i(\overline{\mathbf{z}}_{i-1}) - \bar{f}_i(\overline{\mathbf{z}}_{i-1}) \right\|_2$$

$$\overset{(15)}{=} \mathbb{E} \left\| \text{ReLU} \left( \left( \text{LR}_{RM \wedge \widetilde{R}_i}(\overline{\boldsymbol{W}}_i - \prod_{l \in P_i^{\text{u}}} \boldsymbol{W}_l) - (\overline{\boldsymbol{W}}_i - \prod_{l \in P_i^{\text{u}}} \boldsymbol{W}_l) \right) \overline{\mathbf{z}}_{i-1} \right) \right\|_2$$

$$\leq \mathbb{E} \left\| \left( \text{LR}_{RM \wedge \widetilde{R}_i}(\overline{\boldsymbol{W}}_i - \prod_{l \in P_i^{\text{u}}} \boldsymbol{W}_l) - (\overline{\boldsymbol{W}}_i - \prod_{l \in P_i^{\text{u}}} \boldsymbol{W}_l) \right) \overline{\mathbf{z}}_{i-1} \right\|_2 \qquad \text{(ReLU is 1-Lipschitz)}$$

$$\overset{(1)}{\leq} \left\| \text{LR}_{RM \wedge \widetilde{R}_i}(\overline{\boldsymbol{W}}_i - \prod_{l \in P_i^{\text{u}}} \boldsymbol{W}_l) - (\overline{\boldsymbol{W}}_i - \prod_{l \in P_i^{\text{u}}} \boldsymbol{W}_l) \right\|_2 \mathbb{E} \left\| \overline{\mathbf{z}}_{i-1} \right\|_2$$

$$= \sigma_{RM \wedge \widetilde{R}_i+1}(\overline{\boldsymbol{W}}_i - \prod_{l \in P_i^{\text{u}}} \boldsymbol{W}_l) \mathbb{E} \left\| \overline{\mathbf{z}}_{i-1} \right\|_2.$$

We can further simplify $\mathbb{E} \left\| \overline{\mathbf{z}}_{i-1} \right\|_2$ as :

$$\mathbb{E} \left\| \overline{\mathbf{z}}_{i-1} \right\|_2$$

$$= \mathbb{E} \left\| \text{ReLU}(\overline{\boldsymbol{W}}_{i-1} \overline{\mathbf{z}}_{i-2} + \bar{\boldsymbol{b}}_{i-1}) \right\|_2$$

$$= \mathbb{E} \left\| \overline{\boldsymbol{W}}_{i-1} \overline{\mathbf{z}}_{i-2} + \bar{\boldsymbol{b}}_{i-1} \right\|_2 \qquad \text{(ReLU is 1-Lipschitz)}$$

$$\leq \left\|\overline{\boldsymbol{W}}_{i-1}\right\|_{\mathrm{F}} \mathbb{E} \left\|\overline{\mathbf{z}}_{i-2}\right\|_2 + \left\|\overline{\boldsymbol{b}}_{i-1}\right\|_2 \qquad \text{(Applying triangle inequality and (1))}$$

$$\leq \left\|\overline{\boldsymbol{W}}_{i-1}\right\|_{\mathrm{F}} \left(\left\|\overline{\boldsymbol{W}}_{i-2}\right\|_{\mathrm{F}} \mathbb{E} \left\|\overline{\mathbf{z}}_{i-3}\right\|_2 + \left\|\overline{\boldsymbol{b}}_{i-2}\right\|_2\right) + \left\|\overline{\boldsymbol{b}}_{i-1}\right\|_2 \qquad \text{(Following the same steps)}$$

$$\leq \prod_{j=1}^{i-1} \left\|\overline{\boldsymbol{W}}_j\right\|_{\mathrm{F}} \mathbb{E} \left\|\mathbf{x}\right\|_2 + \sum_{j=1}^{i-1} \prod_{k=j+1}^{i-1} \left\|\overline{\boldsymbol{W}}_k\right\|_{\mathrm{F}} \left\|\overline{\boldsymbol{b}}_j\right\|_2 \qquad \text{(Repeating the same steps)}$$

$$= \sqrt{\|\Sigma\|_{\mathrm{F}}} \prod_{j=1}^{i-1} \left\|\overline{\boldsymbol{W}}_j\right\|_{\mathrm{F}} + \sum_{j=1}^{i-1} \prod_{k=j+1}^{i-1} \left\|\overline{\boldsymbol{W}}_k\right\|_{\mathrm{F}} \left\|\overline{\boldsymbol{b}}_j\right\|_2 \leq \beta.$$

Therefore, we obtain

$$B_i \leq \beta \sigma_{RM \wedge \widetilde{R}_i + 1}(\overline{\boldsymbol{W}}_i - \prod_{l \in P_i^{\mathrm{u}}} \boldsymbol{W}_l).$$

**Error Composition.** Having established upper bounds for $A_i$ and $B_i$, we next evaluate the expected error for the composition of the first $i$ adapted submodels.

$$\mathbb{E} \left\|\widehat{\mathbf{z}}_i - \overline{\mathbf{z}}_i\right\|_2 \overset{(17)}{\leq} A_i + B_i \overset{(20)}{\leq} \alpha \mathbb{E} \left\|\widehat{\mathbf{z}}_{i-1} - \overline{\mathbf{z}}_{i-1}\right\|_2 + B_i \leq \alpha(\alpha \mathbb{E} \left\|\widehat{\mathbf{z}}_{i-2} - \overline{\mathbf{z}}_{i-2}\right\|_2 + B_{i-1}) + B_i$$

$$= \alpha^2 \mathbb{E} \left\|\widehat{\mathbf{z}}_{i-2} - \overline{\mathbf{z}}_{i-2}\right\|_2 + \alpha B_{i-1} + B_i \leq \cdots \leq \alpha^{i-1} \mathbb{E} \left\|\widehat{\mathbf{z}}_1 - \overline{\mathbf{z}}_1\right\|_2 + \sum_{k=2}^{i} \alpha^{i-k} B_k. \qquad (21)$$

To compute the overall approximation error of $f$, which is the composite of all submodels, we have

$$\mathbb{E} \left\|f(\mathbf{x}) - \overline{f}(\mathbf{x})\right\|_2 = \mathbb{E} \left\|f(\mathbf{x}) - \overline{f}(\mathbf{x})\right\|_2 = \mathbb{E} \left\|\widehat{\mathbf{z}}_{\overline{L}} - \overline{\mathbf{z}}_{\overline{L}}\right\|_2$$

$$\overset{(21)}{\leq} \alpha^{\overline{L}-1} \mathbb{E} \left\|\widehat{\mathbf{z}}_1 - \overline{\mathbf{z}}_1\right\|_2 + \sum_{i=2}^{\overline{L}} \alpha^{\overline{L}-i} B_i$$

$$\overset{(16)}{\leq} \alpha^{\overline{L}-1} \beta \sigma_{RM \wedge \widetilde{R}_i + 1}(\overline{\boldsymbol{W}}_i - \prod_{l \in P_i^{\mathrm{u}}} \boldsymbol{W}_l) + \beta \sum_{i=2}^{\overline{L}} \alpha^{\overline{L}-i} \sigma_{RM \wedge \widetilde{R}_i + 1}(\overline{\boldsymbol{W}}_i - \prod_{l \in P_i^{\mathrm{u}}} \boldsymbol{W}_l)$$

$$= \beta \sum_{i=1}^{\overline{L}} \alpha^{\overline{L}-i} \sigma_{RM \wedge \widetilde{R}_i + 1}(\overline{\boldsymbol{W}}_i - \prod_{l \in P_i^{\mathrm{u}}} \boldsymbol{W}_l)$$

$$= \beta \sum_{i=1}^{\overline{L}} \alpha^{\overline{L}-i} \sigma_{RM + 1}(\overline{\boldsymbol{W}}_i - \prod_{l \in P_i^{\mathrm{u}}} \boldsymbol{W}_l).$$

Substituting $\alpha$ with $\max_{k \in [\overline{L}]}(\left\|\overline{\boldsymbol{W}}_k\right\|_{\mathrm{F}} + E_i)$ concludes the proof. $\qquad \square$

### E.3 APPROXIMATING MULTI-LAYER ReLU FNN VIA LoRA WITH GENERAL MODEL PARITION

Firstly, we provide the required non-singular assumption and the lemma demonstrating the mildness of this assumption for the general model partition cases after introducing necessary notations.

**Assumption 2.** *For the given LoRA-rank sequence $(R_l)_{l=1}^L$ and partition $\mathcal{P}$, the weight matrices of the frozen model $\boldsymbol{W}_1, \ldots, \boldsymbol{W}_L$ and $\left(\prod_{l \in P_i} \boldsymbol{W}_l\right) + \mathrm{LR}_r(\overline{\boldsymbol{W}}_i - \prod_{l=\min P_i}^{\max P_i - 1} \boldsymbol{W}_l)$ are non-singular for all $r \leq \sum_{l=\min P_i}^{\max P_i - 1} R_l$ and $i \in [\overline{L}]$.*

Note that $\max P_i$ and $\min P_i$ here represent the maximum and minimum elements in the set $P_i$, respectively.

**Lemma 11.** *Let $(\overline{\boldsymbol{W}}_l)_{l=1}^{\overline{L}}, (\boldsymbol{W}_l)_{l=1}^L \in \mathbb{R}^{D \times D}$ be matrices whose elements are drawn independently from arbitrary continuous distributions. Then, with probability 1, Assumption 2 holds for all $R \in [D]$.*

*Proof of Lemma 11.* Following the same steps in the proof of Lemma 3 but replacing the uniform partition with the general partition completes the proof. □

We now restate Theorem 6 and provide its proof.

**Theorem 6.** *Consider a partition $\mathcal{P}$ for the frozen model. Let Assumption 2 hold. If $\sum_{l \in P_i} R_l \geq \mathrm{rank}(\overline{\boldsymbol{W}}_i - \prod_{l \in P_i} \boldsymbol{W}_l)$ for all $i \in [\overline{L}]$, there exists LoRA adapters $(\Delta \boldsymbol{W}_l)_{l=1}^L$ with $\mathrm{rank}(\Delta \boldsymbol{W}_l) \leq R_l$ and biases $(\widehat{\boldsymbol{b}}_l)_{l=1}^L$ such that the adapted model $f$ can exactly approximate the target model.*

*Moreover, define the approximation error of the $i$-th layer as $E_i = \sigma_{\sum_{l \in P_i} R_l + 1}(\overline{\boldsymbol{W}}_i - \prod_{l \in P_i} \boldsymbol{W}_l)$, and the magnitude of the parameters and the input as $\beta := \max_{i \in [\overline{L}]} \left( \sqrt{\|\Sigma\|_F} \prod_{j=1}^i \|\overline{\boldsymbol{W}}_j\|_F + \sum_{j=1}^i \prod_{k=j+1}^{i-1} \|\overline{\boldsymbol{W}}_k\|_F \|\overline{\boldsymbol{b}}_j\|_2 \right) \vee \sqrt{\|\Sigma\|_F}$.*

*Then, there exists LoRA adapters $(\Delta \boldsymbol{W}_l)_{l=1}^L$ with $\mathrm{rank}(\Delta \boldsymbol{W}_l) \leq R_l$ and biases $(\widehat{\boldsymbol{b}}_l)_{l=1}^L$ such that for any input $\mathbf{x} \in \mathcal{X}$ with $\mathbb{E}\mathbf{x}\mathbf{x}^\top = \Sigma$, the approximation error can be bounded as*

$$\mathbb{E} \left\| f(\mathbf{x}) - \overline{f}(\mathbf{x}) \right\|_2 \leq \beta \sum_{i=1}^{\overline{L}} \max_{k \in [\overline{L}]} \left( \left\| \overline{\boldsymbol{W}}_k \right\|_F + E_k \right)^{\overline{L}-i} E_i.$$

*Proof of Theorem 6.* This proof follows the same steps as the proofs of Theorem 3 and Theorem 5, substituting the uniform partition $\mathcal{P}^u$ with the general partition $\mathcal{P}$ and applying Lemma 10 in place of Lemma 2 to derive the desired outcome. □

### E.4 APPROXIMATING MULTI-LAYER ReLU FNN VIA FINAL LAYERS TUNING

We now aim to examine another commonly used model adaptation method, the final layers tuning, within the same theoretical framework.

The main limitation of this method, as compared to LoRA, is that while LoRA can update all layers, the tuning of final layers keeps the initial layers frozen. Consequently, a clear limitation arises when the initial layers of the frozen model $f$ are less discriminative than the target model $\overline{f}$. That is, if there exist two input vectors $\boldsymbol{x}_1, \boldsymbol{x}_2 \in \mathbb{R}^{D \times D}$ such that the output of the initial layers of the frozen model $f_0$ is the same, but the output of the target model $\overline{f}$ is different, then no matter how the final layers are tuned, it is impossible for the adapted model $f$ to exactly approximate the target model $\overline{f}$.

To formalize this, we observe that for the first layer of the frozen model, the outputs of the inputs in the non-activation region are always zero. In other words, when $\boldsymbol{x}_1, \boldsymbol{x}_2 \in \{\boldsymbol{x} : \boldsymbol{W}_1 \boldsymbol{x} + \boldsymbol{b}_1 \leq \boldsymbol{0}\}$, we have $\mathrm{ReLU}(\boldsymbol{W}_1 \boldsymbol{x}_1 + \boldsymbol{b}_1) = \mathrm{ReLU}(\boldsymbol{W}_1 \boldsymbol{x}_2 + \boldsymbol{b}_1) = 0$. Therefore, no matter how the subsequent layers are tuned, we still have $f(\boldsymbol{x}_1) = f(\boldsymbol{x}_2)$. When we fix the first $l - 1$ layers, the non-activation region becomes $\{\boldsymbol{x} : \boldsymbol{W}_2(\boldsymbol{W}_1 \boldsymbol{x} + \boldsymbol{b}_1) + \boldsymbol{b}_2 \leq \boldsymbol{0}\}$. Similarly, we define the non-active region of the first $l$ layer in the frozen model as $I_l = \left\{ \boldsymbol{x} : \prod_{i=1}^l \boldsymbol{W}_i \boldsymbol{x} + \sum_{i=1}^l \prod_{j=i+1}^l \boldsymbol{W}_j \boldsymbol{b}_i \leq \boldsymbol{0} \right\}$. Correspondingly, we define $\overline{I}_l = \left\{ \boldsymbol{x} : \prod_{i=1}^l \overline{\boldsymbol{W}}_i \boldsymbol{x} + \sum_{i=1}^l \prod_{j=i+1}^l \overline{\boldsymbol{W}}_j \overline{\boldsymbol{b}}_i \leq \boldsymbol{0} \right\}$.

The following lemma is provided based on these definitions.

**Lemma 12.** *If $l \in [L-1]$ such that $I_l \setminus \bigcup_{i=1}^{\overline{L}} \overline{I}_i \neq \emptyset$ and the weight matrices of the target model $(\overline{\boldsymbol{W}}_i)_{i=1}^L$ are non-singular, then for any tuning of the last $L - l$ layers, $f \neq \overline{f}$.*

*Proof of Lemma 12.* For the simplicity of the presentation, we let $\overline{I} = \bigcup_{i=1}^{\overline{L}} \overline{I}_i$ to denote the non-activation region of the target model. Then, the condition $I_l \setminus \bigcup_{i=1}^{\overline{L}} \overline{I}_i \neq \emptyset$ can be written as $I_l \setminus \overline{I} \neq \emptyset$. Clearly, both $\overline{I}$ and $I_l$ are closed convex sets.

**Condition $I_l \setminus \overline{I} \neq \emptyset$.** The condition $I_l \setminus \overline{I} \neq \emptyset$ indicates that there exists a region in $I_l$ where the ReLUs are deactivated in the $l$-th layer of the frozen model, but activated in the entire target model. Therefore, for any $\boldsymbol{x}_1, \boldsymbol{x}_2 \in I_l \setminus \overline{I}$, we have $f(\boldsymbol{x}_1) = f(\boldsymbol{x}_2)$ regardless of how the final $l + 1$ layers

are tuned. If these $x_1, x_2 \in I_l \setminus \bar{I}$ satisfies $\bar{f}(x_1) \neq \bar{f}(x_2)$, this proof is completed. The remaining proof is showing the existence of such $x_1, x_2$.

**Existence of $x_1, x_2$.** Firstly, we show that there exists two $x_1, x_2 \in I_l \setminus \bar{I}$ such that $x_1 \neq x_2$. Let $x_1 \in I_l \setminus \bar{I}$. Since $I_l$ is a closed set, there exists a sequence $(z_i)_{i=1}^{\infty}$ where $z_i \in I_l$ and $z_i \neq x_1$ satisfying $\lim_{i \to \infty} z_i = x_1$. Note that at least one element $z_i$ must not belong to $\bar{I}$, otherwise $x_1$ would be in $\bar{I}$ due to the closed property of $\bar{I}$, contradicting the selection of $x_1$. Let $x_2 = z_i$. Therefore, we have two distinct $x_1, x_2 \in I_l \setminus \bar{I}$ with $x_1 \neq x_2$.

Then, given $x_1, x_2 \in I_l \setminus \bar{I}$ such that $x_1 \neq x_2$, both $x_1, x_2$ activate all the ReLUs in the target model. Since $x_1, x_2 \notin \bar{I}$ and the weight matrices of the target model $(\overline{W}_l)_{l=1}^{L}$ all are non-singular, we have

$$\bar{f}(x_1) - \bar{f}(x_2) = \overline{W}_{\bar{L}} \cdots \overline{W}_1(x_1 - x_2) \neq \mathbf{0},$$

implying $\bar{f}(x_1) \neq \bar{f}(x_2)$. Meanwhile, since $x_1, x_2 \in I_l$, the output of the initial $l$ layers of the frozen model are equal, thus we have $f(x_1) = f(x_2)$ no matter how we tune the last $L - l$ layers. This completes the proof. □

The following lemma reduces the assumptions to the assumption of randomly generated models. This assumption aligns with that of Corollary 4, thereby facilitating a more effective comparison between the expressive power of LoRA and the adaptation of the final layers.

**Lemma 4.** *Let $D \geq 2$ and $\bar{f}$ be a one-layer target FNN. Assume that the elements of weight matrices $(W_l)_{l=1}^{L}$ are independently drawn from arbitrary continuous distributions. With probability 1, for any tuning of the last $L - 1$ layers, $f \neq \bar{f}$.*

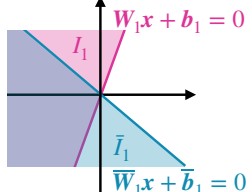

Figure 2: An example of $I_1$ and $\bar{I}_1$ when $D = 2$.

*Proof of Lemma 4.* If we can show that $I_1 \setminus \bar{I}_1 \neq \emptyset$, by Lemma 12, we obtain the desired results. Therefore, the remaining proof aims to show that $I_1 \setminus \bar{I}_1 \neq \emptyset$ with probability 1.

Note that $I_1 \setminus \bar{I}_1 = \emptyset$ holds only when $\overline{W}_1 = W_1$ (not that this is necessary condition not sufficient condition), as demonstrated in Figure 2. However, since the elements of matrices $W_1$ are independently drawn from arbitrary continuous distributions, we have $\mathbb{P}(W_1 \neq \overline{W}_1) = 1$ for all $l \in [L]$. Therefore, $I_1 \setminus \bar{I}_1 = \emptyset$ holds with probability 1. By Lemma 12, we complete the proof. □

## F  PROOFS FOR TFN APPROXIMATION

In this section, we not only provide the proof for the results outlined in Sec. 4, but also introduce the problem setting for TFNs with single-head attention layers and present the corresponding results.

### F.1  APPROXIMATING TRANSFORMER NETWORK WITH SINGLE-HEAD ATTENTION LAYERS

In this part, we outline the problem setting to investigate the expressive power of LoRA in TFNs that utilize single-head attention layers. The primary distinction between this setting and that of TFNs with multi-head attention layers lies in the weight matrices. Specifically, the $\overline{W}_{Ol}^{h}$ matrices for combining different attention heads are absent in this case. Despite this difference, the derived results are consistent, albeit under slightly modified assumptions regarding the weight matrices and a different LoRA adaptation strategy.

We start by introducing necessary notations. For an input matrix $X \in \mathbb{R}^{D \times N}$, where $D$ is the dimension of the token embeddings and $N$ is the number of tokens, the $l$-th Transformer block using single-head self-attention can be expressed as:

$$\mathtt{Attn}_l(Z_{l-1}) = W_{Vl}Z_{l-1} \cdot \mathrm{softmax}\left((W_{Kl}Z_{l-1})^{\top}W_{Ql}Z_{l-1}\right),$$
$$Z_l := W_{2l} \cdot \mathrm{ReLU}(W_{1l} \cdot \mathtt{Attn}_l(Z_{l-1}) + b_{1l}\mathbf{1}_N^{\top}) + b_{2l}\mathbf{1}_N^{\top},$$

where the weight matrices $\boldsymbol{W}_{Kl}, \boldsymbol{W}_{Ql}, \boldsymbol{W}_{Vl}, \boldsymbol{W}_{1l}, \boldsymbol{W}_{2l} \in \mathbb{R}^{D \times D}$, bias vectors $\boldsymbol{b}_{1l}, \boldsymbol{b}_{2l \in \mathbb{R}^D}$, $\boldsymbol{Z}_l$ is the output of $l$-th transformer block, with $\boldsymbol{Z}_0 = \boldsymbol{X}$. The output of the first $L$ Transformer blocks are subsequently fed into the output layer. This produces the final output of the TFN, given by $\text{softmax}(\boldsymbol{W}_o \boldsymbol{Z}_L)$, where $\boldsymbol{W}_o \in \mathbb{R}^{D \times D}$ represents the weight matrix of the output layer.

For single-head self-attention layers, the target model $\bar{f}$, frozen model $f$, and the adapted model $f$ can be formally represented as:

$$\text{Target TFN} \quad g = \text{TFN}_{L,D}\left(\cdot; \left((\overline{\boldsymbol{W}}_{Vl}, \overline{\boldsymbol{W}}_{Kl}, \overline{\boldsymbol{W}}_{Ql}, \overline{\boldsymbol{W}}_{2l}, \overline{\boldsymbol{W}}_{1l})_{l=1}^L, \overline{\boldsymbol{W}}_o\right), (\bar{\boldsymbol{b}}_{1l}, \bar{\boldsymbol{b}}_{2l})_{l=1}^L\right),$$

$$\text{Frozen TFN} \quad f_0 = \text{TFN}_{L,D}\left(\cdot; \left((\boldsymbol{W}_{Vl}, \boldsymbol{W}_{Kl}, \boldsymbol{W}_{Ql}, \boldsymbol{W}_{2l}, \boldsymbol{W}_{1l})_{l=1}^L, \boldsymbol{W}_o\right), (\boldsymbol{b}_{1l}, \boldsymbol{b}_{2l})_{l=1}^L\right),$$

$$\text{Adapted TFN} \quad f = \text{TFN}_{L,D}\Big(\cdot; \big((\boldsymbol{W}_{Vl} + \Delta \boldsymbol{W}_{Vl}, \boldsymbol{W}_{Kl} + \Delta \boldsymbol{W}_{Kl}, \boldsymbol{W}_{Ql} + \Delta \boldsymbol{W}_{Ql},$$

$$\boldsymbol{W}_{2l} + \Delta \boldsymbol{W}_{2l}, \boldsymbol{W}_{1l} + \Delta \boldsymbol{W}_{1l})_{l=1}^L, \boldsymbol{W}_o + \Delta \boldsymbol{W}_o), (\widehat{\boldsymbol{b}}_{1l}, \widehat{\boldsymbol{b}}_{2l})_{l=1}^L\Big).$$

Here, $\overline{\boldsymbol{W}}_{Kl}, \overline{\boldsymbol{W}}_{Ql}, \overline{\boldsymbol{W}}_{Vl}$ are the weight matrices for generating key, query, and values in the $l$-th transformer block of the target TFN; $\overline{\boldsymbol{W}}_{1l}, \overline{\boldsymbol{W}}_{2l}$ and $\bar{\boldsymbol{b}}_{1l}, \bar{\boldsymbol{b}}_{2l}$ serve as the weight matrices and bias vectors, respectively, for the feedforward layer in the same block; $\overline{\boldsymbol{W}}_o$ is the weight matrix for the output layer. For the frozen TFN, the same roles are played by $\boldsymbol{W}_{Kl}, \boldsymbol{W}_{Ql}, \boldsymbol{W}_{Vl}$, $\boldsymbol{W}_{1l}, \boldsymbol{W}_{2l}$, and $\boldsymbol{b}_{1l}, \boldsymbol{b}_{2l}$ for all $l \in [L]$ and $\boldsymbol{W}_o$. For the adapted model, low-rank adapters $\Delta \boldsymbol{W}_{Kl}, \Delta \boldsymbol{W}_{Ql}, \Delta \boldsymbol{W}_{Vl}, \Delta \boldsymbol{W}_{1l}, \Delta \boldsymbol{W}_{2l}, \Delta \boldsymbol{W}_o$ with a rank constraint $R \in [D]$ are added to each weight matrix, and the bias vectors are updated to $\widehat{\boldsymbol{b}}_{1l}, \widehat{\boldsymbol{b}}_{2l}$ for all $l \in [L]$.

Given the problem setting outlined above, we give the non-singularity assumption for TFNs with single-head attention layers.

**Assumption 3** (Non-Singularity). *All the weight matrices of both the target model and the frozen model, as well as the following matrices for all $r \in [D]$,*

$$\boldsymbol{W}_{Kl}^\top \boldsymbol{W}_{Ql} + \text{LR}_r\left(\overline{\boldsymbol{W}}_{Kl}^\top \overline{\boldsymbol{W}}_{Ql} - \boldsymbol{W}_{Kl}^\top \boldsymbol{W}_{Ql}\right), \text{ where } l = 1,$$

$$\boldsymbol{W}_{Kl}\boldsymbol{W}_{Ql} + \text{LR}_r\left(\boldsymbol{W}_{2,l-1}^{-1\top}\overline{\boldsymbol{W}}_{2,l-1}^\top \overline{\boldsymbol{W}}_{Kl}^\top \overline{\boldsymbol{W}}_{Ql}\overline{\boldsymbol{W}}_{2,l-1}\boldsymbol{W}_{2,l-1}^{-1} - \boldsymbol{W}_{Kl}\boldsymbol{W}_{Ql}\right), \text{ for } l \in [L] \setminus \{1\},$$

$$\boldsymbol{W}_{1l}\boldsymbol{W}_{Vl} + \text{LR}_r\left(\overline{\boldsymbol{W}}_{1l}\overline{\boldsymbol{W}}_{Vl} - \boldsymbol{W}_{1l}\boldsymbol{W}_{Vl}\right), \text{ for } l = 1,$$

$$\boldsymbol{W}_{1l}\boldsymbol{W}_{Vl} + \text{LR}_r\left(\overline{\boldsymbol{W}}_{1l}\overline{\boldsymbol{W}}_{Vl}\overline{\boldsymbol{W}}_{2,l-1}\boldsymbol{W}_{2,l-1}^{-1} - \boldsymbol{W}_{1l}\boldsymbol{W}_{Vl}\right), \text{ for all } l \in [L] \setminus \{1\},$$

$$\boldsymbol{W}_o\boldsymbol{W}_{2L} + \text{LR}_r(\overline{\boldsymbol{W}}_o\overline{\boldsymbol{W}}_{2L} - \boldsymbol{W}_o\boldsymbol{W}_{2L}),$$

*are non-singular.*

**Lemma 13.** *Let the elements of all weight matrices in target model $\bar{f}$ and the frozen model $f$ be independently sampled from continuous distributions. Then, Assumption 3 holds with probability 1.*

*Proof of Lemma 13.* The results can be obtained by replicating the same steps outlined in the proof of Lemma 3. □

**Theorem 8.** *Consider the rank of the adapter weight matrices $R \in [D]$. Let Assumption 3 hold. Define the rank-based functionality gap $G_i$ to $i$-th transformer block ($i \in [L]$) or output layer ($i = L + 1$) as*

$$G_i = \begin{cases} \max_h \left(\text{rank}(\overline{\boldsymbol{W}}_{Ki}^{h\top}\overline{\boldsymbol{W}}_{Qi}^h - \boldsymbol{W}_{Ki}^{h\top}\boldsymbol{W}_{Qi}^h)\right) \vee \max_h \left(\text{rank}(\overline{\boldsymbol{W}}_{1i}\overline{\boldsymbol{W}}_{Vi}^h - \boldsymbol{W}_{1i}\boldsymbol{W}_{Vi}^h)\right), i = 1, \\ \max_h \left(\text{rank}(\overline{\boldsymbol{W}}_{2,i-1}^\top \overline{\boldsymbol{W}}_{Ki}^{h\top}\overline{\boldsymbol{W}}_{Qi}^h \overline{\boldsymbol{W}}_{2,i-1} - \boldsymbol{W}_{2,i-1}^\top \boldsymbol{W}_{Ki}^{h\top}\boldsymbol{W}_{Qi}^h \boldsymbol{W}_{2,i-1})\right) \\ \qquad \vee \max_h \left(\text{rank}(\overline{\boldsymbol{W}}_{1i}\overline{\boldsymbol{W}}_{Vi}^h \overline{\boldsymbol{W}}_{2,i-1} - \boldsymbol{W}_{1i}\boldsymbol{W}_{Vi}^h \boldsymbol{W}_{2,i-1})\right), \quad 2 \leq i \leq L, \\ \text{rank}(\overline{\boldsymbol{W}}_o \overline{\boldsymbol{W}}_{2L} - \boldsymbol{W}_o\boldsymbol{W}_{2L}), \qquad\qquad\qquad\qquad\qquad i = L + 1. \end{cases}$$

*If $R \geq \max_{i \in [L+1]} \lceil \frac{G_i}{2} \rceil$, there exists rank-$R$ or lower weight matrices for low-rank adapters $(\Delta \boldsymbol{W}_{Kl}, \Delta \boldsymbol{W}_{Ql}, \Delta \boldsymbol{W}_{Vl}, \Delta \boldsymbol{W}_{1l})_{l=1}^L, \Delta \boldsymbol{W}_{2L}, \Delta \boldsymbol{W}_o$ with other low-rank adapters set to $\boldsymbol{O}$, and updated bias vectors: $(\widehat{\boldsymbol{b}}_{1l}, \widehat{\boldsymbol{b}}_{2l})_{l=1}^L$, such that for any $\boldsymbol{X} \in \mathbb{R}^{D \times N}$, the adapted model $f$ exactly approximates $\bar{f}$, i.e., $f(\boldsymbol{X}) = \bar{f}(\boldsymbol{X})$, with probability 1.*

*Proof of Theorem 8.* Let $\overline{\boldsymbol{H}}_l \in \mathbb{R}^{D \times N}$ and $\overline{\boldsymbol{Z}}_l \in \mathbb{R}^{D \times N}$ denote the intermediate and final outputs of the $l$-th transformer block in the target model $\bar{f}$, respectively. Specifically, $\overline{\boldsymbol{H}}_l$ represents the output from the first feedforward layer in the $l$-th transformer block. They are defined as

$$\overline{\boldsymbol{H}}_l = \text{ReLU}\left(\overline{\boldsymbol{W}}_{1l}\overline{\boldsymbol{W}}_{Vl}\overline{\boldsymbol{Z}}_{l-1} \cdot \text{softmax}\left(\overline{\boldsymbol{Z}}_{l-1}^{\top}\overline{\boldsymbol{W}}_{Kl}^{\top}\overline{\boldsymbol{W}}_{Ql}\overline{\boldsymbol{Z}}_{l-1}\right) + \bar{\boldsymbol{b}}_{1l}\mathbf{1}_N^{\top}\right),$$

$$\overline{\boldsymbol{Z}}_l = \overline{\boldsymbol{W}}_{2l}\overline{\boldsymbol{H}}_l + \bar{\boldsymbol{b}}_{2l}\mathbf{1}_N^{\top},$$

where $l \in [L]$. For the adapted model $f$, we introduce $\widehat{\boldsymbol{H}}_l$ and $\widehat{\boldsymbol{Z}}_l$ to denote the corresponding intermediate output of the first feedforward layer and the final output of the $l$-th transformer block for the adapted model, respectively:

$$\widehat{\boldsymbol{H}}_l = \text{ReLU}\Bigg((\boldsymbol{W}_{1l} + \Delta\boldsymbol{W}_{1l})(\boldsymbol{W}_{Vl} + \Delta\boldsymbol{W}_{Vl}) \cdot \widehat{\boldsymbol{Z}}_{l-1}$$

$$\cdot \text{softmax}\left(\widehat{\boldsymbol{Z}}_{l-1}^{\top}(\boldsymbol{W}_{Kl} + \Delta\boldsymbol{W}_{Kl})^{\top}(\boldsymbol{W}_{Ql} + \Delta\boldsymbol{W}_{Ql})\widehat{\boldsymbol{Z}}_{l-1}\right) + \widehat{\boldsymbol{b}}_{1l}\mathbf{1}_N^{\top}\Bigg),$$

$$\widehat{\boldsymbol{Z}}_l = (\boldsymbol{W}_{2l} + \Delta\boldsymbol{W}_{2l})\widehat{\boldsymbol{H}}_l + \widehat{\boldsymbol{b}}_{2l}\mathbf{1}_N^{\top},$$

where $l \in [L]$. We note that $\overline{\boldsymbol{Z}}_0 = \widehat{\boldsymbol{Z}}_0 = \boldsymbol{X}$.

In this proof, we set $\Delta\boldsymbol{W}_{2l} = \boldsymbol{O}$ for all $l \in [L]$. Our goal is to show that adding low-rank adapters to self-attention layers and the first feedforward layers in all transformer blocks enables the adapted model $f$ to be functionally equivalent to the target model $\bar{f}$ of the same dimensions. We start by inductively constructing the adapter weight matrices $(\Delta\boldsymbol{W}_{1l}, \Delta\boldsymbol{W}_{Vl}, \Delta\boldsymbol{W}_{Kl}, \Delta\boldsymbol{W}_{Ql}, \widehat{\boldsymbol{b}}_{1l}, \widehat{\boldsymbol{b}}_{2l})_{l=1}^{L}$ such that $\widehat{\boldsymbol{H}}_l = \overline{\boldsymbol{H}}_l$ for all $l \in [L]$. We then select the low-rank adapters for $\boldsymbol{W}_{2L}$ and the $\boldsymbol{W}_o$ to approximate the output of the target model. For unmentioned low-rank adapters, we set them as $\boldsymbol{O}$.

**When $l = 1$.** To achieve $\widehat{\boldsymbol{H}}_l$ with $\overline{\boldsymbol{H}}_l$ for all $\boldsymbol{X}$, the following conditions must be satisfied:

Bias Vector: $\widehat{\boldsymbol{b}}_{1l} = \bar{\boldsymbol{b}}_{1l}$,

Query and Key: $(\boldsymbol{W}_{Kl} + \Delta\boldsymbol{W}_{Kl})^{\top}(\boldsymbol{W}_{Ql} + \Delta\boldsymbol{W}_{Ql}) = \overline{\boldsymbol{W}}_{Kl}^{\top}\overline{\boldsymbol{W}}_{Ql}$

Value and First Feedforward Layer: $(\boldsymbol{W}_{1l} + \Delta\boldsymbol{W}_{1l})(\boldsymbol{W}_{Vl} + \Delta\boldsymbol{W}_{Vl}) = \overline{\boldsymbol{W}}_{1l}\overline{\boldsymbol{W}}_{Vl}$.

To achieve this, we set $\widehat{\boldsymbol{b}}_{1l} = \bar{\boldsymbol{b}}_{1l}$ to achieve (24), and select rank-$R$ or lower matrices $\Delta\boldsymbol{W}_{Kl}, \Delta\boldsymbol{W}_{Ql}, \Delta\boldsymbol{W}_{1l}, \Delta\boldsymbol{W}_{Vl}$ as suggested by Lemma 7. This ensures $\widehat{\boldsymbol{H}}_l = \overline{\boldsymbol{H}}_l$ for $l = 1$.

**When $l > 1$.** Now we focus on the cases where $l = 2, \ldots, L$. Assume the induction hypothesis holds for $l - 1$, which is $\widehat{\boldsymbol{H}}_{l-1} = \overline{\boldsymbol{H}}_{l-1}$. This implies

$$\overline{\boldsymbol{H}}_{l-1} = \overline{\boldsymbol{W}}_{2,l-1}^{-1}(\overline{\boldsymbol{Z}}_{l-1} - \bar{\boldsymbol{b}}_{2,l-1}\mathbf{1}_N^{\top}) = \boldsymbol{W}_{2,l-1}^{-1}(\widehat{\boldsymbol{Z}}_{l-1} - \widehat{\boldsymbol{b}}_{2,l-1}\mathbf{1}_N^{\top}) = \widehat{\boldsymbol{H}}_{l-1}.$$

Using this assumption, we express $\widehat{\boldsymbol{Z}}_{l-1}$ in terms of $\overline{\boldsymbol{Z}}_{l-1}$:

$$\widehat{\boldsymbol{Z}}_{l-1} = \boldsymbol{W}_{2,l-1}\overline{\boldsymbol{W}}_{2,l-1}^{-1}(\overline{\boldsymbol{Z}}_{l-1} - \bar{\boldsymbol{b}}_{2,l-1}\mathbf{1}_N^{\top}) + \widehat{\boldsymbol{b}}_{2,l-1}\mathbf{1}_N^{\top}.$$

Let $\widehat{\boldsymbol{b}}_{2,l-1} = \boldsymbol{W}_{2,l-1}\overline{\boldsymbol{W}}_{2,l-1}^{-1}\bar{\boldsymbol{b}}_{2,l-1}$, then we have

$$\widehat{\boldsymbol{Z}}_{l-1} = \boldsymbol{W}_{2,l-1}\overline{\boldsymbol{W}}_{2,l-1}^{-1}\overline{\boldsymbol{Z}}_{l-1}. \tag{22}$$

To achieve $\widehat{\boldsymbol{H}}_l = \overline{\boldsymbol{H}}_l$, we express both $\widehat{\boldsymbol{H}}_l$ and $\overline{\boldsymbol{H}}_l$ in terms of $\overline{\boldsymbol{Z}}_{l-1}$:

$$\overline{\boldsymbol{H}}_l = \text{ReLU}\left(\overline{\boldsymbol{W}}_{1l}\overline{\boldsymbol{W}}_{Vl} \cdot \overline{\boldsymbol{Z}}_{l-1} \cdot \text{softmax}\left(\overline{\boldsymbol{Z}}_{l-1}^{\top}\overline{\boldsymbol{W}}_{Kl}^{\top}\overline{\boldsymbol{W}}_{Ql}\overline{\boldsymbol{Z}}_{l-1}\right) + \bar{\boldsymbol{b}}_{1l}\mathbf{1}_N^{\top}\right)$$

$$\widehat{\boldsymbol{H}}_l = \text{ReLU}\left((\boldsymbol{W}_{1l} + \Delta\boldsymbol{W}_{1l})(\boldsymbol{W}_{Vl} + \Delta\boldsymbol{W}_{Vl}) \cdot \widehat{\boldsymbol{Z}}_{l-1}\right.$$

$$\cdot \operatorname{softmax}\left(\widehat{\boldsymbol{Z}}_{l-1}^{\top}(\boldsymbol{W}_{Kl} + \Delta \boldsymbol{W}_{Kl})^{\top}(\boldsymbol{W}_{Ql} + \Delta \boldsymbol{W}_{Ql})\widehat{\boldsymbol{Z}}_{l-1}\right) + \widehat{\boldsymbol{b}}_{1l}\mathbf{1}_{N}^{\top}\right),$$

$$\stackrel{(22)}{=} \operatorname{ReLU}\Bigg((\boldsymbol{W}_{1l} + \Delta \boldsymbol{W}_{1l})(\boldsymbol{W}_{Vl} + \Delta \boldsymbol{W}_{Vl}) \cdot \boldsymbol{W}_{2,l-1}\overline{\boldsymbol{W}}_{2,l-1}^{-1}\overline{\boldsymbol{Z}}_{l-1}$$

$$\cdot \operatorname{softmax}\Big(\overline{\boldsymbol{Z}}_{l-1}^{\top}\overline{\boldsymbol{W}}_{2,l-1}^{-1\top}\boldsymbol{W}_{2,l-1}^{\top}(\boldsymbol{W}_{Kl} + \Delta \boldsymbol{W}_{Kl})^{\top}$$

$$(\boldsymbol{W}_{Ql} + \Delta \boldsymbol{W}_{Ql})\boldsymbol{W}_{2,l-1}\overline{\boldsymbol{W}}_{2,l-1}^{-1}\overline{\boldsymbol{Z}}_{l-1}\Big) + \widehat{\boldsymbol{b}}_{1l}\mathbf{1}_{N}^{\top}\Bigg).$$

Therefore, we need to align the following three components:

Bias Vector: $\widehat{\boldsymbol{b}}_{1l} = \bar{\boldsymbol{b}}_{1l}$,

Query and Key: $(\boldsymbol{W}_{Kl} + \Delta \boldsymbol{W}_{Kl})^{\top}(\boldsymbol{W}_{Ql} + \Delta \boldsymbol{W}_{Ql}) = \boldsymbol{W}_{2,l-1}^{-1\top}\overline{\boldsymbol{W}}_{2,l-1}^{\top}\overline{\boldsymbol{W}}_{Kl}^{\top}\overline{\boldsymbol{W}}_{Ql}\overline{\boldsymbol{W}}_{2,l-1}\boldsymbol{W}_{2,l-1}^{-1}$,

Value and First Feedforward Layer: $(\boldsymbol{W}_{1l} + \Delta \boldsymbol{W}_{1l})(\boldsymbol{W}_{Vl} + \Delta \boldsymbol{W}_{Vl}) = \overline{\boldsymbol{W}}_{1l}\overline{\boldsymbol{W}}_{Vl}\overline{\boldsymbol{W}}_{2,l-1}\boldsymbol{W}_{2,l-1}^{-1}$.

By setting $\widehat{\boldsymbol{b}}_{1l}$ based on (26) and adjusting $\Delta \boldsymbol{W}_{Kl}, \Delta \boldsymbol{W}_{Ql}, \Delta \boldsymbol{W}_{1l}, \Delta \boldsymbol{W}_{Vl}$ based on Lemma 7, we satisfy all three conditions above, thereby obtaining $\widehat{\boldsymbol{H}}_{l} = \overline{\boldsymbol{H}}_{l}$ for $l \in [L] \setminus \{1\}$.

**Output Layer Analysis.** By the induction method, we have established $\widehat{\boldsymbol{H}}_{l} = \overline{\boldsymbol{H}}_{l}$ for all $l \in [L]$. We will complete the proof by showing that $\bar{f}(\boldsymbol{X}) = f(\boldsymbol{X})$ for all $\boldsymbol{X} \in \mathcal{X}$.

The final output distribution of the target TFN $\bar{f}$ can be written as

$$\bar{f}(\boldsymbol{X}) = \operatorname{softmax}(\overline{\boldsymbol{W}}_{o}\overline{\boldsymbol{Z}}_{L}) = \operatorname{softmax}\left(\overline{\boldsymbol{W}}_{o}\left(\overline{\boldsymbol{W}}_{2L}\overline{\boldsymbol{H}}_{L} + \bar{\boldsymbol{b}}_{2L}\mathbf{1}_{N}^{\top}\right)\right).$$

We can similarly formulate the final output distribution of the adapted model $f$:

$$f(\boldsymbol{X}) = \operatorname{softmax}((\boldsymbol{W}_{o} + \Delta \boldsymbol{W}_{o})\widehat{\boldsymbol{Z}}_{L})$$

$$= \operatorname{softmax}\left((\boldsymbol{W}_{o} + \Delta \boldsymbol{W}_{o})\left((\boldsymbol{W}_{2L} + \Delta \boldsymbol{W}_{2L})\widehat{\boldsymbol{H}}_{L} + \widehat{\boldsymbol{b}}_{2L}\mathbf{1}_{N}^{\top}\right)\right),$$

To align these two expressions, we select $\Delta \boldsymbol{W}_{2L}$ and $\Delta \boldsymbol{W}_{o}$ based on Lemma 7, and let $\widehat{\boldsymbol{b}}_{2L} = (\boldsymbol{W}_{o} + \Delta \boldsymbol{W}_{o})^{-1}\overline{\boldsymbol{W}}_{o}\bar{\boldsymbol{b}}_{2L}$, where $\boldsymbol{W}_{o} + \Delta \boldsymbol{W}_{o}$ is invertible as shown in the proof of Lemma 7. Thus, the proof is complete. $\square$

The following corollary identifies the specific LoRA-rank required to achieve exact representation for random model cases in the current setting.

**Corollary 9.** *Assume that the elements of all the weight matrices of both the target TFN and the frozen TFN are independently drawn from arbitrary continuous distributions. If $R \geq \lceil \frac{D}{2} \rceil$, adding low-rank adapters of rank at most $R$ to weight matrices in $(\Delta \boldsymbol{W}_{Kl}, \Delta \boldsymbol{W}_{Ql}, \Delta \boldsymbol{W}_{Vl}, \Delta \boldsymbol{W}_{1l})_{l=1}^{L}, \Delta \boldsymbol{W}_{2L}, \Delta \boldsymbol{W}_{o}$ and tuning the bias vectors, enables the adapted model $f$ to exactly approximate the target model $\bar{f}$, i.e., $f(\boldsymbol{X}) = \bar{f}(\boldsymbol{X})$ for all $\boldsymbol{X} \in \mathbb{R}^{D \times N}$.*

*Proof of Corollary 9.* By combining Lemma 13 and Theorem 8, and following the same steps in the proof of Corollary 4 which yields $\max_{i} G_{i} = D$, we can obtain the desired outcome. $\square$

### F.2 APPROXIMATING TRANSFORMER NETWORK WITH MULTI-HEAD ATTENTION LAYERS

In this section, we first provide the explicit formulation of TFN with multi-head attention layers.

Consider an input matrix $\boldsymbol{X} \in \mathbb{R}^{D \times N}$, where $D$ is the dimension of the token embeddings and $N$ is the number of tokens. The output of the $l$-th transformer block is denoted as $\boldsymbol{Z}_{l}$, which can be computed as follows:

$$\operatorname{Attn}_{l}(\boldsymbol{Z}_{l-1}) := \sum_{h=1}^{H} \boldsymbol{W}_{Ol}^{h}\boldsymbol{W}_{Vl}^{h}\boldsymbol{Z}_{l-1} \cdot \operatorname{softmax}\left((\boldsymbol{W}_{Kl}^{h}\boldsymbol{Z}_{l-1})^{\top}\boldsymbol{W}_{Ql}^{h}\boldsymbol{Z}_{l-1}\right),$$

$$\boldsymbol{Z}_{l} := \boldsymbol{W}_{2l} \cdot \operatorname{ReLU}(\boldsymbol{W}_{1l} \cdot \operatorname{Attn}_{l}(\boldsymbol{Z}_{l-1}) + \boldsymbol{b}_{1l}\mathbf{1}_{N}^{\top}) + \boldsymbol{b}_{2l}\mathbf{1}_{N}^{\top},$$

where we define $\boldsymbol{Z}_0 = \boldsymbol{X}$. Here, $H$ is the number of attention heads. The weight matrices for each head $h \in [H]$ in the $l$-th transformer block are $\boldsymbol{W}_{Ol}^h, \boldsymbol{W}_{Vl}^h, \boldsymbol{W}_{Kl}^h, \boldsymbol{W}_{Ql}^h \in \mathbb{R}^{D \times D}$. The softmax operator $\mathrm{softmax}(\cdot)$ is applied column-wise to the matrix. Further, $\boldsymbol{W}_{2l}, \boldsymbol{W}_{1l} \in \mathbb{R}^{D \times D}$ are the weight matrices and $\boldsymbol{b}_{1l}, \boldsymbol{b}_{2l} \in \mathbb{R}^D$ are the bias vectors in the feedforward layers.

A Transformer network, denoted as $\mathrm{TFN}_{L,D}$, is a composition of $L$ Transformer blocks, followed by an softmax output layer $\mathrm{softmax}(\boldsymbol{W}_o \cdot)$, where $\boldsymbol{W}_o \in \mathbb{R}^{D \times D}$. The final output of the TFN is given by $\mathrm{softmax}(\boldsymbol{W}_o \boldsymbol{Z}_L)$. To study the expressive power of LoRA within TFNs featuring multi-head attention layers, we next specify the parameters of the target model $\bar{f}$, frozen model $f_0$, and the adapted model $f$, each with $L$ transformer blocks and a dimension $D$.

To study the expressive power of LoRA within TFNs featuring multi-head attention layers, we next specify the parameters of the target model $\bar{f}$, frozen model $f_0$, and the adapted model $f$, each with $L$ transformer blocks and a dimension $D$. For ease of presentation, we drop the subscript in $\mathrm{TFN}_{L,D}$, referring to it simply as TFN. Given a specified rank $R \in [D]$ for LoRA, these models are defined as follows:

$$\text{Target TFN } \bar{f} = \mathrm{TFN}\left(\cdot; \left(((\overline{\boldsymbol{W}}_{Ol}^h, \overline{\boldsymbol{W}}_{Vl}^h, \overline{\boldsymbol{W}}_{Kl}^h, \overline{\boldsymbol{W}}_{Ql}^h)_{h=1}^H, \overline{\boldsymbol{W}}_{2l}, \overline{\boldsymbol{W}}_{1l})_{l=1}^L, \overline{\boldsymbol{W}}_o\right), (\bar{\boldsymbol{b}}_{1l}, \bar{\boldsymbol{b}}_{2l})_{l=1}^L\right),$$

$$\text{Frozen TFN } f_0 = \mathrm{TFN}\left(\cdot; (((\boldsymbol{W}_{Ol}^h, \boldsymbol{W}_{Vl}^h, \boldsymbol{W}_{Kl}^h, \boldsymbol{W}_{Ql}^h)_{h=1}^H, \boldsymbol{W}_{2l}, \boldsymbol{W}_{1l})_{l=1}^L, \boldsymbol{W}_o), (\boldsymbol{b}_{1l}, \boldsymbol{b}_{2l})_{l=1}^L\right),$$

$$\text{Adapted TFN } f = \mathrm{TFN}\Big(\cdot; (((\boldsymbol{W}_{Ol}^h + \Delta\boldsymbol{W}_{Ol}^h, \boldsymbol{W}_{Vl}^h + \Delta\boldsymbol{W}_{Vl}^h, \boldsymbol{W}_{Kl}^h + \Delta\boldsymbol{W}_{Kl}^h, \boldsymbol{W}_{Ql}^h + \Delta\boldsymbol{W}_{Ql}^h)_{h=1}^H,$$

$$\boldsymbol{W}_{2l} + \Delta\boldsymbol{W}_{2l}, \boldsymbol{W}_{1l} + \Delta\boldsymbol{W}_{1l})_{l=1}^L, \boldsymbol{W}_o + \Delta\boldsymbol{W}_o), (\widehat{\boldsymbol{b}}_{1l}, \widehat{\boldsymbol{b}}_{2l})_{l=1}^L\Big),$$

where the weight matrices $\in \mathbb{R}^{D \times D}$, and the bias vectors $\in \mathbb{R}^D$. Moreover, the weight matrices of the low-rank adapters $\Delta\boldsymbol{W}_{Ol}^h, \Delta\boldsymbol{W}_{Vl}^h, \Delta\boldsymbol{W}_{Kl}^h, \Delta\boldsymbol{W}_{Ql}^h, \Delta\boldsymbol{W}_{2l}, \Delta\boldsymbol{W}_{1l}$ for all $h \in [H]$ and $l \in [L]$ are of rank $R$ or lower.

We next introduce non-singularity Assumption 4 for TFN with multi-head attention layers scenarios, which is then validated by Lemma 14. We then provide proof of our main results for TFNs — Theorem 7. Additionally, we introduce a supplementary theorem that amalgamates results for TFNs with both single-head and multi-head attention layers when the weight matrices are randomly initialized. This is articulated in Corollary 10.

**Assumption 4** (Non-Singularity). *For a fixed $R \in [D]$, all the weight matrices of both the target model and the frozen model and the following matrices for all $r \in [R]$,*

$$\boldsymbol{W}_{Kl}^{h\top} \boldsymbol{W}_{Ql}^h + \mathrm{LR}_r\left(\overline{\boldsymbol{W}}_{Kl}^{h\top} \overline{\boldsymbol{W}}_{Ql}^h - \boldsymbol{W}_{Kl}^{h\top} \boldsymbol{W}_{Ql}^h\right), \text{ for all } h \in [H] \text{ and } l = 1,$$

$$\boldsymbol{W}_{Kl}^{h\top} \boldsymbol{W}_{Ql}^h + \mathrm{LR}_r\left(\boldsymbol{W}_{2,l-1}^{-1\top} \overline{\boldsymbol{W}}_{2,l-1}^\top \overline{\boldsymbol{W}}_{Kl}^{h\top} \overline{\boldsymbol{W}}_{Ql}^h \overline{\boldsymbol{W}}_{2,l-1} \boldsymbol{W}_{2,l-1}^{-1} - \boldsymbol{W}_{Kl}^{h\top} \boldsymbol{W}_{Ql}^h\right), \text{ for all } h \in [H] \text{ and } l \in [L] \setminus \{1\},$$

$$\boldsymbol{W}_{Ol}^h \boldsymbol{W}_{Vl}^h + \mathrm{LR}_r\left(\boldsymbol{W}_{1l}^{-1} \overline{\boldsymbol{W}}_{1l} \overline{\boldsymbol{W}}_{Ol}^h \overline{\boldsymbol{W}}_{Vl}^h - \boldsymbol{W}_{Ol}^h \boldsymbol{W}_{Vl}^h\right), \text{ for all } h \in [H] \text{ and } l = 1,$$

$$\boldsymbol{W}_{Ol}^h \boldsymbol{W}_{Vl}^h + \mathrm{LR}_r\left(\boldsymbol{W}_{1l}^{-1} \overline{\boldsymbol{W}}_{1l} \overline{\boldsymbol{W}}_{Ol}^h \overline{\boldsymbol{W}}_{Vl}^h \overline{\boldsymbol{W}}_{2,l-1} \boldsymbol{W}_{2,l-1}^{-1} - \boldsymbol{W}_{Ol}^h \boldsymbol{W}_{Vl}^h\right), \text{ for all } h \in [H] \text{ and } l \in [L] \setminus \{1\},$$

$$\boldsymbol{W}_o \boldsymbol{W}_{2L} + \mathrm{LR}_r(\overline{\boldsymbol{W}}_o \overline{\boldsymbol{W}}_{2L} - \boldsymbol{W}_o \boldsymbol{W}_{2L}),$$

*are non-singular.*

**Lemma 14.** *Let the elements of all weight matrices in the target model $\bar{f}$ and frozen model $f_0$ be independently sampled from continuous distributions. Then, Assumption 4 holds with probability 1.*

*Proof of Lemma 14.* The results can be obtained by replicating the same steps outlined in the proof of Lemma 3. $\square$

For the reader's reference, we restate Theorem 7 here integrated with the explicit formulation of the rank-based functionality gap $G_i$.

**Theorem 7.** *Consider a given LoRA-rank $R \in [D]$. Let Assumption 4 hold. Define the rank-based functionality gap $G_i$ to $i$-th transformer block ($i \in [L]$) or output layer ($i = L + 1$) as*

$$G_i = \begin{cases} \max_h \left( \text{rank}(\overline{\boldsymbol{W}}_{Ki}^{h\top}\overline{\boldsymbol{W}}_{Qi}^h - \boldsymbol{W}_{Ki}^{h\top}\boldsymbol{W}_{Qi}^h) \right) \vee \max_h \left( \text{rank}(\overline{\boldsymbol{W}}_{1i}\overline{\boldsymbol{W}}_{Oi}^h\overline{\boldsymbol{W}}_{Vi}^h - \boldsymbol{W}_{1i}\boldsymbol{W}_{Oi}^h\boldsymbol{W}_{Vi}^h) \right), i = 1, \\ \max_h \left( \text{rank}(\overline{\boldsymbol{W}}_{2,i-1}^{\top}\overline{\boldsymbol{W}}_{Ki}^{h\top}\overline{\boldsymbol{W}}_{Qi}^h\overline{\boldsymbol{W}}_{2,i-1} - \boldsymbol{W}_{2,i-1}^{\top}\boldsymbol{W}_{Ki}^{h\top}\boldsymbol{W}_{Qi}^h\boldsymbol{W}_{2,i-1}) \right) \\ \qquad \vee \max_h \left( \text{rank}(\overline{\boldsymbol{W}}_{1i}\overline{\boldsymbol{W}}_{Oi}^h\overline{\boldsymbol{W}}_{Vi}^h\overline{\boldsymbol{W}}_{2,i-1} - \boldsymbol{W}_{1i}\boldsymbol{W}_{Oi}^h\boldsymbol{W}_{Vi}^h\boldsymbol{W}_{2,i-1}) \right), \qquad 2 \le i \le L, \\ \text{rank}(\overline{\boldsymbol{W}}_o\overline{\boldsymbol{W}}_{2L} - \boldsymbol{W}_o\boldsymbol{W}_{2L}), \qquad\qquad\qquad\qquad\qquad\qquad i = L+1. \end{cases}$$

$$(23)$$

*If $R \ge \max_{i \in [L+1]} \lceil \frac{G_i}{2} \rceil$, then there exists low-rank adapters with rank lower than $R \in [D]$ $((\Delta \boldsymbol{W}_{Kl}^h, \Delta \boldsymbol{W}_{Ql}^h, \Delta \boldsymbol{W}_{Vl}^h, \Delta \boldsymbol{W}_{Ol}^h)_{h=1}^H)_{l=1}^L, \Delta \boldsymbol{W}_{2L}, \Delta \boldsymbol{W}_o$ with other low-rank adapters set to $\boldsymbol{O}$, and updated bias vectors $(\widehat{\boldsymbol{b}}_{1l}, \widehat{\boldsymbol{b}}_{2l})_{l=1}^L$, such that for any $\boldsymbol{X} \in \mathbb{R}^{D \times N}$, the adapted model $f$ exactly approximates target model $\bar{f}$, i.e., $f(\boldsymbol{X}) = \bar{f}(\boldsymbol{X})$.*

*Proof of Theorem 7.* The key idea of this proof is the same as the proof of Theorem 8: our first step is to ensure that, for each transformer block, the output from the first feedforward layer in the target model matches that in the adapted model. Once this is established, we select an appropriate output layer weight matrix to complete the proof.

Similar to the proof of Theorem 8, we define $\overline{\boldsymbol{H}}_l \in \mathbb{R}^{D \times N}$ and $\overline{\boldsymbol{Z}}_l \in \mathbb{R}^{D \times N}$ as the intermediate and final outputs of the $l$-th transformer block in the target model $\bar{f}$, respectively. In particular, $\overline{\boldsymbol{H}}_l$ corresponds to the output of the first feedforward layer in the $l$-th transformer block. They are formulated as

$$\overline{\boldsymbol{H}}_l = \text{ReLU}\left( \overline{\boldsymbol{W}}_{1l} \left( \sum_{h=1}^H \overline{\boldsymbol{W}}_{Ol}^h \overline{\boldsymbol{W}}_{Vl}^h \cdot \overline{\boldsymbol{Z}}_{l-1} \cdot \text{softmax}\left( \overline{\boldsymbol{Z}}_{l-1}^\top \overline{\boldsymbol{W}}_{Kl}^{h\top} \overline{\boldsymbol{W}}_{Ql}^h \overline{\boldsymbol{Z}}_{l-1} \right) \right) + \bar{\boldsymbol{b}}_{1l} \boldsymbol{1}_N^\top \right),$$

$$\overline{\boldsymbol{Z}}_l = \overline{\boldsymbol{W}}_{2l} \overline{\boldsymbol{H}}_l + \bar{\boldsymbol{b}}_{2l} \boldsymbol{1}_N^\top.$$

For the adapted model $f$, we introduce $\widehat{\boldsymbol{H}}_l$ and $\widehat{\boldsymbol{Z}}_l$ accordingly to denote the intermediate output of the first feedforward layer and the final output of the $l$-th transformer block for the adapted model, respectively:

$$\widehat{\boldsymbol{H}}_l = \text{ReLU}\left( \boldsymbol{W}_{1l}\left( \sum_{h=1}^H (\boldsymbol{W}_{Ol}^h + \Delta \boldsymbol{W}_{Ol}^h)(\boldsymbol{W}_{Vl}^h + \Delta \boldsymbol{W}_{Vl}^h) \cdot \widehat{\boldsymbol{Z}}_{l-1} \right.\right.$$

$$\left.\left. \cdot \text{softmax}\left( \widehat{\boldsymbol{Z}}_{l-1}^\top (\boldsymbol{W}_{Kl}^h + \Delta \boldsymbol{W}_{Kl}^h)^\top (\boldsymbol{W}_{Ql}^h + \Delta \boldsymbol{W}_{Ql}^h)\widehat{\boldsymbol{Z}}_{l-1} \right) \right) + \widehat{\boldsymbol{b}}_{1l}\boldsymbol{1}_N^\top \right),$$

$$\widehat{\boldsymbol{Z}}_l = \boldsymbol{W}_{2l}\widehat{\boldsymbol{H}}_l + \widehat{\boldsymbol{b}}_{2l}\boldsymbol{1}_N^\top.$$

Note that $\overline{\boldsymbol{Z}}_0 = \widehat{\boldsymbol{Z}}_0 = \boldsymbol{X}$.

We aim to demonstrate that adding low-rank adapters to the weight matrices allows the adapted TFN $f$ to be functionally equivalent to the target TFN of identical dimensions. We will initiate our proof by inductively constructing the adapter weight matrices $((\Delta \boldsymbol{W}_{Ol}^h, \Delta \boldsymbol{W}_{Vl}^h, \Delta \boldsymbol{W}_{Kl}^h, \Delta \boldsymbol{W}_{Ql}^h)_{h=1}^H, \widehat{\boldsymbol{b}}_{1l}, \widehat{\boldsymbol{b}}_{2l})_{l=1}^L$ such that $\widehat{\boldsymbol{H}}_l = \overline{\boldsymbol{H}}_l$ for all $l \in [L]$, and then select the $\Delta \boldsymbol{W}_{2L}$ and the low-rank adapter for the output layer $\Delta \boldsymbol{W}_o$ to approximate the output of the target model. For unmentioned low-rank adapters, we set them as $\boldsymbol{O}$.

**When $l = 1$.** To achieve $\widehat{\boldsymbol{H}}_l$ with $\overline{\boldsymbol{H}}_l$ for all $\boldsymbol{X}$, we must satisfy the following conditions:

Bias Vector: $\widehat{\boldsymbol{b}}_{1l} = \bar{\boldsymbol{b}}_{1l}$, $\qquad\qquad\qquad\qquad\qquad\qquad\qquad\qquad\qquad\qquad$ (24)

Query and Key: $(\boldsymbol{W}_{Kl}^h + \Delta \boldsymbol{W}_{Kl}^h)^\top (\boldsymbol{W}_{Ql}^h + \Delta \boldsymbol{W}_{Ql}^h) = \overline{\boldsymbol{W}}_{Kl}^{h\top} \overline{\boldsymbol{W}}_{Ql}^h$,

Value and Output Projection: $(\boldsymbol{W}_{Ol}^h + \Delta \boldsymbol{W}_{Ol}^h)(\boldsymbol{W}_{Vl}^h + \Delta \boldsymbol{W}_{Vl}^h) = \boldsymbol{W}_{1l}^{-1} \overline{\boldsymbol{W}}_{1l} \overline{\boldsymbol{W}}_{Ol}^h \overline{\boldsymbol{W}}_{Vl}^h$.

To achieve this, we set $\widehat{\boldsymbol{b}}_{1l} = \bar{\boldsymbol{b}}_{1l}$ to achieve (24), and select rank-$R$ or lower matrices $\Delta \boldsymbol{W}_{Kl}^h, \Delta \boldsymbol{W}_{Ql}^h, \Delta \boldsymbol{W}_{Ol}^h, \Delta \boldsymbol{W}_{Vl}^h$ for all $h \in [H]$ as suggested by Lemma 7. This ensures $\widehat{\boldsymbol{H}}_l = \overline{\boldsymbol{H}}_l$ for $l = 1$.

**When $l > 1$.** Now we focus on the cases where $l = 2, \ldots, L$. Assume the induction hypothesis holds for $l - 1$, which is $\widehat{\boldsymbol{H}}_{l-1} = \overline{\boldsymbol{H}}_{l-1}$. Following the same steps in the proof of Theorem 8, we let $\widehat{\boldsymbol{b}}_{2,l-1} = \boldsymbol{W}_{2,l-1}\overline{\boldsymbol{W}}_{2,l-1}^{-1}\overline{\boldsymbol{b}}_{2,l-1}$, thereby obtaining,

$$\widehat{\boldsymbol{Z}}_{l-1} = \boldsymbol{W}_{2,l-1}\overline{\boldsymbol{W}}_{2,l-1}^{-1}\overline{\boldsymbol{Z}}_{l-1}. \tag{25}$$

To achieve $\widehat{\boldsymbol{H}}_l = \overline{\boldsymbol{H}}_l$, we express both $\widehat{\boldsymbol{H}}_l$ and $\overline{\boldsymbol{H}}_l$ in terms of $\overline{\boldsymbol{Z}}_{l-1}$:

$$\overline{\boldsymbol{H}}_l = \texttt{ReLU}\Big(\overline{\boldsymbol{W}}_{1l}\Big(\sum_{h=1}^{H}\overline{\boldsymbol{W}}_{Ol}^h\overline{\boldsymbol{W}}_{Vl}^h\cdot\overline{\boldsymbol{Z}}_{l-1}\cdot\mathrm{softmax}\Big(\overline{\boldsymbol{Z}}_{l-1}^\top\overline{\boldsymbol{W}}_{Kl}^{h\top}\overline{\boldsymbol{W}}_{Ql}^h\overline{\boldsymbol{Z}}_{l-1}\Big)\Big) + \overline{\boldsymbol{b}}_{1l}\mathbf{1}_N^\top\Big)$$

$$\widehat{\boldsymbol{H}}_l = \texttt{ReLU}\Big(\boldsymbol{W}_{1l}\Big(\sum_{h=1}^{H}(\boldsymbol{W}_{Ol}^h + \Delta\boldsymbol{W}_{Ol}^h)(\boldsymbol{W}_{Vl}^h + \Delta\boldsymbol{W}_{Vl}^h)\cdot\widehat{\boldsymbol{Z}}_{l-1}$$

$$\cdot\mathrm{softmax}\Big(\widehat{\boldsymbol{Z}}_{l-1}^\top(\boldsymbol{W}_{Kl}^h + \Delta\boldsymbol{W}_{Kl}^h)^\top(\boldsymbol{W}_{Ql}^h + \Delta\boldsymbol{W}_{Ql}^h)\widehat{\boldsymbol{Z}}_{l-1}\Big)\Big) + \widehat{\boldsymbol{b}}_{1l}\mathbf{1}_N^\top\Big),$$

$$\overset{(25)}{=} \texttt{ReLU}\Big(\boldsymbol{W}_{1l}\Big(\sum_{h=1}^{H}(\boldsymbol{W}_{Ol}^h + \Delta\boldsymbol{W}_{Ol}^h)(\boldsymbol{W}_{Vl}^h + \Delta\boldsymbol{W}_{Vl}^h)\cdot\boldsymbol{W}_{2,l-1}\overline{\boldsymbol{W}}_{2,l-1}^{-1}\overline{\boldsymbol{Z}}_{l-1}$$

$$\cdot\mathrm{softmax}\Big(\overline{\boldsymbol{Z}}_{l-1}^\top\overline{\boldsymbol{W}}_{2,l-1}^{-1\top}\boldsymbol{W}_{2,l-1}^\top(\boldsymbol{W}_{Kl}^h + \Delta\boldsymbol{W}_{Kl}^h)^\top$$

$$(\boldsymbol{W}_{Ql}^h + \Delta\boldsymbol{W}_{Ql}^h)\boldsymbol{W}_{2,l-1}\overline{\boldsymbol{W}}_{2,l-1}^{-1}\overline{\boldsymbol{Z}}_{l-1}\Big)\Big) + \widehat{\boldsymbol{b}}_{1l}\mathbf{1}_N^\top\Big).$$

Therefore, we need to align the following three components:

Bias Vector: $\widehat{\boldsymbol{b}}_{1l} = \overline{\boldsymbol{b}}_{1l}$, $\tag{26}$

Query and Key: $(\boldsymbol{W}_{Kl}^h + \Delta\boldsymbol{W}_{Kl}^h)^\top(\boldsymbol{W}_{Ql}^h + \Delta\boldsymbol{W}_{Ql}^h) = \boldsymbol{W}_{2,l-1}^{-1\top}\overline{\boldsymbol{W}}_{2,l-1}^\top\overline{\boldsymbol{W}}_{Kl}^{h\top}\overline{\boldsymbol{W}}_{Ql}^h\overline{\boldsymbol{W}}_{2,l-1}\boldsymbol{W}_{2,l-1}^{-1}$,

Value and Output Projection:

$$(\boldsymbol{W}_{Ol}^h + \Delta\boldsymbol{W}_{Ol}^h)(\boldsymbol{W}_{Vl}^h + \Delta\boldsymbol{W}_{Vl}^h) = \boldsymbol{W}_{1l}^{-1}\overline{\boldsymbol{W}}_{1l}\overline{\boldsymbol{W}}_{Ol}^h\overline{\boldsymbol{W}}_{Vl}^h\overline{\boldsymbol{W}}_{2,l-1}\boldsymbol{W}_{2,l-1}^{-1}.$$

By setting $\widehat{\boldsymbol{b}}_{1l}$ based on (26) and adjusting $\Delta\boldsymbol{W}_{Kl}^h, \Delta\boldsymbol{W}_{Ql}^h, \Delta\boldsymbol{W}_{Ol}^h, \Delta\boldsymbol{W}_{Vl}^h$ for all $h \in [H]$ based on Lemma 7, we satisfy all three conditions above, thereby obtaining $\widehat{\boldsymbol{H}}_l = \overline{\boldsymbol{H}}_l$ for $l \in [L] \setminus \{1\}$.

**Output Layer Analysis.** By applying the induction method, we have established $\widehat{\boldsymbol{H}}_l = \overline{\boldsymbol{H}}_l$ for all $l \in [L]$. Lastly, we choose the $\Delta\boldsymbol{W}_o, \Delta\boldsymbol{W}_{2L}$ and the bias vector $\widehat{\boldsymbol{b}}_{2L}$ using the same approach as in the proof of Theorem 8. This concludes the proof. $\square$

The following corollary identifies the specific LoRA-rank required to achieve exact representation for random model cases in the current setting.

**Corollary 10.** *Assume that the elements of all the weight matrices of both the target TFN and the frozen TFN are independently drawn from arbitrary continuous distributions. If $R \geq \lceil\frac{D}{2}\rceil$, adding low-rank adapters of rank at most $R$ to weight matrices in $((\Delta\boldsymbol{W}_{Kl}^h, \Delta\boldsymbol{W}_{Ql}^h, \Delta\boldsymbol{W}_{Vl}^h, \Delta\boldsymbol{W}_{Ol}^h)_{h=1}^H)_{l=1}^L, \Delta\boldsymbol{W}_{2L}, \Delta\boldsymbol{W}_o$ and tuning the bias vectors, enables the adapted model $f$ to exactly approximate the target model $\overline{f}$, i.e., $f(\boldsymbol{X}) = \overline{f}(\boldsymbol{X})$ for all $\boldsymbol{X} \in \mathbb{R}^{D \times N}$.*

*Proof of Corollary 10.* By combining Lemma 14 and Theorem 7, and following the same steps in the proof of Corollary 4 which yields $\max_i G_i = D$, we can obtain the desired outcome. $\square$

## G    EXPERIMENTS

In this section, we perform experiments on both synthetic and real datasets to corroborate our theoretical results. Firstly, we focus on validating the construction of the LoRA adapter in our proof.

Subsequently, we extend our experimental validation to encompass the effects of tuning final layers and the significance of updatable bias. Additionally, we offer visual representations of training curves, assess the generalization performance of LoRA, and evaluate its efficacy on classification tasks. We also conduct experiments on real datasets to further support our theoretical insights in real-world scenarios.

### G.1   ADDITIONAL DETAILS OF EXPERIMENT SETUP

We implement LoRA adapter $\Delta W$ by reparameterizing it as $\Delta W = AB^\top$, where $A, B \in \mathbb{R}^{D \times R}$, and we use the same initialization scheme as proposed by Hu et al. (2022a). For experiments presented in Sec. 5, G.3.1, G.3.2, G.4, and G.5, we consider two variants of frozen models:

- **(Random)** The first method involves randomly generating all the weight matrices using the Xavier uniform distribution, which is the default weight initialization method used in PyTorch.

- **(Pretrained)** The second method aims to simulate scenarios where the pretrained model is relatively closer to the target model. We achieve this by initially creating the target model and the frozen model in the same way as the first method and then performing full-rank updates on the frozen model via gradient descent to approximate the target model until the approximation error is reduced by 1/3.

For other experiments on synthetic datasets, we default to the randomly parameterized frozen model unless specified otherwise.

### G.2   ADDITIONAL DETAILS ON GRADIENT UPDATE METHOD

In our experiments, we utilize the Adam optimizer. We tune the learning rate $\in \left\{10^{-2}, 10^{-3}, 10^{-4}\right\}$ and the weight decay $\in \left\{0, 10^{-2}, 10^{-3}, 10^{-4}\right\}$. The optimal configuration is determined based on the validation loss on a set of 256 samples independently drawn from a standard normal distribution. We run 5,000 iterations for each hyperparameter setting, where at each step 256 fresh standard Gaussian samples are generated for loss and gradient computation.

### G.3   VALIDATION OF OUR LORA ADAPTER CONSTRUCTION

Recall that all our theoretical statements are based on our construction of the LoRA adapters presented in their corresponding proofs. To validate these results, here we empirically examine the relationship between approximation error and rank by integrating the LoRA adapters, which are constructed with the uniform partition in our proof, into the frozen model. Furthermore, we evaluate the effectiveness of our constructed LoRA adapters by comparing their performance against adapters updated through gradient descent and optimized by Adam. All simulations are conducted five times using different seeds, and the reported values represent the median computed across different runs.

### G.3.1   FNN APPROXIMATION

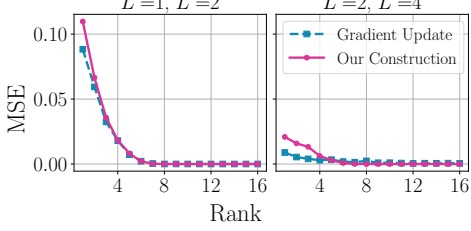
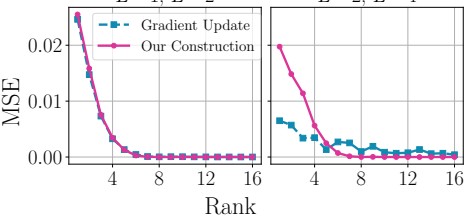

(a) Frozen model is randomly generated.

(b) Frozen model is pretrained.

Figure 3: Approximation error (measured by MSE) versus LoRA-rank on FNNs.

In this experiment, we assess the effectiveness of our low-rank adapter construction for FNN approximation, which is detailed in the proof of Theorem 5.

**Setup.** We consider two scenarios: one with $\overline{L} = 1$ and $L = 2$ and the other one with $\overline{L} = 2$ and $L = 4$. It should be noted that for both these cases, we have $M = \lfloor L/\overline{L} \rfloor = 2$ here. We employ the gradient update method and the construction outlined in the proof of Theorem 5 to update the LoRA adapters.

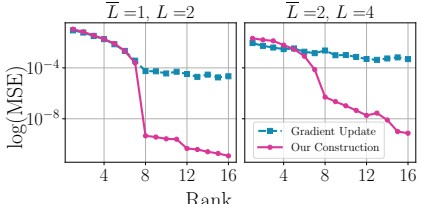

Figure 4: Log-scale MSE versus LoRA-rank on randomly initialized FNNs.

**Results.** Fig. 3 presents the results for FNN approximation. Consistent with the implications drawn in Sec. 5, the $y$ limit changes from Fig. 3a to Fig. 3b suggest that the pretrained frozen model results in less approximation error. Additionally, we observe that our construction's performance aligns closely with the gradient update method when the target model depth $\overline{L} = 1$. However, this alignment is not observed when $\overline{L} = 2$ on low-rank region (i.e., $R \leq 4$), This further underscores the limitation of our LoRA adapter construction, which inherently assumes that the intermediate outputs of the frozen model and the target model need to align.

To facilitate a more effective comparison between our construction and the gradient update method in the higher-rank region (i.e., $R \geq 6$), we present the curves on a logarithmic scale, as depicted in Fig. 4. While the gradient update appears to reach the optimal performance achieved by our LoRA construction in FNNs, a gap is still discernible when viewed on a logarithmic scale. The MSE of the gradient update method is approximately $10^{-4}$, while for our LoRA construction, it's around $10^{-8}$ for a sufficiently large rank.

### G.3.2 TFN APPROXIMATION

We assess the effectiveness of our LoRA adapter construction in approximating TFN, as detailed in the proof of Theorem 7.

**Setup.** We examine target model $\overline{f}$ and frozen model $f$, both featuring the same architecture with $L$ transformer blocks, a single output layer, two attention heads, and embedding size $D = 16$. We focus on two scenarios: $L = 1$ and $L = 2$. The weight matrices for the attention layers follow a standard Gaussian distribution, while those for the linear layers are initialized using the Xavier uniform distribution, which is PyTorch's default scheme for linear layer initialization.

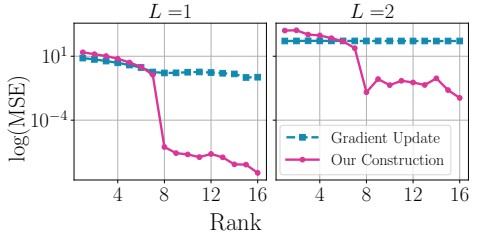
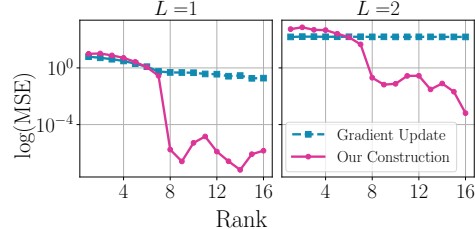

(a) Frozen model is randomly generated.    (b) Frozen model is pretrained.

Figure 5: Approximation error (measured by MSE) versus LoRA-rank on TFNs.

**Results.** The observations here align with those from the experiments of FNN approximation. We note that the gradient update method outperforms our approach when the rank is relatively small but lags behind as the rank increases. This advantage of the gradient update method at minimal ranks arises from the inherent complexity of TFNs, which allows for more flexible low-rank adapter construction. Meanwhile, the gradient update method's performance does not significantly improve as the rank increases. This arises from the inherent complexity involved in optimizing TFNs. Nonetheless, our results corroborate the claims made in Theorem 7, as the approximation error must be eradicated when the rank reaches $\lceil \frac{D}{2} \rceil = 8$.

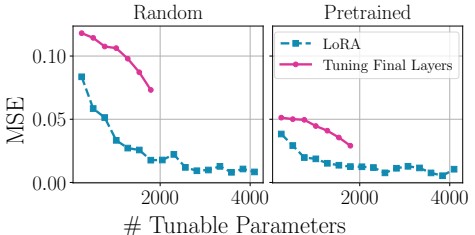 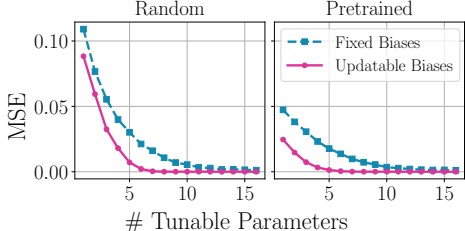

(a) Comparison between LoRA and tuning final layers.

(b) Comparison between LoRA with fixed biases and LoRA with updatable biases.

Figure 6: Approximation error (measured by MSE) versus the number of tunable parameters when various methods are employed. The analyses are conducted on FNN models.

### G.4 COMPARISON TO TUNING FINAL LAYERS

Tuning or adding the final layers only is also a common adaptation method used in various domains, including computer vision (Chatfield et al., 2014; Donahue et al., 2014; Sharif Razavian et al., 2014), and natural language processing (Devlin et al., 2019; Gira et al., 2022). Recall that Corollary 4 and Lemma 12 demonstrate that tuning final layers does not perform as well as LoRA for randomly generated models, provided the LoRA-rank satisfies the rank constraints shown in Corollary 4. In this experiment, we aim to validate this assertion and compare the performance of tuning final layers and LoRA in more general scenarios, such as when the frozen model has been pretrained, and when the LoRA-rank is smaller than required.

**Setup.** We consider FNN models with $D = 16, \overline{L} = 1, L = 8$. In this experiment, we employ two baselines: LoRA and tuning final layers. The LoRA adapters and the final layers are updated using the gradient update method.

**Results.** Figure 6a compares the MSE of LoRA and final layer tuning when the same number of tunable parameters are used. In the case of randomly generated models, we observe that final layer tuning yields a significantly higher MSE when using the same number of tunable parameters, corroborating our results in Lemma 12. However, when the frozen model has been pretrained, the performance of final layer tuning improves considerably, though it still falls short of LoRA. This aligns with conclusions drawn from previous theoretical studies such as Tripuraneni et al. (2020), which asserts that the performance of final layer tuning heavily depends on the quality of the shared representations.

### G.5 BENEFITS OF TUNING BIASES

In our proof, as detailed in Sec. 3.2 and E.1, the updatable biases in the FNN play a crucial role in eliminating the nonlinearity of ReLUs. In this experiment, we investigate the importance of updatable biases in ensuring the success of LoRA in FNN cases.

**Setup.** We consider FNN models with parameters $D = 16, \overline{L} = 1, L = 2$, and examine the performance of LoRA both with and without biases tuning for adapting it to match the target FNN. The LoRA adapters and biases are updated using the gradient update method.

**Results.** The performance of LoRA with and without updatable biases is presented in Figure 6b. We observe that in both random and pretrained model cases, LoRA with updatable biases outperforms LoRA with fixed biases when the number of tunable parameters is relatively small. However, the performance gap is not significant and diminishes as the number of tunable parameters increases. This suggests that while tuning biases in conjunction with the low-rank adapters does enhance performance, the gain is not substantial. In other words, even without bias tuning, LoRA's performance remains competitive.

### G.6 TRAINING CURVES

Although our theoretical study does not incorporate any training process, we present the training curves of the LoRA gradient update method to illuminate the optimization aspects of LoRA.

**Setup** We depict the training curves of LoRA fine-tuning on randomly generated FNNs for $R = 1, 4, 8, 13, 16$. Unless stated otherwise, all settings strictly adhere to the FNN experiments described in Sec. 5.

**Results** The training curves visualized in Fig. 7 reveal that models with smaller ranks (e.g., $R$=1,4) converge swiftly due to their limited search space, but they settle at a relatively high training loss. Medium rank models (e.g., $R$=8) converge more slowly. Highly overparameterized models (g, $R$=13,18) appear to converge faster, aligning with recent advancements in optimization theory, which suggest that overparameterized models are easier to optimize (Liu et al., 2022a).

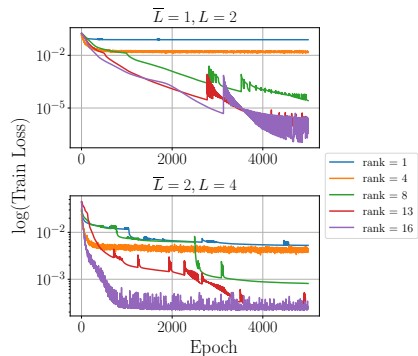

Figure 7: Training curves of LoRA with varying LoRA-ranks when $D = 16$.

### G.7 GENERALIZATION PERFORMANCES

While our theoretical study only establishes the upper bound of LoRA's performance with infinite data samples, it does not consider LoRA's generalization performance in practice. Although this is beyond the current scope of our paper, we empirically investigate LoRA's generalization performance in this experiment.

**Setup.** We include a training set of 400 samples for the cases where $\overline{L} = 1, L = 2$, and 800 training samples for the cases where $\overline{L} = 2, L = 4$. We evaluate how well LoRA's training performance transfers to the test set.

**Results.** Fig. 8 presents the training and test MSE versus LoRA-ranks. However, no clear pattern is observed in the variation of the gap between the training and test MSE with respect to the LoRA-ranks. This could be due to Adam not precisely finding the minimum (see Fig. 4), potentially avoiding overfitting.

To assess LoRA's generalization performance, we fine-tuned the frozen model on the training set and reported the training and test MSE. We notice an increasing generalization gap (test MSE - train MSE) as the LoRA rank

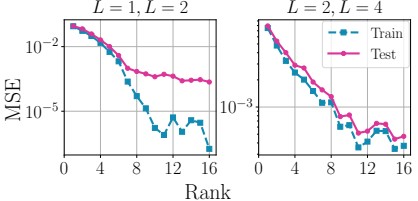

Figure 8: Assessment of LoRA's generalization performance on FNNs.

increases – this is very evident with $L$=2, and less so with $L$=4. This is intuitive as larger LoRA ranks imply a larger hypothesis class (e.g., the Rademacher complexity), so it is expected. We defer a detailed analysis of LoRA's generalization performance to future work but believe our simulation results provide a valuable starting point for further discussion and investigation.

### G.8 EVALUATION ON CLASSIFICATION TASKS

Our theory and previous experiments all focus on regression cases. In this experiment, we consider binary and multi-class classification tasks to optimize the LoRA adapter vias cross-entropy and report the performance of LoRA using accuracy.

**Multi-class Classification.** As shown in Fig. 9a, consistent with our theoretical results, our construction achieves 100% accuracy when $R \geq 8$. The performance of gradient update is also similar to our observation when MSE is employed as the metric, particularly when MSE is plotted on a logarithmic scale (Fig. 4). This observation echoes the findings of Hui & Belkin (2021), which indicate that optimizing MSE is fundamentally equivalent to optimizing cross-entropy.

**Binary Classification.** We have conducted binary classification tasks. We use the same setup as before but add one more output layer $\in \mathbb{R}^{2 \times D}$ which is a block diagonal matrix, with the first 8

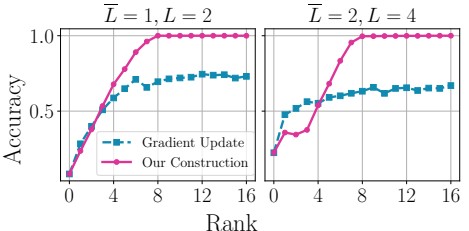 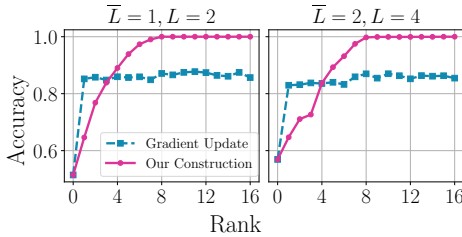

(a) Multi-class classification tasks with 16 classes.   (b) Binary classification task.

Figure 9: Accuracy versus the rank on classification tasks. The analyses are conducted on FNN models.

elements in the first rows and the last 8 elements in the second row are 1 and all remaining elements are 0. We fix this output layer, optimize the cross entropy on the LoRA adapters, and report the test accuracy.

As shown in Fig. 9b, we observe that in this binary classification scenario, even with a very low LoRA-rank $R = 1$, the accuracy has been significantly improved, comparable to the results achieved by higher ranks. In the region of higher ranks, our construction significantly outperforms the gradient update method. The suboptimal performance of the gradient update method in this simulation suggests that, despite LoRA's current impressive performance in practical applications, there is potential for further refinement.

### G.9 EVALUATION ON REAL DATASETS

In our theoretical analysis, we demonstrate how the sizes of frozen models and the distance between the frozen and target models influence the necessary LoRA-ranks to achieve the desired performance (see Lemma 1, 2, and Theorem 5, 6, 7). Specifically, our results suggest that larger models require fewer LoRA-ranks to reach the desired performance. Similarly, when the frozen model is closer to the target model, a lower LoRA-rank is sufficient to achieve the same performance. We validate these theoretical insights through experiments on the GLUE benchmark (Wang et al., 2018).

**Setup** Our experiments are conducted using Tesla V100-PCIE-16GB, NVIDIA A100-SXM4-80GB, NVIDIA A100-SXM4-40GB, and NVIDIA L40 GPUs. For each run, a single GPU is utilized. Unless otherwise specified, all our settings align with those established by Hu et al. (2022a).

**Impact of Model Size on LoRA Rank** In practice, most existing studies on LoRA use the same LoRA-rank for models of varying sizes. For instance, in the original LoRA paper (Hu et al., 2022a), Tables 9 and 10 demonstrate the use of the same LoRA-rank for RoBERTa-base (Liu et al., 2019), RoBERTa-large (Liu et al., 2019), and DeBERTa-XXL (He et al., 2021). Similarly, in the QLoRA paper (Dettmers et al., 2023), a LoRA-rank of 64 is set for different models ranging from 13B to 65B parameters (see their Appendix B.2). To validate our theoretical findings, we evaluated the performance of LoRA on models of different sizes, specifically RoBERTa-base with 110M parameters and RoBERTa-large with 340M parameters. The results are presented in Table 2.

Initially, we observe that, in the absence of fine-tuning (LoRA-rank $R = 0$), there is no consistent trend – RoBERTa-base performs better on 3 datasets, while RoBERTa-large performs better on 4 datasets. However, after LoRA fine-tuning, we observe that RoBERTa-large outperforms in most cases. In fact, even when the base model is trained with a LoRA-rank three times larger, RoBERTa-large still performs better on 6 out 8 datasets. Given that the pretrained RoBERTa-large model was performing no differently from the base model, this observation supports our theoretical findings that deeper models are more expressive with LoRA training.

**Impact of Model Proximity on LoRA Rank** While our theoretical results (Lemma 1, 2, and Theorem 5, 6, 7) imply that the frozen model that is closer to the target model achieves better results for a fixed LoRA-rank. To validate this, we compare the performance of pretrained RoBERTa-base with the randomly initialized RoBERTa-base fine-tuned using the same LoRA-ranks.

| Model | $R$ | MNLI | SST-2 | MRPC | CoLA | QNLI | QQP | RTE | STS-B |
|---|---|---|---|---|---|---|---|---|---|
| RoBERTa$_{base}$ | 0 | **.330** | .491 | .316 | 0 | .495 | **.682** | **.527** | .024 |
| RoBERTa$_{large}$ | 0 | .318 | **.505** | **.684** | 0 | **.505** | .369 | .473 | **.032** |
| RoBERTa$_{base}$ | 2 | .861 | .950 | .892 | **.632** | .928 | .891 | .780 | .907 |
| RoBERTa$_{base}$ | 6 | .870 | .948 | .892 | .629 | .931 | **.900** | .773 | .909 |
| RoBERTa$_{large}$ | 2 | **.904** | **.956** | **.917** | .631 | **.946** | .887 | **.884** | **.916** |

Table 2: **Comparison of the fine-tuned performance of RoBERTa-base and RoBERTa-large using LoRA with different LoRA-ranks on the GLUE benchmark.** Following Hu et al. (2022a), we report the overall (matched and mismatched) accuracy for MNLI, Matthew's correlation for CoLA, Pearson correlation for STS-B, and accuracy for other tasks. Higher is better for all metrics. Despite the absence of a clear pattern indicating which pretrained model is generally superior, after fine-tuning using LoRA, we observe that RoBERTa-large (340M) fine-tuned with LoRA-rank $R = 2$ outperforms RoBERTa-base (110M) with LoRA-rank $R = 6$ in 7 out of 8 tasks. This observation aligns with our theoretical conclusion that larger models require lower LoRA-ranks to achieve the desired performance.

| Model | $R$ | MNLI | SST-2 | MRPC | CoLA | QNLI | QQP | RTE | STS-B |
|---|---|---|---|---|---|---|---|---|---|
| Random | 2 | .523 | .775 | .691 | .154 | .627 | .761 | .542 | .213 |
| Pretrained | | **.861** | **.950** | **.892** | **.632** | **.928** | **.891** | **.780** | **.907** |
| Random | 4 | .535 | .788 | .696 | .145 | .625 | .768 | .542 | .224 |
| Pretrained | | **.868** | **.950** | **.890** | **.634** | **.929** | **.898** | **.805** | **.910** |
| Random | 6 | .544 | .799 | .696 | .154 | .632 | .768 | .542 | .210 |
| Pretrained | | **.868** | **.948** | **.892** | **.629** | **.931** | **.900** | **.773** | **.909** |

Table 3: **Comparison of the fine-tuned performance of randomly initialized and pretrained RoBERTa-base.** Following Hu et al. (2022a), we report the overall (matched and mismatched) accuracy for MNLI, Matthew's correlation for CoLA, Pearson correlation for STS-B, and accuracy for other tasks. Higher is better for all metrics. We observe that the performance of the pretrained RoBERTa-base significantly surpasses that of the randomly initialized RoBERTa-base given the same LoRA-rank. This observation is consistent with our theoretical findings, which suggest that a frozen model closer to the target model yields better performance given the same LoRA-rank.

The results in Table 3 demonstrate that the pretrained RoBERTa-base significantly surpasses the randomly initialized RoBERTa-base. This observation is consistent with our theoretical findings, suggesting that the pretrained model requires lower LoRA-ranks to achieve the desired performance.

## H    EXTENSION TO CASES WITH DIFFERENT MODEL DIMENSIONS

This discussion only applies to linear model approximation and FNN approximation. As highlighted in Sec. 2, our results can be easily extended to scenarios where the target model, $\bar{f}$, and the frozen model, $f$, have different model dimensions. Specifically, for linear model or FNN approximation, we use $\bar{D}$ to represent the number of hidden neurons per layer in the target model and $D$ for the frozen model. We particularly consider the cases where the frozen model is wider than the target model, i.e., $D \geq \bar{D}$. This is because the frozen model is typically overparameterized in practical applications.

The key idea for extending our analysis to scenarios with different model dimensions is expanding the dimension of the target model. For the sake of simplicity, we focus on the simplest case, the linear model approximation, as an example. In this setting, the difference between the output of the

adapted model and the target model can be measured by

$$f\left(\begin{bmatrix} \boldsymbol{x} \\ \boldsymbol{0} \end{bmatrix}\right) - \begin{bmatrix} \bar{f}(\boldsymbol{x}) \\ \boldsymbol{0} \end{bmatrix} = \prod_{l=1}^{L} (\boldsymbol{W}_l + \Delta\boldsymbol{W}_l) \begin{bmatrix} \boldsymbol{x} \\ \boldsymbol{0} \end{bmatrix} - \begin{bmatrix} \overline{\boldsymbol{W}}\boldsymbol{x} \\ \boldsymbol{0} \end{bmatrix}, \tag{27}$$

where $\boldsymbol{x} \in \mathbb{R}^{\overline{D}}$. Consequently, the last $(D - \overline{D})$ columns and rows of $\prod_{l=1}^{L} (\boldsymbol{W}_l + \Delta\boldsymbol{W}_l)$ does not affect the results at all. Denote the submatrix consisting of the first $d$ rows and $d$ columns of a matrix $\boldsymbol{W}$ by $[\boldsymbol{W}]_d$. Then, to approximate the target model, we aim to solve the following constrained optimization problem for a given LoRA-rank $R \in [D]$:

$$\min_{\mathrm{rank}(\Delta\boldsymbol{W}_l) \leq R} \left\| \left[ \prod_{l=1}^{L} (\boldsymbol{W}_l + \Delta\boldsymbol{W}_l) \right]_{\overline{D}} - \overline{\boldsymbol{W}} \right\|_{\mathrm{F}}.$$

To solve this problem, we first define an expanded target matrix, denoted by $\widetilde{\boldsymbol{W}} \in \mathbb{R}^{D \times D}$. The expanded target matrix $\widetilde{\boldsymbol{W}}$ is constructed such that $\left[\widetilde{\boldsymbol{W}}\right]_{\overline{D}} = \overline{\boldsymbol{W}}$, while the remaining entries matches the corresponding entries in $\prod l = 1^L \boldsymbol{W}_l$. Then, the error matrix $\boldsymbol{E} = \widetilde{\boldsymbol{W}} - \prod_{l=1}^{L} \boldsymbol{W}_l$, consists entirely of zeros except for the first $\overline{D}$ rows and $\overline{D}$ columns. Therefore, we obtain $R_{\boldsymbol{E}} = \mathrm{rank}(\boldsymbol{E}) \leq \overline{D}$.

Given the expanded target matrix, we consider the updated constrained optimization problem as follows:

$$\min_{\mathrm{rank}(\Delta\boldsymbol{W}_l) \leq R} \left\| \prod_{l=1}^{L} (\boldsymbol{W}_l + \Delta\boldsymbol{W}_l) - \widetilde{\boldsymbol{W}} \right\|_{\mathrm{F}}. \tag{28}$$

By Lemma 1, we obtain that when the LoRA-rank $R \geq \lfloor \frac{\overline{D}}{L} \rfloor$, the optimal solution to (28) satisfies $\prod_{l=1}^{L} (\boldsymbol{W}_l + \Delta\boldsymbol{W}_l) = \widetilde{\boldsymbol{W}}$, given that $\overline{D} \geq R_{\boldsymbol{E}}$. This result implies that $\left[\prod_{l=1}^{L} (\boldsymbol{W}_l + \Delta\boldsymbol{W}_l)\right]_{\overline{D}} = \overline{\boldsymbol{W}}$ and therefore the approximation error defined in (27) is 0 for all input $\boldsymbol{x}$.

A similar analysis can be conducted for FNN approximation.

## I    EXTENDED FUTURE WORKS

To the best of our knowledge, this paper is the first to offer a theoretical understanding of LoRA fine-tuning on both FNN and TFN. Our work delivers insightful results, elucidating the impact of rank, depth of the pre-trained model, and the distance between the pre-trained model and the target model on the expressive power of LoRA. Those theoretical results are further corroborated via our experiments. Despite these advancements, several intriguing questions still remain open. First, as observed in the numerical experiments, our construction of LoRA adapters for FNN and TFN may not be always optimal. Given that more complex models offer increased flexibility, an open question is whether we can devise a more parameter-efficient scheme to construct the LoRA adapters, thereby deriving a tighter bound on approximation error. Second, for TFN, we have only identified the conditions under which the LoRA-adapted model exactly matches the target model, due to the analytical complexity of TFN. It would be interesting to quantify the approximation error when the rank is lower than required. Furthermore, for TFN, we constrain the target model and the frozen model to have identical embedding size and depth, and we omit the skip connections and layer norms for simplicity. Another intriguing direction would be to study the expressive power of LoRA under TFN cases with more general settings on TFN architectures. While our analysis does not involve any training process, an interesting direction for future research would be to consider gradient-based optimization algorithms and examine how efficiently LoRA can be optimized. Finally, theoretical questions about LoRA's generalization to unseen data also remain unresolved.

