# OpenReview forum: "The Expressive Power of Low-Rank Adaptation"
_ICLR.cc/2024/Conference — ICLR 2024 poster_

### Official Review · Reviewer_SRgL · 2023-11-01

**Soundness:** 3 good
**Presentation:** 3 good
**Contribution:** 3 good
**Rating:** 6
**Confidence:** 2

**Summary:**

The authors conduct theoretical analysis for LoRA, a popular PEFT method for LLMs. For linear models with LoRA,  the “effective rank” is the sum of these low ranks. For multi-layer ReLU FNN, the effective expressive power of LoRA is nearly optimal up to a constant factor of 2. For transformer networks, adding LoRA adapters primarily to the self-attention layers enables the adapted model to exactly represent the target model.

**Strengths:**

+ The first theoretical analysis to understand the expressive power of LoRA. The first known results on the expressive power of LoRA
+ Linear models, FFNs, and transforms with LoRA are analyzed, providing comprehensive theoretical results.
+ Empirical results matches the rank requirements in theoretical analysis.

**Weaknesses:**

+ A notation table would help understand all the notations, since the paper is mostly about theoretical proof.

**Questions:**

In section 2, why a L-layer (instead of one-layer) linear model is considered, which is still a linear model?

---

> ### Author Response · Authors · 2023-11-21
> **To Reviewer SRgL**
>
> We sincerely thank the reviewer for acknowledging the theoretical significance and appreciating the empirical results of our paper. We are pleased to inform you that we have carefully addressed each of your concerns, as elaborated below.
>
> ---
>
> > Q1: A notation table would help understand all the notations since the paper is mostly about theoretical proof.
>
> Thanks for the suggestion. Due to the page limit, We have summarized a list of the notations in Appendix A.
>
> > Q2: In section 2, why a L-layer (instead of one-layer) linear model is considered, which is still a linear model?
>
> Correct. In our paper, we employ *deep linear models* (L-layer linear model) as a preliminary model, which serves as a foundation for extending our results to nonlinear models (i.e., FNN and TFN). Studying this toy model is a common technique in theoretical deep learning research, which offers valuable insights into deep nonlinear models, and has been employed in many notable studies, including those by `Saxe et al., 2014`, `Kawaguchi, 2016`, `Lu & Kawaguchi et al., 2017`, `Hardt & Ma`, and `Laurent & Brecht, 2018`.
>
> ---
>
> **Final Note:** We want to thank you again for all the questions and comments you have provided. If there are any remaining questions, please do not hesitate to let us know. If our responses have resolved your concerns, we kindly request you to consider increasing your score and championing our paper.
>
> *References*
>
> * Saxe, Andrew M., James L. McClelland, and Surya Ganguli. Exact solutions to the nonlinear dynamics of learning in deep linear neural networks. ICLR, 2014.
> * Kawaguchi, Kenji. Deep learning without poor local minima. NeurIPS, 2016.
> * Lu, Haihao, and Kenji Kawaguchi. Depth creates no bad local minima. arXiv, 2017.
> * Hardt, Moritz, and Tengyu Ma. Identity matters in deep learning. ICLR, 2017.
> * Laurent, Thomas, and James Brecht. Deep linear networks with arbitrary loss: All local minima are global. ICML, 2018.

---

### Official Review · Reviewer_U2Yh · 2023-11-01

**Soundness:** 4 excellent
**Presentation:** 4 excellent
**Contribution:** 4 excellent
**Rating:** 8
**Confidence:** 4

**Summary:**

The paper provides an initial theoretical exploration of the popular parameter-efficient finetuning method LoRA. It proves for fully connected models, LoRA should be sufficient to finetune any base model for a smaller target model with a certain LoRA rank (threshold). They further provide approximation errors for the case when the rank is smaller than the threshold.

**Strengths:**

This is a theoretically strong paper, studying a very timely topic. While empirically, LoRA has been shown to do surprisingly well, a theoretical explanation for why has been missing. This paper is a good starting point in understanding how/why/when LoRA works.

**Weaknesses:**

While it is okay to not have them in this paper, I think it would be interesting to study other effects of LoRA theoretically. For example, how does LoRA affect generalization? What can we say about how fast LoRA can converge even if the target model can eventually be found by LoRA exactly.

**Questions:**

See above.

---

> ### Author Response · Authors · 2023-11-21
> **To Reviewer U2Yh**
>
> We thank the reviewer for the encouraging feedback, especially for recognizing the theoretical contribution, timeliness, and importance of our paper in illuminating the empirical success of LoRA. Our response is detailed below.
>
> ---
>
> > Q1: While it is okay to not have them in this paper, I think it would be interesting to study other effects of LoRA theoretically. For example, how does LoRA affect generalization? What can we say about how fast LoRA can converge even if the target model can eventually be found by LoRA exactly.
>
> We agree with the reviewer that it is an interesting research topic to explore the generalization and optimization problems of LoRA theoretically. While an in-depth analysis of these properties is beyond the scope of this paper, as the reviewer acknowledges, we have included additional simulation results and discussion based on your comments that provide initial insight into these issues. `Please find our responses in the general comment to all reviewers and AC, specifically under "(Major Update #2) Illuminating the Optimization and Generalization Aspects of LoRA"`.
>
>
> ---
> **Final Note:** We are excited that you consider our work both timely and theoretically strong. We plan to incorporate these responses into our updated version. Thank you once again for your insightful comments and for your encouraging feedback!

---

> > ### Comment · Reviewer_U2Yh · 2023-11-22
> > **response to the rebuttal**
> >
> > I thank the authors for their response. I will keep my original score.

---

### Official Review · Reviewer_F9Xm · 2023-11-02

**Soundness:** 2 fair
**Presentation:** 3 good
**Contribution:** 2 fair
**Rating:** 6
**Confidence:** 4

**Summary:**

This paper presents a theoretical analysis of Low-Rank Adaptation (LoRA), a technique for efficiently fine-tuning pre-trained models, including large language and diffusion models. It establishes that LoRA can effectively adapt a fully connected neural network to represent a smaller target model if the LoRA-rank is sufficiently high. Specifically, the required rank is at least the product of the model's width and depth. For Transformer networks, the study demonstrates that a model can be fine-tuned to match a target of the same size using LoRA adapters of a particular rank. These theoretical assertions are underpinned by practical numerical experiments.

The paper concludes by highlighting the importance of LoRA's rank and the pre-trained model's depth in achieving close approximation to the target model. Despite these advances, it points out that the construction method for LoRA adapters might not be fully optimized and that better parameter efficiency could be achieved with more refined techniques. The paper also calls for additional research to measure approximation errors when LoRA-rank is not ideal, especially in the context of Transformer networks, and to further explore the application of LoRA in more complex network architectures.

**Strengths:**

- The study conducts a thorough analysis of the expressive capabilities of LoRA, underpinned by a set of well-founded assumptions.

- The findings from this research offer a theoretical foundation for applying LoRA to a diverse range of models, including Transformers and Diffusion models, and furnish insights on how to select hyper-parameters for designing LoRA effectively.

- The insights provided by this work can streamline the design process for LoRA, especially when the depth and width of the model in question are specified.

**Weaknesses:**

The experimental approach raises significant concerns. Given the widespread application of LoRA to various large language models (LLMs), such as LLaMA, there's an opportunity for the authors to substantiate their findings using models tasked with different challenges. Considering the availability of various model sizes in LLaMA and the comprehensive range of results provided by the original LoRA study, a comparison between the proposed theoretical analysis and empirical observations of LoRA would be insightful.

The use of Mean Squared Error (MSE) as a metric in the authors' presentation is questionable. Performance scores for LLMs typically exhibit a weak correlation with perplexity (PPL) or loss values. Therefore, relying solely on MSE for validation, particularly in the context of generative AI, may not adequately address the nuances of expressive capability. A multifaceted evaluation, including different performance metrics, would offer a more robust validation of the claims made in this work.

Additionally, it is acknowledged within the community that even very low ranks (such as 4, 2, or even 1) can yield satisfactory fine-tuning results. Readers would benefit from an exploration into how low-rank adjustments are able to achieve effective fine-tuning. The experimental outcomes presented in the paper currently do not offer practical insights for practitioners working with LoRA-based tuning, who would be looking for such guidance.

The authors are urged to establish a clearer connection between their theoretical discoveries and the empirical results previously reported for LoRA. Doing so could significantly streamline the hyper-parameter selection process for LoRA, reducing the effort required to fine-tune models effectively.

**Questions:**

Please refer to Weakness comments

---

> ### Author Response · Authors · 2023-11-21
> **To Reviewer F9Xm**
>
> **We wish to clarify that our primary objective is to understand the effectiveness of LoRA via theory, rather than experiments.** We thank the reviewer for recognizing the thoroughness of our theoretical analysis and their contribution to the empirical understanding of LoRA. We believe this recognition affirms the significant theoretical contribution of our paper. While we acknowledge that not all of our findings may be directly applicable to practical settings, the theoretical insights provided by our work form a crucial foundation for future practical applications.
>
> In light of the feedback, and although it was beyond the original scope of our work, we have conducted additional empirical studies and are excited to share these findings with you! If there are any remaining questions, please do not hesitate to let us know. If all of your concerns have been fully resolved, we kindly request you to consider increasing your score.
>
> ---
>
> > Q1: Given the widespread application of LoRA to various large language models (LLMs), there's an opportunity for the authors to substantiate their findings using models tasked with different challenges.
>
> As per your question, we have conducted additional experiments using LLMs such as RoBERTa-base and RoBERTa-large on GLUE benchmark to substantiate our theoretical results. `Please find our response in general comments to all reviewers and AC, specifically the first two findings under major update #1.`
>
> > Q2: A multifaceted evaluation, including different performance metrics, would offer a more robust validation of the claims made in this work.
>
> Our real-world experiments conducted on the GLUE benchmark include various metrics such as overall (matched and mismatched) accuracy for MNLI, Matthew's correlation for CoLA, Pearson correlation for STS-B, and accuracy for other tasks.
>
> For our synthetic simulations, we have expanded the scope to include multi-class classifications. We quantize the results and optimize cross-entropy rather than MSE, and subsequently report the test accuracy.
>
> As shown in Figure 5b (https://hackmd.io/_uploads/BkedJmrNT.png), consistent with our theoretical results, our construction achieves 100% accuracy when $R \geq 8$. The performance of gradient update is also similar to our observation when MSE is employed as the metric, particularly when MSE is plotted on a logarithmic scale (Figure 4: https://hackmd.io/_uploads/HksV_SBNp.png). This observation echoes the findings of `Hui & Belkin et al. (2021)`, which indicate that optimizing MSE is fundamentally equivalent to optimizing cross-entropy.
>
> The suboptimal performance of gradient update method in this simulation suggests that, despite LoRA's current impressive performance in practical applications, there is potential for further refinement.
>
> > Q3: Additionally, it is acknowledged within the community that even very low ranks (such as 4, 2, or even 1) can yield satisfactory fine-tuning results. Readers would benefit from an exploration into how low-rank adjustments are able to achieve effective fine-tuning.
>
> We believe that our current results indeed explain when and why low-rank adjustments can yield satisfactory performance. We further elucidate this with additional empirical insights.
>
> 1. **Theoretical Understanding**: Our theoretical findings demonstrate that larger models requires lower ranks for effective fine-tuning. This insight is particularly relevant when employing large language models or stable diffusion models.
>
> 2. **New Real-World Experiments**: We have also added new experiments comparing RoBERTa-base and RoBERTa-large, `as detailed in the first two findings of our summary in major update #1 in general comments to AC and all reviewers.` These experiments further validate our theory, which suggests that larger models require smaller ranks for effective fine-tuning.
>
>
> > Q4: The authors are urged to establish a clearer connection between their theoretical discoveries and the empirical results previously reported for LoRA. Doing so could significantly streamline the hyper-parameter selection process for LoRA, reducing the effort required to fine-tune models effectively.
>
> In response to your insightful suggestion to better integrate our theoretical findings with empirical observations, we have summarized both our theoretical and corresponding (additional) empirical results `in our summary of major update #1 in general comments to all reviewers and AC`. This enhanced and integrated discussion has been thoughtfully incorporated into the final version of our paper (Sec. H).
>
> ---
>
> **Final Note**: Thank you for your detailed comments. We are excited that you found the theoretical study of our paper to be thorough. If there are any remaining questions, please do not hesitate to let us know. If our responses have resolved your concerns, we kindly request you to consider increasing your score and support the acceptance of our paper.

---

> > ### Author Response · Authors · 2023-11-21
> > **References**
> >
> > * Hu, Edward J., et al. LoRA: Low-rank adaptation of large language models. ICLR, 2022.
> > * Zhang, Qingru, et al. Adaptive budget allocation for parameter-efficient fine-tuning. ICLR, 2023.
> > * Wang, Alex, et al. GLUE: A multi-task benchmark and analysis platform for natural language understanding. ACL, 2018.
> > * Dettmers, Tim, et al. QLoRA: Efficient finetuning of quantized llms. arXiv, 2023.
> > * Hui, Like, and Mikhail Belkin. Evaluation of neural architectures trained with square loss vs cross-entropy in classification tasks. ICLR, 2021.

---

> > ### Comment · Reviewer_F9Xm · 2023-11-21
> > **Response from Reviewer F9Xm**
> >
> > Thank you for your detailed comments and the addition of further experiments. I appreciate the efforts made to address my major concerns, particularly through the comprehensive experimental results provided. Based on these improvements, I have decided to raise my evaluation score.

---

### Official Review · Reviewer_YxPR · 2023-11-03

**Soundness:** 3 good
**Presentation:** 3 good
**Contribution:** 3 good
**Rating:** 6
**Confidence:** 3

**Summary:**

While conventional fine-tuning updates all model parameters for specialized tasks, full weight updating would be prohibitive for large language models (LLMs). Many methods were proposed to selectively update smaller parameter subsets or introduce lightweight adapters, significantly reducing computational and storage costs. The dominant method in this context is Low-Rank Adaptation (LoRA), which employs low-rank adapters to pre-trained weight matrices. Empirical evidence shows that LoRA can match or surpass the performance of full fine-tuning. However, there is a lack of theoretical understanding regarding how LoRA works, including questions about the minimum rank of adapters required for effective adaptation and how model architecture influences this threshold. Addressing these theoretical questions will provide valuable insights into the effectiveness and principles behind LoRA's adaptation of LLMs.

**Strengths:**

1. This paper claims that they are the first to study the expressive power of Low-Rank Adaptation (LoRA) for different model architectures. So, if this is true (I do not have sufficient knowledge to check), the novel of this paper is significant.

2.  Their theoretical results align well with the recent advances of LoRA on LLMs.

3.  Not only FNN but TFN is explored with the both theoretical and emperical study.

**Weaknesses:**

(1) From Figure 1, I can see that LoRA of FNN performs on par with gradient update, whereas LoRA of TFNs significantly outperform gradient updates. Could the author explain this performance difference?

(2) It is impressive that LoRA with rank=1 can match the performance of gradient update in Figure 3. Does this mean the gradient update does not actually learn well?

**Questions:**

Please see the above weaknesses.

---

> ### Author Response · Authors · 2023-11-21
> **To Reviewer YxPR**
>
> We sincerely appreciate your recognition of the novelty of our paper, its consistency with recent advances of LoRA, and the comprehensiveness of our theoretical and empirical results. Your concerns are addressed below. If our responses have resolved your concerns, we kindly request you to consider increasing your score and championing our paper.
>
> ---
>
> > Q1: From Figure 1, I can see that LoRA of FNN performs on par with gradient update, whereas LoRA of TFNs significantly outperform gradient updates. Could the author explain this performance difference?
>
> In Fig. 1, we plot MSE, while in Fig. 5, we plot log(MSE). Sorry for the confusion, and we will make this clear in the revision. We have added Fig. 4 (https://hackmd.io/_uploads/Skgt22SV6.png) to report performances of FNN in log(MSE), and we observe that gradient update underperforms our construction on both FNN and TFN.
>
>
> > Q2: It is impressive that LoRA with rank=1 can match the performance of gradient update in Figure 3 (which is Figure 5 in the updated version). Does this mean the gradient update does not actually learn well?
>
> The reviewer is correct that gradient update might fail to find the optimal LoRA parameters in the TFN cases. This underperformance of gradient update method, especially compared to our LoRA adapter construction, suggests that LoRA's current effectiveness might still have room for further improvement. A more sophisticated implementation could potentially yield even better results, indicating a promising direction for future exploration and development in this area.
>
> ---
>
> **Final Note:** We are excited that you find our work novel and appreciate our theoreical study and the corresponding empirical validation. If you have any further questions, please do not hesitate to let us know. If our responses have resolved your concerns, we kindly request you to consider increasing your score and championing our paper.

---

### Author Response · Authors · 2023-11-21
**To AC and All Reviewers**

We thank the reviewers for providing valuable suggestions that helepd us improve our paper.

We are particularly encouraged that the reviewers have found that (i) the novelty of our paper is important `(R-YxPR, R-U2Yh, R-SRgL)`, (ii) the topic is timely `(R-U2Yh)`, (iii) our theoretical analysis is thorough and strong `(R-YxPR, R-F9Xm, R-U2Yh, R-SRgL)`, (iv) our theoretical results are validated by numerical experiments `(R-SRgL)`, (v) it aligns well with the recent advances of LoRA on LLMs `(R-YxPR)`, and (vi) the insights we provide help in designing LoRA processes more efficiently `(R-F9Xm)`.

In response to the feedback, we've addressed each concern, added new theoretical results and experiments, and updated our paper accordingly. These additions clarify the connections between our theory and empirical observations, and provide insights into LoRA's optimization and generalization aspects.

The summary of our new major results is detailed below.

### **(Major Update #1) Bridging Theoretical Insights and Empirical Observations**

In our original submission, our primary focus was on the theoretical understanding of LoRA's expressive power, and we did not extensively investigate the consistency of our theoretical findings with practical observations reported in the literature. Following the insightful suggestions from the reviewers, we have added an extensive set of experiments to validate that our findings are indeed highly consistent with the practical performance of LoRA.

The full details of the additional experimental results are provided in the appendix, and below we offer a summary of our findings.

1. **For a fixed downstream task, larger models require a lower LoRA-rank to achieve the desired performance.**

    * Our Theoretical Results: This finding is substantiated by Lemma 1, 2, and Theorem 5, 6.
    * Empirical Observation: We have conducted new experiments to further support this claim. The following discussion has been incorporated into Sec. G.8 of our revised version.

        In practice, most existing studies on LoRA use the same LoRA-rank for models of varying sizes. For instance, see the original LoRA paper`(Hu et al., 2022)`'s Tables 9 and 10 and the QLoRA paper `(Dettmers et al., 2023)`'s Appendix B.2.

        However, our analysis indicates that deeper models require a lower LoRA-rank to achieve the same performance for a given downstream task. To empirically validate this, we evaluated the performance of LoRA on models of different sizes, specifically RoBERTa-base (110M) and RoBERTa-large (340M), using the GLUE benchmark `(Wang et al., 2018)`.

        | Model | MNLI | SST-2 | MRPC | CoLA | QNLI | QQP | RTE | STS-B |
        | ----- | ---- | ----- | ---- | ---- | ---- | --- | --- | ----- |
        | base @ R = 0 | **.330** | .491 | .316 | 0 | .495 | **.682** | **.527** | .024 |
        | large @ R = 0 | .318 | **.505** | **.684** | 0 | **.505** | .369 | .473 | **.032** |
        | base @ R = 2 | .861 | .950 | .892 | **.632** | .928 | .891 | .780 | .907
        | base @ R = 6 | .870 | .948 | .892 | .629 | .931 | **.900** | .773 | .909 |
        | large @ R = 2 | **.904** | **.956** | **.917** | .631 | **.946** | .887 | **.884** | **.916** |

        Initially, we observe that, in the absence of fine-tuning (LoRA-rank R = 0), there is no consistent trend -- RoBERTa-base performs better on 3 datasets, while RoBERTa-large performs better on 4 datasets.

        However, after LoRA fine-tuning, we observe that RoBERTa-large outperforms in most cases. In fact, even when the base model is trained with a LoRA-rank three times larger, RoBERTa-large still performs better on 6 out of 8 datasets. Given that the pretrained RoBERTa-large model was performing no differently from the base model, this observation supports our theoretical findings that deeper models are more expressive with LoRA training.

---

> ### Author Response · Authors · 2023-11-21
> **(Major Update #1)  Bridging Theoretical Insights and Empirical Observations: Continued**
>
> 2. **When the frozen model is closer to the target model, a lower LoRA-rank is sufficient to attain the desired performance.**
>
>     * Our Theoretical Results: supported by Lemma 1, 2, and Theorem 5, 6
>     * Empirical Observation: We have conducted new experiments to further support this claim. The following discussion has been incorporated into Sec. G.8 of our revised version.
>
>         In these experiments, we compared the performance of pretrained RoBERTa-base with that of randomly initialized RoBERTa-base. The results clearly show that the pretrained model requires lower LoRA-ranks to achieve the desired performance, which aligns with our theoretical findings.
>
>         | Dataset |  MNLI | MNLI |  SST-2 | SST-2 |  MRPC | MRPC | CoLA | CoLA |  QNLI | QNLI | QQP | QQP | RTE | RTE | STS-B | STS-B |
>         |---------|--------|---|--------|---|-------|---|-------|---|-------|---|------|---|------|---|--------|---|
>         | Model   | random | pretrained | random | pretrained | random | pretrained | random | pretrained | random | pretrained | random | pretrained | random | pretrained | random | pretrained |
>         | R=2 | .523 | **.861** | .775 | **.950** | .691 | **.892** | .154 | **.632** | .627 | **.928** | .761 | **.891** | .542 | **.780** | .213 | **.907** |
>         | R=4 | .535 | **.868** | .788 | **.950** | .696 | **.890** | .145 | **.634** | .625 | **.929** | .768 | **.898** | .542 | **.805** | .224 | **.910** |
>         | R=6 | .544 | **.868** | .799 | **.948** | .696 | **.892** | .154 | **.629** | .632 | **.931** | .768 | **.900** | .542 | **.773** | .210 | **.909** |
>
> 3. **LoRA outperforms tuning of the final layers if the quality of the shared representation is suboptimal.**
>
>     * Our Theoretical Results: This is supported by Corollary 4 and our newly added Lemma 4 in Sec. 3.3.
>
>         **Lemma 4**: Let $D \geq 2$ and $\overline{f}$ be a one-layer target FNN. Assume that the elements of weight matrices $(W_l)_{l=1}^L$ are independently drawn from arbitrary continuous distributions. With probability 1, for any tuning of the last $L-1$ layers, the frozen model cannot be fine-tuned to match the target model, i.e., $f\neq\overline{f}$.
>
>         This lemma highlights the limitations of final-layer tuning in a randomly initialized model for simple tasks achievable by a one-layer FNN. Conversely, Corollary 4 shows that LoRA can fine-tune a randomly initialized frozen model to align with any smaller target model, given a small rank requirement.
>
>     * Empirical Observation: This finding aligns with `Kornblith et al., (2019)`, emphasizing that the success of final layer tuning depends on the quality of the model's initial layers.
>
>         We also added a simulation result comparing LoRA and final layer tuning with varying tunable parameters, as shown in Figure 6a. This validates our theory that, with equal tunable parameters, LoRA surpasses final layer tuning when the frozen model is randomly generated and the shared representation is poor.
>
> 4. **In addition to applying low-rank updates to weight matrices, updating the bias is also crucial.**
>
>     * Our Theoretical Results: The update of biases plays a significant role in our proofs (see Sec. 3.2 and E.1).
>     * Empirical Observation: `Hu et al. (2022)` recommend bias updates during LoRA training (see their 6th footnote). Our added toy experiments (Figure 6b in Section G.4) further examine this claim, underscoring the role of tunable bias vectors.
>
> 5. **Tuning only the attention weights is sufficient to achieve good performance on TFNs.**
>
>      * Our Theoretical Results: This finding is supported by Theorem 7.
>      * Empirical Observation: `Hu et al. (2020)` recommend applying LoRA only to the attention weights in Sec. 4.2.
>
> 6. **Current optimization algorithms for LoRA training might be suboptimal.**
>     * Empirical Observation: Our current and newly added simulations, illustrated in Figures 4, 5, and 9, all support this claim. Thus, a more advanced algorithm could potentially lead to improved results, suggesting a promising avenue for future exploration and development in this field.

---

> > ### Author Response · Authors · 2023-11-21
> > **(Major Update #2) Illuminating the Optimization and Generalization Aspects of LoRA**
> >
> > ### **(Major Update #2) Illuminating the Optimization and Generalization Aspects of LoRA**
> >
> > In our initial submission, we primarily focused on the expressive power of LoRA, and didn't include any results on generalization and optimization. However, following the insightful suggestions from the reviewers, we have now added simulation results and discussion materials. These additions will help readers gain a more comprehensive understanding of LoRA's properties and potential future research directions in this field.
> >
> > * Optimization (Fig. 7: https://hackmd.io/_uploads/HkOP_USV6.png)
> >
> >     Our simulations show that models with smaller ranks (e.g., rank=1,4) converge swiftly due to their limited search space, but they settle at a relatively high training loss. Medium rank models (e.g., rank=8) converge more slowly. Highly overparameterized models (e.g., rank=13,18) appear to converge faster, aligning with recent advancements in optimization theory, which suggest that overparameterized models are easier to optimize.
> >
> >
> > * Generalization (Fig. 8: https://hackmd.io/_uploads/rJYnRzO4p.png)
> >
> >
> >     To assess LoRA's generalization performance, we fine-tuned the frozen model on the training set and reported the training and test MSE. We noticed an increasing generalization gap (test MSE - train MSE) as the LoRA rank increases -- this is very evident with L=2, and less so with L=4. This is intuitive as larger LoRA ranks imply a larger hypothesis class (e.g., the Rademacher complexity), so it's expected. We defer a detailed analysis of LoRA's generalization performance to future work, but believe our simulation results provide a valuable starting point for further discussion and investigation.

---

### Meta-Review · Area_Chair_kAT2 · 2023-12-07

**Metareview:**

The paper provides an initial theoretical exploration of the popular parameter-efficient finetuning method LoRA. It proves for fully connected models, LoRA should be sufficient to finetune any base model for a smaller target model with a certain LoRA rank (threshold). They further provide approximation errors for the case when the rank is smaller than the threshold.
All reviewers agree that this paper makes important and timely contributions to this emerging problem. The AC also agrees and thus recommends acceptance.

**Justification For Why Not Higher Score:**

This paper only studies the approximation power, but not optimization and generalization properties.

**Justification For Why Not Lower Score:**

This work initiates the study of the approximation power of LoRA, which is an important topic.

---

### Decision · Program_Chairs · 2024-01-16

Accept (poster)